# CryoET of β-amyloid and tau within postmortem Alzheimer's disease brain

Madeleine A. G. Gilbert[1,7], Nayab Fatima[1,7], Joshua Jenkins[1,7], Thomas J. O'Sullivan[2,7], Andreas Schertel[3], Yehuda Halfon[2], Martin Wilkinson[4], Tjado H. J. Morrema[5], Mirjam Geibel[3], Randy J. Read[6], Neil A. Ranson[4], Sheena E. Radford[4], Jeroen J. M. Hoozemans[4] & René A. W. Frank[1✉]

A defining pathological feature of most neurodegenerative diseases is the assembly of proteins into amyloid that form disease-specific structures[1]. In Alzheimer's disease, this is characterized by the deposition of β-amyloid and tau with disease-specific conformations. The in situ structure of amyloid in the human brain is unknown. Here, using cryo-fluorescence microscopy-targeted cryo-sectioning, cryo-focused ion beam-scanning electron microscopy lift-out and cryo-electron tomography, we determined in-tissue architectures of β-amyloid and tau pathology in a postmortem Alzheimer's disease donor brain. β-amyloid plaques contained a mixture of fibrils, some of which were branched, and protofilaments, arranged in parallel arrays and lattice-like structures. Extracellular vesicles and cuboidal particles defined the non-amyloid constituents of β-amyloid plaques. By contrast, tau inclusions formed parallel clusters of unbranched filaments. Subtomogram averaging a cluster of 136 tau filaments in a single tomogram revealed the polypeptide backbone conformation and filament polarity orientation of paired helical filaments within tissue. Filaments within most clusters were similar to each other, but were different between clusters, showing amyloid heterogeneity that is spatially organized by subcellular location. The in situ structural approaches outlined here for human donor tissues have applications to a broad range of neurodegenerative diseases.

Alzheimer's disease (AD) is defined neuropathologically by the abnormal accumulation of aggregated Aβ peptides and tau that form extracellular and intracellular amyloid deposits, respectively[2]. Inherited forms of AD (familial Alzheimer's disease, FAD) are caused by autosomal dominant mutations in the amyloid precursor protein (*APP*) and presenilin (*PSEN1* and *PSEN2*) genes[3,4]. The *APP* gene encodes the precursor of Aβ, whereas the *PSEN1/PSEN2* genes encode subunits of the γ-secretase complex that catalyse the final step of Aβ peptide production. Aβ peptides of varying lengths (including $Aβ_{1-40}$, $Aβ_{1-42}$ and $Aβ_{1-43}$) are produced, of which $Aβ_{1-42}$ is the major constituent of AD β-amyloid[5]. An immunotherapy raised against Aβ aggregates with the Arctic FAD mutation[6], removes β-amyloid and delays the progression of AD[7]. Mutations in the gene encoding microtubule associated protein tau cause neurodegeneration in the form of frontotemporal dementia and parkinsonism linked to chromosome 17, which result in tau pathology that lacks β-amyloid deposits. In AD, the spread of aggregated tau correlates with neuronal loss and the sequence of cognitive decline[8,9].

Aβ and tau are highly aggregation prone, self-assembling into low-molecular weight oligomers or protofibrils that precede the formation of larger Aβ fibrils and tau filaments[10,11]. Over decades, Aβ fibrils and tau filaments accumulate to form amyloid plaques and tau tangles in the parenchyma of the AD brain[12]. β-Amyloid plaques have been morphologically categorized as diffuse, dense-cored, fibrillar or neuritic[13,14], all of which contain $Aβ_{1-42}$ fibrillar deposits[15,16]. In addition, amyloid fibrils composed primarily of $Aβ_{1-40}$ accumulate in and around blood vessels in various types of cerebral amyloid angiopathy[12,17]. By contrast, tau filaments deposit within neuronal cell bodies and neurites forming tau tangles and tau threads, respectively[15,16]. At later stages, tau filaments can reside extracellularly in the form of ghost tangles and the remnants of atrophic neurites[15,18,19].

The structure of $Aβ_{1-42}$ fibrils purified from postmortem AD brain have recently been solved to high resolution using single-particle cryo-electron microscopy (cryo-EM)[20]. These ex vivo fibrils contain two structural conformers of $Aβ_{1-42}$ amyloid (type I and II), both of which are found in sporadic and FAD cases[20]. These structures differ from $Aβ_{1-42}$ fibrils prepared in vitro[21] and from $Aβ_{1-40}$ purified from the meninges of cerebral amyloid angiopathy cases[22,23]. Atomic structures of tau filaments purified from AD[24,25] and other tauopathies[26-29] suggest that tau forms disease-specific conformers. In AD, tau forms distinct ultrastructural polymorphs of paired helical filaments (PHF) and straight filaments (SF), both composed of three-repeat (3R) and four-repeat (4R) tau[24]. However, the native molecular architecture and

[1]Astbury Centre for Structural Molecular Biology, School of Biomedical Sciences, Faculty of Biological Sciences, University of Leeds, Leeds, UK. [2]Astbury Biostructure Laboratory CryoEM facility, Astbury Centre for Structural Molecular Biology, Faculty of Biological Sciences, University of Leeds, Leeds, UK. [3]ZEISS Microscopy Customer Center Europe, Carl Zeiss Microscopy GmbH, Oberkochen, Germany. [4]Astbury Centre for Structural Molecular Biology, School of Molecular and Cellular Biology, Faculty of Biological Sciences, University of Leeds, Leeds, UK. [5]Department of Pathology, Unit Neuropathology, Amsterdam University Medical Centers, Amsterdam, The Netherlands. [6]Department of Haematology, Cambridge Institute for Medical Research, University of Cambridge, Cambridge, UK. [7]These authors contributed equally: Madeleine A. G. Gilbert, Nayab Fatima, Joshua Jenkins, Thomas J. O'Sullivan. ✉e-mail: r.frank@leeds.ac.uk

organization of Aβ and tau pathology within unfixed, human brain tissue remains unknown.

We recently reported the in-tissue molecular architecture of Aβ$_{1-42}$ fibrils in a mouse model of FAD by cryo-correlated light and EM (cryo-CLEM) and cryo-electron tomography (cryoET) of tissue cryo-sections, identifying that these β-amyloid plaques are composed of fibrils, protofilament-like rods and branched amyloid, interdigitated by extracellular vesicles, extracellular droplets and multilamellar bodies[30]. The extent to which this pathological architecture is representative of β-amyloid plaques in human AD brain is unknown. Furthermore, FAD mouse models of β-amyloidosis do not recapitulate the full spectrum of AD pathology, including tau inclusions and neurodegeneration[31].

Here, we determined the in-tissue three-dimensional (3D) architectures within β-amyloid plaque and tau pathology from human postmortem AD brain by cryoET. These data were collected using cryo-fluorescence microscopy (cryo-FM) to target specific pathology within cryo-sections and cryo-focused ion beam (FIB)-scanning EM (SEM) lift-out lamellae. Reconstructed tomographic volumes revealed extracellular β-amyloid plaques composed of Aβ fibrils, branched fibrils and protofilament-like rods interlaced with non-amyloid constituents akin to our earlier studies of a murine FAD model[30]. Tau deposits consisted of unbranched filaments that formed clusters situated in intracellular and extracellular regions of the tissue. We determined the in situ structure of tau filaments within each cluster by subtomogram averaging (8.7–31.8 Å resolution), which identified PHFs with variable twist and SFs. Filaments within a cluster were similar to each other, but different between clusters, showing that fibril heterogeneity is spatially organized.

## Clinical history and neuropathology

CryoET was performed on rapid autopsy, freeze–thaw postmortem brain samples of the mid-temporal gyrus from an AD donor and a non-demented donor (postmortem delay 6 h 10 min and 5 h 45 min, respectively, Methods). The AD case was a 70-year-old woman with neuropathologically confirmed diagnosis following a 12 year history of progressive dementia. The donor started to have memory problems at the age of 54, and by the age of 58 was diagnosed with dementia, showing considerable memory deficits and disturbed executive functions. There was no family history of dementia and the genotype for *APOE* was e3/e3. Neuropathological analysis showed abundant amyloid plaques, tau tangles, tau threads and very few cerebral amyloid angiopathy vessels across the mid-temporal gyrus (Fig. 1a and Extended Data Fig. 1a). Ageing-associated somatic and neurite inclusions of TMEM106B were observed (Extended Data Fig. 1a), as reported previously[32]. No α-synuclein or TDP43 inclusions were detected, indicating the absence of pathologies that are associated with other common neurodegenerative diseases (Extended Data Fig. 1a). To assess the AD donor tissue biochemically, sarkosyl-insoluble tau aggregates were purified and immunoblotted (Fig. 1b). Tau aggregates were hyperphosphorylated and contained both 3R and 4R tau (Extended Data Fig. 1b). Cryo-EM of sarkosyl-insoluble aggregates resolved PHF and SF in the AD donor brain[24,25] (Extended Data Fig. 1c–i). Overall, this immunohistochemical and biochemical profile is typical for AD cases[12,25].

To detect amyloid deposits in AD tissue, acute brain slices were incubated in methoxy-X04 (MX04), a fluorescent label that generically binds amyloid[33]. Confocal microscopy showed wide-spread distribution of MX04-labelled amyloid, including 30–50-μm-diameter deposits characteristic of β-amyloid plaques, as well as tau tangles and tau-filled neuropil threads (Fig. 1c). Immunofluorescence detection of Aβ/APP and phospho-tau confirmed the identity of these morphologically distinct MX04-labelled deposits (Fig. 1c and Extended Data Fig. 2a).

## In situ cryo-CLEM of postmortem brain

To locate β-amyloid and tau pathology within frozen postmortem tissue, we adapted a workflow from our earlier cryoET studies of brain tissue from a FAD mouse model[30]. A frozen tissue block was thawed, acute slices prepared, labelled with MX04 and high-pressure frozen (HPF) (Fig. 1d). Cryo-FM of these vitrified postmortem AD tissues revealed the location of MX04-labelled amyloid pathology, including neuritic plaques (Fig. 1e), tau tangles (Fig. 1f) and threads (Fig. 1e,f), which resembled those in fixed tissue (Fig. 1c and Extended Data Fig. 2b). Amyloid pathology was absent in MX04-labelled non-demented control tissue (Extended Data Fig. 2b).

To prepare postmortem brain for cryoET, roughly 70-nm-thick tissue cryo-sections were collected (Extended Data Fig. 2c) from a MX04-labelled β-amyloid plaque (Fig. 1e) and from a second location enriched in tau tangles and threads (Fig. 1f). Cryo-FM confirmed the presence of MX04-labelled amyloid within these tissue cryo-sections (Extended Data Fig. 1d). MX04-labelled amyloid was mapped by cryo-CLEM onto medium magnification electron micrographs (Fig. 1g). In regions with strong MX04 signal, putative amyloid was directly observed by cryo-EM (Extended Data Fig. 3a,b).

Cryo-CLEM maps were used to target the collection of tomographic tilt series, each encompassing a roughly 1 μm$^2$ area of the tissue cryo-section (2.38 Å pixel size). We collected 42 and 25 tomograms in and around regions of cryo-sections that contained a MX04-labelled β-amyloid plaque (Supplementary Table 1) and tau tangles (Supplementary Table 2), respectively. An extra 64 tomograms collected from non-demented donor tissue cryo-sections were used as a control (Supplementary Table 3). Reconstructed tomographic volumes revealed the native, in-tissue, 3D molecular architecture of AD pathology in postmortem brain (Figs. 1h,i and 2, also Supplementary Videos 1–3 from an amyloid plaque, Supplementary Videos 4–6 from tau tangles and threads and Supplementary Videos 7–9 from non-demented control).

## In-tissue architecture of β-amyloid and tau

Fibrils were apparent in all tomographic volumes collected from the MX04-labelled β-amyloid plaque (Supplementary Table 1). No fibrils were present in any of the control donor tomograms (n = 64, Supplementary Data Table 3). The fibrils within the β-amyloid plaque formed parallel arrays or a lattice (Fig. 1h,i and Extended Data Fig. 3c–e). This architecture is similar to that we reported in the amyloid plaques of an FAD mouse model[30]. In a subset of tomograms, the extracellular location of the amyloid fibrils could be determined unambiguously by its juxtaposition to subcellular compartments enclosing intracellular organelles or a myelinated axon (Extended Data Fig. 3d and Supplementary Table 1).

β-Amyloid plaques were interdigitated by various membrane-bound subcellular compartments and associated macromolecular constituents (Fig. 1h,i, Extended Data Fig. 3c–e and Supplementary Data Table 1). These included a prevalence of extracellular vesicles, extracellular droplets, 70–200 nm cuboidal particles and membrane fragments (open sheets of lipid bilayer). These non-amyloid constituents were absent from non-demented donor control tissue tomograms (Extended Data Fig. 4 and Supplementary Table 3). Except for extracellular cuboidal particles and membrane fragments, this repertoire of constituents was comparable to those we observed previously by cryoET in FAD mouse model tissue cryo-sections (*App*$^{NL-G-F}$-HPF)[30]. The presence of open membrane fragments could be a consequence of the postmortem interval (PMI) or freeze–thaw step[34]. To control for this, we prepared tissue cryo-sections of FAD mouse model tissue with the same PMI and freeze–thaw step as the postmortem human donor tissue (*App*$^{NL-G-F}$-PMI-FT-HPF), from which we collected a cryoET dataset (Supplementary Table 7) and compared it to tomograms from tissue that was directly prepared (no PMI or freeze–thaw step)

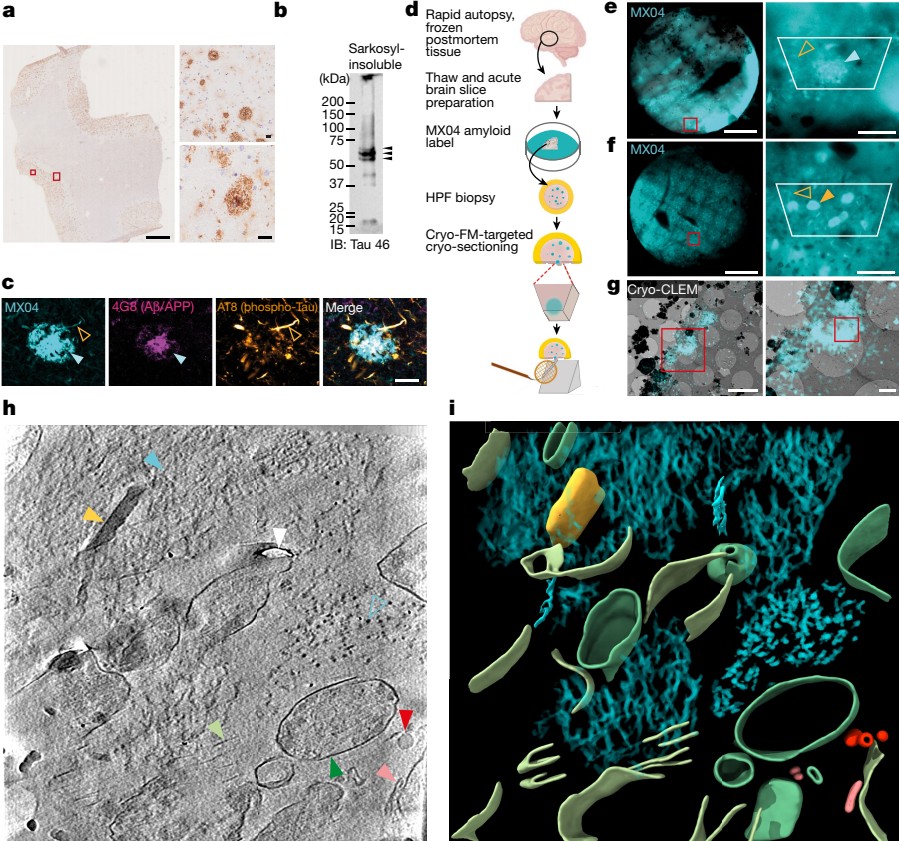

**Fig. 1 | In situ cryoET of vitrified postmortem AD brain.**
**a**, Immunohistochemical detection of Aβ/APP pathology in postmortem AD donor. Scale bar, 2 mm. Large and small red rectangles, indicate close-up images shown in upper- and lower-right panels, respectively. Scale bar, 20 μm. **b**, Immunoblot detection of sarkosyl-insoluble tau (tau 46). Arrowheads, indicate full-length phospho-tau bands. **c**, Fluorescence confocal microscopic detection (left to right) of amyloid (MX04), Aβ/APP (4G8), phospho-Tau (AT8) and merged in an AD postmortem donor brain. Cyan arrowhead, β-amyloid plaque; open orange arrowhead, tau thread. Scale bar, 20 μm. **d**, Schematic showing the preparation of AD postmortem brain for in situ structure determination by cryo-CLEM and cryoET. Schematic adapted from ref. 30, Springer Nature Limited). **e**, Left, cryo-FM of HPF AD postmortem brain biopsy. Cyan, MX04 fluorescence; red box, region shown in close up. Scale bar, 0.5 mm. Right, close up. Cyan arrowhead, putative β-amyloid plaque; open orange arrowhead, putative tau thread; white trapezium, area encompassing tissue from which tissue cryo-sections were collected. Scale bar, 50 μm. **f**, Same as **e** but showing tau tangle and threads indicated by closed and open orange arrowheads, respectively. **g**, Left, cryo-CLEM targeting of MX04-labelled β-amyloid plaque in tissue cryo-section. Red rectangle, region shown in close up. Scale bar, 1 μm. Right, close up. Red rectangle, location from which cryoET data were collected (Supplementary Video. 2). Scale bars, 5 μm (left) and 1 μm (right). **h**, Tomographic slice of β-amyloid pathology in postmortem AD brain cryo-section. Filled and open cyan arrowhead, fibril in the *x*–*y* plane and axially (*z* axis) of the tomogram, respectively; yellow arrowhead, extracellular cuboidal particle; red arrowhead, extracellular droplet; pink arrowhead, extracellular vesicle; dark green arrowhead, subcellular compartment; light green arrowhead, burst plasma membrane compartment; white arrowhead, knife damage. Scale bar, 10 nm. **i**, Segmentation of tomogram coloured as in **h**.

($App^{NL-G-F}$-HPF)[30]. Although the architecture of β-amyloid fibrils in $App^{NL-G-F}$-PMI-FT-HPF was indistinguishable from $App^{NL-G-F}$-HPF, a small subset of $App^{NL-G-F}$-PMI-FT-HPF tomograms contained membrane fragments (2 out of 60) or burst plasma membrane (10 out of 60), both of which were absent in $App^{NL-G-F}$-HPF samples[30] (Extended Data Figs. 5 and 6). Therefore, although no membrane fragments or burst plasma membrane were observed in non-demented control postmortem tissue (5 h 40 min PMI and freeze thawed, Supplementary Table 3 and Extended Data Fig. 4), we cannot exclude the possibility that these arise as a consequence of the PMI and freeze thawing.

Extracellular cuboidal particles in β-amyloid plaques of postmortem AD brain contained distinctive internal features in the form of regularly spaced striations of higher density (Extended Data Fig. 7). Extracellular cuboidal particles were not observed in $App^{NL-G-F}$ plaques[30]. Fourier analysis indicated these striated layers were spaced 2.5 or 2.8 nm apart (Extended Data Fig. 7). This morphology was reminiscent but different from 3.5 nm layers observed by cryo-EM of low-density lipoprotein particles[35] and cryoET of intracellular lipid droplets[36].

Tomograms collected from MX04-labelled tau-containing tissue cryo-sections revealed filaments arranged as 300–800 nm parallel clusters (Fig. 2, Extended Data Fig. 8a and Supplementary Table 2), which were absent in control tissue cryo-sections (Supplementary Table 3 and Extended Data Fig. 4). These deposits were within the cytoplasm of neurites and within myelinated axons (Fig. 2a,b). A cluster of this amyloid was also found extracellularly, located next to a damaged mitochondrion outside a myelinated axon, and without any evidence of an enclosing plasma membrane in the vicinity (Fig. 2c,d). By contrast, damaged mitochondria, frequently observed in non-demented control and $App^{NL-G-F}$-PMI-FT-HPF tomograms (Extended Data Fig. 6 and Supplementary Tables 3 and 7), were all completely or partially enclosed by plasma membrane. Nonetheless, we could not determine definitively whether or not the extracellular location of this tau filament cluster and mitochondrion was a consequence of sample preparation.

To assess further the architecture and identity of fibrils in β-amyloid plaques and tau deposits, populations from each were compared. The cross-section of fibrils from β-amyloid plaques and tau pathology-containing tomograms showed fibrils and filaments of 6 ± 2 and

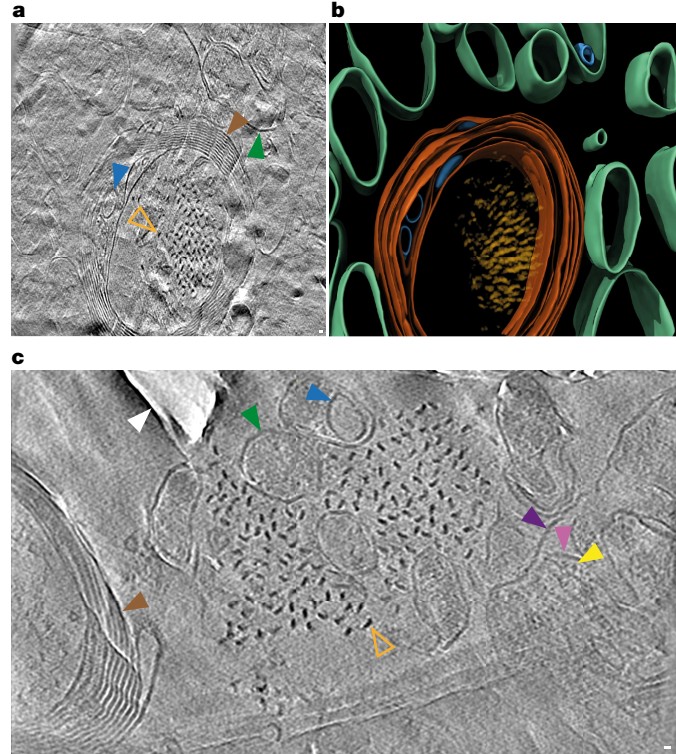

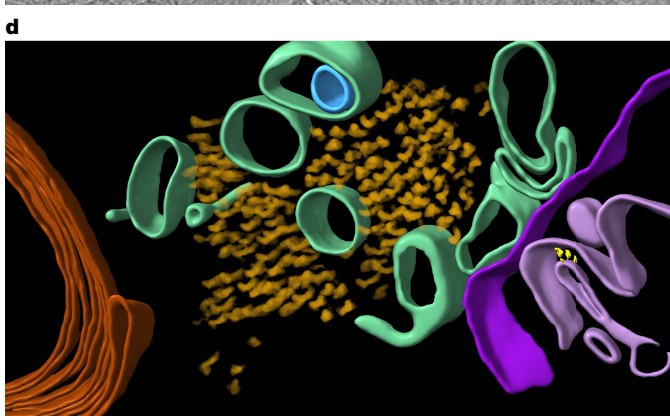

**Fig. 2 | In situ cryoET of tau deposits in vitrified postmortem AD brain.**
**a**, Tomographic slice of intracellular tau pathology in postmortem AD brain cryo-section (Supplementary Video 4). Open orange arrowhead, filament oriented axially (*z* axis) within the tomogram; brown arrowhead, myelinated axon; green arrowhead, subcellular compartment; blue arrowhead, intracellular membrane-bound organelle. Scale bar, 10 nm. **b**, Segmentation of tomogram coloured as in **a**. **c**, Tomographic slice of extracellular tau pathology in AD postmortem brain cryo-section (Supplementary Video 5). Open orange arrowhead, tau filament oriented axially (*z* axis) within the tomogram; brown arrowhead, myelinated axon; dark and light purple arrowheads, outer and inner membranes of damaged mitochondrion, respectively; yellow arrowhead, putative Fo-F1 ATPase; dark green arrowhead, subcellular compartment; blue arrowhead, intracellular membrane-bound organelle; white arrowhead, knife damage. Scale bar, 10 nm. **d**, Segmentation of tomogram coloured as in **c**.

15 ± 3 nm (mean ± s.d.) maximum diameter, respectively (Fig. 3a,b and Extended Data Fig. 8b). These widths were broadly consistent with the expected dimensions of atomic structures of ex vivo purified β-amyloid fibrils[20] and tau filaments[24], respectively, and confirm the MX04-targeted collection of tomographic volumes from β-amyloid plaque and tau inclusions. Aβ fibrils were oriented both in the *x*–*y* plane and axially in the *z* direction of the tomographic volume (Fig. 1h and Extended Data Fig. 3c–e). By contrast, the tau filaments were oriented either in the *x*–*y* plane or axially, but not mixed (Figs. 2 and 3

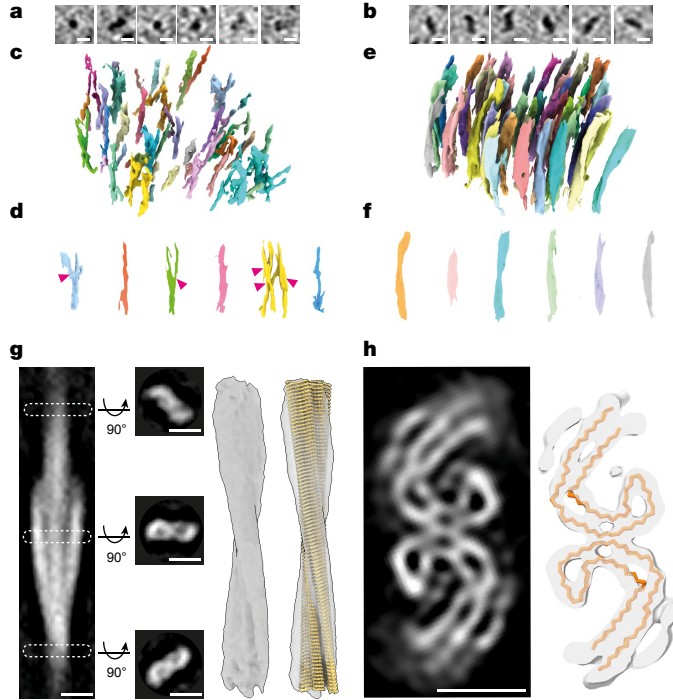

**Fig. 3 | In-tissue architecture of β-amyloid fibrils and tau filaments, and subtomogram averaging of tau filaments within postmortem AD brain.**
**a**, Tomographic slices showing a panel of β-amyloid fibrils oriented axially (*z* axis) within the tomogram. Scale bar, 10 nm. **b**, Tomographic slices showing a panel of tau filaments oriented axially (*z* axis) within the tomogram. Scale bar, 10 nm. **c,d**, Side views of raw tomographic density containing a lattice of β-amyloid (**c**) and individual fibrils (**d**). Each individual amyloid fibril is coloured differently: 33% (17 out of 51) of β-amyloid fibrils had branch points. Magenta arrowhead, branch point. **e,f**, As in **c** (**e**) and **d** (**f**), but for tau. **g**, Subtomogram average of 136 tau filaments (stalkInit, Methods) located extracellularly from one tomogram (Fig. 2c,d and Supplementary Video 5). The left panel shows a side view of tomographic slice through averaged volume showing helical twist. White dashed rectangles, position of middle left panels along filament axis of three top view tomographic slices (23.75 nm apart) showing a pair of C-shaped protofilaments consistent with the substructure of ex vivo purified tau PHF[24]. The middle right and right panels show a subtomogram average map of tau filament with and without atomic model of ex vivo purified tau PHF (yellow, PDB 5o3l)[24] fitted into the map, respectively. Scale bar, 10 nm. **h**, Helical averaging of tau filament subvolumes from one AD cryo-section tomogram (Fig. 2c,d and Supplementary Video 5) (8.7 Å resolution at FSC 0.143, Extended Data Fig. 8c). Left, slice through averaged subvolume. Right shows a Cα trace of ex vivo purified tau PHF atomic model PHF (yellow, PDB 5o3l)[24] fitted using EM placement[37,38] into averaged map. Scale bar, 5 nm.

and Extended Data Fig. 8a). Axially oriented Aβ fibrils and tau filaments provided sufficient contrast to observe directly individual fibrils and tau filament crossovers throughout the raw tomographic volume (Fig. 3). In such regions of β-amyloid plaque tomograms, 33–40% of fibrils showed branch points, in which fibrils bifurcated from one another (Fig. 3c,d). This branched amyloid architecture was similar to that in an FAD mouse model of β-amyloidosis[30]. In tau tomograms, only unbranched filaments were present (Fig. 3e,f).

## In-tissue structure of tau filaments

To obtain higher resolution structural information we performed subtomogram averaging of amyloid in tau tangles and β-amyloid plaques. The subcellular environment within different locations of the tissue could influence the structure of the filaments. Therefore, we aligned and averaged subvolumes from each cluster of tau filaments

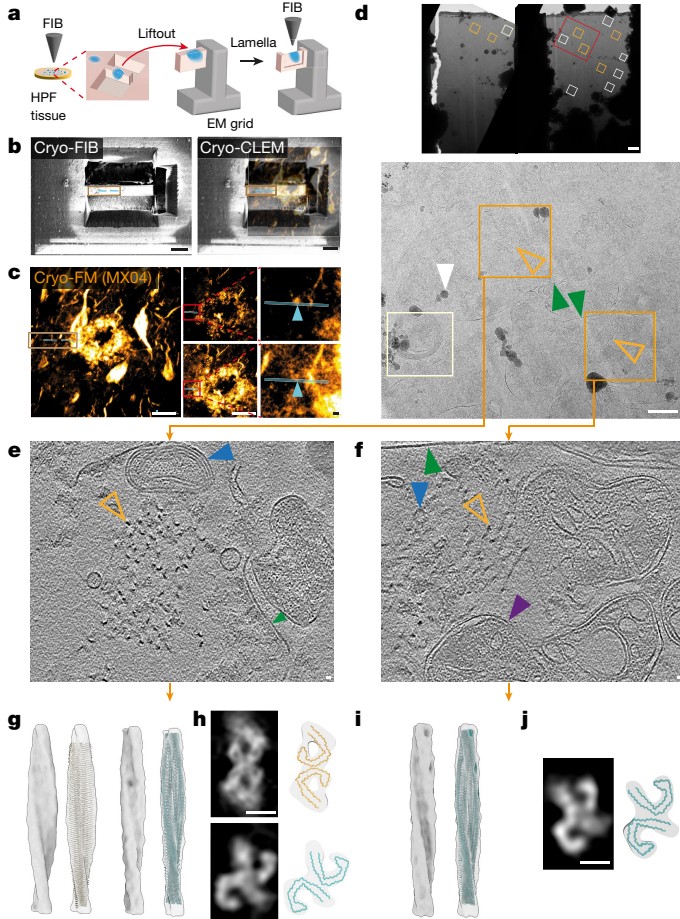

**Fig. 4 | cryo-CLEM-targeted cryo-FIB-SEM lift-out lamellae of tau thread in AD brain. a**, Schematic summarizing cryo-CLEM-targeted cryo-FIB-SEM lift-out lamellae preparation of MX04-labelled amyloid (blue) from HPF brain. **b**, Left, cryo-FIB image of HPF brain showing tissue chunk before lift-out. Right, cryo-FIB image aligned with confocal cryo-FM of MX04-labelled amyloid. Brown rectangle, tissue chunk; cyan line, locations of tissue lamella. Scale bar, 20 μm. **c**,**e**,**f**, The left shows MX04 confocal cryo-FM of HPF tissue targeted for preparation of lift-out lamellae. The middle shows cryo-FM optical *z* slices 1.9 μm apart. Scale bar, 20 μm. Red rectangles, regions in close-ups. The right shows close-ups. Cyan line, location of tissue lamella; cyan arrowhead, microscopic regions of MX04-labelled amyloid corresponding to locations above and below the first (**e**) and second (**f**) tomograms. Scale bar, 1 μm. **d**, Top, cryo-EM of two lift-out lamellae. Red rectangle, region enlarged below. Scale bar, 1 μm. Bottom, close up. Orange rectangle, tomograms of tissue lamella containing tau filaments; orange arrowhead, tau filament cluster; green arrowhead, plasma membrane-bound subcellular compartment; white arrowhead, ice contamination; white rectangle, location of tomogram lacking tau filaments. Scale bar, 500 nm. **e**,**f**, Tomographic slices of tau thread in tissue lamella. Orange arrowhead, tau filament; purple arrowhead, mitochondrion; green arrowhead, membrane enclosing subcellular compartment; blue arrowhead, intracellular membrane-bound organelle. Scale bar, 10 nm. **g**, Subtomogram averaging of 52 PHF and 19 SF. Left and middle left, averaged maps without and with tau PHF atomic model (yellow, PDB 5o3l)[24] fitted into the subtomogram average map, respectively. Middle right and right, same as left and middle left but for SF without and with atomic model of ex vivo purified SF (cyan, PDB 5o3t)[24]. **h**, Helical averaging of tau filament subvolumes. Top and bottom panels, PHF and SF maps, respectively. Top left, slice through averaged subvolume. Top right, tau PHF Cα trace (yellow, PDB 5o3l) fitted using EM placement[36,37] into an averaged map. Bottom left and right, same as top but for SF (cyan, PDB 5o3t)[24]. Scale bar, 5 nm. **i**,**j**, As in **g** (**i**) and **h** (**j**) but for the neighbouring SF only cluster.

independently. Using helical parameters for subtomogram averaging that were obtained by aligning whole filaments (Fig. 3g), the highest resolution average (8.7 Å, Extended Data Fig. 8c) was obtained from an

extracellular cluster of 136 tau filaments in a single tomogram, resolving β-sheets and loops formed by the backbone of each polypeptide. These revealed the AD fold of two C-shaped protofilaments (Fig. 3h). We quantified the fit of atomic models in the subtomogram average map by the log-likelihood gain (LLG) score obtained by 'EM placement'[37,38], in which the best model will have the largest score and a score greater than 60 indicates a non-random fit. The AD PHF model[24] was well accommodated: 944.6 LLG and 0.56 correlation coefficient compared to 234.6 LLG and 0.40 correlation coefficient for AD SF (Fig. 3h and Supplementary Table 5). An extra partial shell of weaker, β-sheet-like density was unoccupied by the PHF atomic model (Fig. 3h). Similar extra density was reported in the unsharpened map of a sarkosyl-extracted AD PHF tau filament model[24,25].

Subtomogram averages from six other tau clusters within the tissue also produced lower-resolution maps (18.9–31.8 Å resolution), two of which were compatible with AD PHF on the basis of atomic model fit (LLG > 60, Extended Data Fig. 9a–c and Supplementary Table 5). The remaining four tau clusters did not accommodate PHFs or SFs and higher resolution features that could unambiguously indicate a specific tau amyloid fold were absent (LLG < 60 for PHF and SF atomic models, Extended Data Fig. 9d–f and Supplementary Table 5). Subtomogram averaging of fibrils in β-amyloid plaque tomograms that are thinner and less featured than tau filaments could not resolve the helical twist, and so high-resolution subtomogram averaging was not possible (Extended Data Fig. 10).

## cryo-FIB-SEM lift-out of AD brain

The achievable resolution of subtomogram averaging in cryo-sections could be limited by knife damage during sample preparation[39]. Therefore, we established an alternative workflow to prepare tissue lamella[40] from postmortem AD brain by cryo-FIB-SEM lift-out (Fig. 4a). 3D cryo-FM was used to target the preparation of tissue lamellae to a location containing tau threads that were situated adjacent to a MX04-labelled amyloid plaque (Fig. 4b,c and Extended Data Fig. 11). Thirteen tomograms were collected throughout the lamellae, six of which contained tau clusters (Fig. 4d).

Subtomogram averaging was performed with two tomograms that contained the highest copy number of tau filaments. These two tau clusters were separated by roughly 1 μm within two membrane-bound subcellular compartments and were mapped to two distinct MX04-labelled tau threads located at the periphery of an MX04-labelled amyloid plaque (Fig. 4c–f). Aligning filaments from each location independently produced structurally distinct subtomogram averages (Fig. 4g–h compared to Fig. 4i,j). In the first cluster of 71 filaments, two different classes were identified (18.1 and 18.2 Å resolution, Extended Data Fig. 12a), in which the atomic models of AD PHF and SF[24,25] were well accommodated (Fig. 4h, PHF model fit: 379.0 LLG, 0.48 correlation coefficient and SF model fit: 233.5 LLG, 0.48 correlation coefficient). The second cluster of 78 filaments was composed entirely of SF[24,25] (Fig. 4i,j and Extended Data Fig. 12b, 18.3 Å resolution, SF model fit: 180.9 LLG, 0.47 correlation coefficient). These in situ cryoET data highlight the co-existence of several distinct ensembles of tau filaments (a mixed cluster of PHF and SF compared to an SF only cluster) organized within two neighbouring microscopic regions of pathology.

## Tau filament twist and orientation

Our dataset of subtomogram averages enabled us to assess the similarity of tau filaments across anatomically distinct in-tissue locations. Comparing the helical parameters of each tau filament cluster with that of the ex vivo purified atomic model of PHF[24] indicated that in situ tau filaments showed location-specific variability in helical twist (Extended Data Fig. 12c and Supplementary Table 5). Half the in-tissue PHF subtomogram averages were 19–38% less twisted than

that of previously reported ex vivo PHFs (79–129 versus 65–80 nm crossover distance of in situ PHF filaments versus ex vivo PHF[24], respectively).

Three of the tau cluster subtomogram averages were of sufficiently high resolution to resolve the polarity of tau filaments. Mapping these structures back into the in-tissue tomographic map showed that within a parallel tau cluster, the orientation of filament polarity was the same for 114 PHF filaments versus 22 filaments that were facing in the opposite direction (Extended Data Fig. 12d). This highly skewed distribution was non-random (binomial distribution, $P_{x\geq114} = 1.8 \times 10^{-16}$, $n = 136$). No obvious pattern in the location of filaments of the same versus opposite orientation was apparent. The polarity orientation of filaments in a lift-out tomogram containing a cluster of both PHF and SF filaments also indicated a skewed distribution (binomial distribution, $P_{x\geq45} = 0.0014$, $n = 51$ PHF and $n = 19$ SF, Extended Data Fig. 12e). By contrast, a SF only tau cluster contained filaments with a near random distribution of polarity orientations (binomial distribution, $P_{x\geq29} = 0.12$, $n = 78$, Extended Data Fig. 12f). These data suggest that in some but not all tau clusters, inter-filament interactions or other cellular constituents influence filament orientation.

## Discussion

Light microscopic characterization of amyloid in AD brain has formed the basis of diagnostic and disease classification over decades. Recent atomic models of ex vivo AD Aβ fibrils and tau filaments prepared by bulk purification from whole brain regions have elucidated fibril and filament conformers specific to AD and other neurodegenerative diseases[1]. Here, using cryo-FM to guide cryoET and subtomogram averaging we delineate a relationship between molecular structure, cellular context and the characteristic pattern of microscopic neuropathology in an AD brain. These in situ structures of β-amyloid plaque and tau pathology revealed the heterogeneity of Aβ fibrils, the location-specific variability in helical twist and polarity orientation of tau filaments within different cellular contexts from a single brain region of an individual postmortem AD donor.

β-amyloid plaques were characterized by a lattice-like architecture of amyloid fibrils interdigitated by non-amyloid constituents, including extracellular vesicles and cuboidal particles, which were absent from non-demented postmortem tissue tomograms. These constituents are consistent with a recent cryoET study of a mouse model of AD[30] and plastic-embedded EM of AD brain[15,16]. The extracellular cuboidal particles in human postmortem β-amyloid plaques resembled the cuboidal droplet-like architecture of ApoE and pre-melanosomal protein-associated intraluminal vesicles, which are proteins necessary for the amyloid structure of retinal melanosomes[41]. We suggest these non-amyloid constituents are a component of AD pathology, perhaps related to β-amyloid biogenesis[42,43] or a cellular response to amyloid[44].

Aβ fibrils analysed in the postmortem AD human brain were similar to those observed by cryoET in fresh tissue cryo-sections and ex vivo purified amyloid from an FAD mouse model, which included fibrils, protofilament-like rods and branched amyloid[30]. The existence of branched fibrils and protofilament-like rods is suggestive of fibril growth mediated by secondary nucleation mechanisms[45,46] and could contribute to the high local concentration of Aβ that characterizes β-amyloid plaques[30].

In-tissue tomographic volumes of tau pathology indicated filaments were unbranched and arranged in parallel clusters. The limited width of the neuropil in which tau clusters were situated may explain the parallel organization of tau filaments. However, we cannot rule out that lateral interactions during filament biogenesis could contribute to this parallel arrangement. Tau clusters were observed within cells and in extracellular locations. The latter are consistent with 'ghost neurites' that remain after degeneration of the neuron[18].

Plastic-embedded conventional EM of postmortem AD brain showed the first evidence of PHF and SF in different tangles or threads[47,48]. Cryo-EM of purified amyloid identified that two distinct tau ultrastructural polymorphs, PHF and SF, are associated with AD[24,49]. These differ in the symmetric versus asymmetric arrangements of protofilaments within an otherwise twisted filament of tau (2.5 and 2.2° nm$^{-1}$ twists of PHF and SF, respectively)[24]. In-tissue subtomogram averaging reached subnanometre resolution, revealing the backbone polypeptide fold of PHF from a single cluster of 136 filaments within a single cryo-section tomogram. Most other tomograms in our dataset, from either cryo-sections or cryo-FIB-SEM lamellae, produced subtomogram averaged maps that did not resolve the polypeptide fold. Most of these were nonetheless of sufficient resolution to distinguish the protofilament substructure of PHF and SF tau filaments in situ, that originated from distinct clusters in different tissue locations. Each cluster was composed of different ensembles of tau filaments (PHF only, SF only, or a combination of PHF and SF), with variations in the average helical twist. Four tau filament cluster subtomogram average maps accommodated the atomic model of PHF[24] and most showed a decrease in their helical twist compared to ex vivo, sarkosyl-insoluble PHFs[24]. Because PHFs within most clusters were similar to each other, but were different between clusters, we suggest that this variability is spatially restricted or may be organized by subcellular location.

Filamentous proteins possess a polarity whose orientation is a functionally important organizing principle in neurons. For example, microtubules in neuronal axons maintain an almost uniform polarity orientation, whereas in dendrites their orientation is mixed[50]. Polarity could also be an important consideration in tau filament biogenesis and models of secondary nucleation that explain the rapid kinetics of fibril assembly in vitro[51] and potential mechanisms of prion-like spread within tissues[52]. Given that mature tau filaments are several micrometres long, the polarity orientation of each filament must become fixed at an early stage of assembly because once a filament grows longer than the width of the subcellular compartment in which it resides, steric hindrance would prevent it from rotating to point in the opposite direction. Subtomogram averaging resolved a tau cluster with a highly skewed, non-random distribution of polarity orientations favouring one direction over the other. This skewed distribution is consistent with some degree of interaction between filaments[53], templating the growth of new tau filaments at the early stages of its assembly. However, we cannot exclude the possibility that other cellular constituents organize the polarity orientation of tau filament clusters.

Application of the in situ structural workflows reported here to larger cohorts of diverse AD donors, across different brain regions and at earlier stages of AD, may reveal how the spatial organization of amyloid of different structures relates to individual neuropathological profiles. It will also be important to apply these approaches to other neurodegenerative diseases, many of which share related, or overlapping, types of amyloid neuropathology.

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

## Methods

### Data reporting

No statistical methods were used to predetermine sample size. Experiments were not randomized. The investigators were not blinded to allocation during experiments and outcome assessment.

### Donor and ethical information

Postmortem brain tissue was obtained through the Netherlands Brain Bank (Amsterdam, The Netherlands, https://www.brainbank.nl). In compliance with all ethical standards, brain donors signed informed consent regarding the usage of their brain tissue and clinical records for research purposes. This study was performed at Netherlands Brain Bank, Amsterdam University Medical Centres (location VUmc), and the University of Leeds. This study was approved by both VUmc and the University of Leeds Research Ethics Committee. Brain dissection and neuropathological diagnosis were performed according to international guidelines of Brain Net Europe II (BNE) consortium (http://www.brainnet-europe.org) and NIA-AA[54].

Unfixed, flash-frozen, rapid autopsy postmortem AD and non-demented donor postmortem brain were stored at −80 °C and provided a source of tissue for these studies. The control case was a 90-year-old man with a history of depression and prostate cancer. At the age of 87, the donor was admitted to a nursing home. In the last phase, he was passive with a concentration disorder, but he was not demented, with normal language skills, speaking skills and communicative ability. Neuropathological examination revealed slight atrophy of the temporal lobe. No β-amyloid plaques or neurofibrillary tangles were observed in the temporal lobe.

### Neuropathology of donor tissue

A tissue block containing mid-temporal gyrus that was adjacent to the HPF unfixed tissue, was formalin-fixed and paraffin embedded. Sections of 5 μm thickness were prepared and mounted on Superfrost+ microscope slides (VWR). After overnight incubation at 37 °C, slides were deparaffinized using xylene and alcohol and subsequently washed in phosphate buffered saline (pH 7.4).

The histochemical detection of plaques and neurofibrillary tangles was as previously described[55]. In short, tissue was pretreated using 5% w/v periodic acid for 30 min. Subsequently, the tissue was silver impregnated using a 0.035% w/v silver nitrate solution for 30 min. After silver impregnation, the bound silver was developed using a reduction reaction induced by the development solution (2.5% w/v sodium carbonate, 0.1% w/v silver nitrate, 0.5% w/v tungstosilicic acid hydrate, 0.1% w/v ammonium nitrate and 0.1% w/v formaldehyde). The development was stopped by washing in 0.5% w/v acetic acid for 5 min and unbound silver was removed by washing in 5% w/v sodium thiosulfate for 5 min. Sections were counterstained using haematoxylin (Diapath). The sections were dehydrated using alcohol and xylene and coverslipped using Depex (BDH Laboratories Supplies).

For immunohistochemistry, deparaffinized sections were pretreated with 0.3% hydrogen peroxide in phosphate buffered saline for 30 min to block endogenous peroxidase activity, followed by autoclave heating (121 °C for 20 min) in 10 mM sodium citrate buffer (pH 6) for antigen retrieval. Primary antibodies were incubated overnight at room temperature and diluted in antibody diluent (Sigma-Aldrich) as follows: anti-pTauSer202/Thr205 clone AT8 (Thermo Fisher) 1:800, anti-amyloid beta clone 4G8 (Biolegend) 1:1,000, pTau-Thr217 (Thermo Fisher) 1:6,400, P62-lck (BD Biosciences) 1:1,000, anti-alpha-synuclein (phospho-S129) (Abcam) 1:500, and anti-pTDP-43 Ser409/410 (Cosmo Bio) dilution 1:6,000 and anti-TMEM106B (C terminal, Sigma-Aldrich) dilution 1:1,000. Envision mouse/rabbit HRP (DAKO) was used in the secondary detection step, and 3,3′-diaminobenzine (DAKO) was used as a chromogen. Immunostained sections were counterstained using haematoxylin, dehydrated using alcohol and xylene, and coverslipped using Depex.

### Immunohistochemistry and confocal fluorescence microscopy

Free-floating (200 μm) acute brain slices were incubated for 1 h in carboxygenated NMDG buffer to which 15 μM MX04 was diluted. Next, the slices were transferred to fresh NMDG buffer (93 mM NMDG, 2.5 mM potassium chloride, 1.2 mM sodium hydrogen carbonate, 20 mM HEPES, 25 mM glucose, 5 mM sodium ascorbate, 2 mM thiourea, 3 mM sodium pyruvate, 10 mM magnesium sulfate heptahydrate, 0.5 mM calcium chloride dihydrate, pH 7.4, 300–315 mOsmol)[56] and fixed with 4% v/v paraformaldehyde. Slices were permeabilized with 2% v/v Triton X-100 for 30 min and incubated for 1 h in blocking buffer (3% w/v BSA, 0.1% v/v Triton X-100, 50 mM Tris-HCl, 150 mM NaCl, pH 7.4) at room temperature. To detect β-amyloid and tau inclusions, slices were incubated in 1:750 dilution 6E10 (Biolegend, catalogue no. 803001) or 1:750 dilution 4G8 (Biolegend, catalogue no. 803001) and 1:750 dilution AT8 (Thermo Fisher) in blocking buffer at 4 °C for 16 h, respectively. After three washes in TBS (50 mM Tris-HCl, 150 mM NaCl, pH 7.4) for 5 min each, slices were incubated in 1:1,000 diluted antimouse IgG2b AF-633 (Thermo Fisher, catalogue no. A21126) and 1:1,000 diluted anti-mouse IgG1 AF-568 (Thermo Fisher) in blocking buffer for 2 h at room temperature. Following three washes in TBS for 5 min each, slices were mounted with Vectashield (Vector Laboratories) on a microscope slide (Erpredia, catalogue no. J1810AMNZ). Images were acquired with a confocal laser scanning microscope (ZEISS LSM 700) using a ×10/0.3 and a ×63/1.4 numerical aperture (NA) air objective lens, with frame sizes of 1,024 × 1,024 and 512 × 512 pixels, respectively. MX04, AF-568 and AF-633 were detected with excitation and emission maxima of 405 and 435, 579 and 603 and 639 and 669 nm, respectively.

### Sarkosyl-insoluble tau purification and immunoblotting

Sarkosyl-insoluble tau purification followed a previously published protocol[57]. In brief, freeze–thawed postmortem brain tissue was homogenized in 10 vol (w/v) of homogenization buffer (10 mM Tris-HCl (pH 7.4), 0.8 M NaCl, 1 mM EDTA, 10% w/v sucrose). The homogenate was centrifuged at 20,000g for 20 min, at 4 °C and the supernatant was retained. The pellet was rehomogenized in 5 vol (w/v) of homogenization buffer and recentrifuged. Both supernatants were combined, brought to 1% iN-lauroylsarcosinate (w/v) and were centrifuged at 100,000g for 1 h at 21 °C. The sarkosyl-insoluble pellets were resuspended in 50 mM Tris-HCl, pH 7.4 (0.2 ml per g of starting material) and stored at 4 °C for immunoblots. Samples were analysed using 4–12% Bis-Tris gels (Thermo Fisher) and transferred onto polyvinyl difluoride (PVDF) membranes using iBlot gel transfer stacks (Thermo Fisher). The PVDF membrane was blocked (2.5% w/v casein in 0.1% v/v Tween 20, 50 mM Tris-HCl pH 7.4, 100 mM NaCl) for 1 h at room temperature. The following primary antibodies were diluted in 1.25% w/v casein in TBS-T (0.1% Tween 20, 50 mM Tris-HCl pH 7.4, 100 mM NaCl): 1:2,000 Tau 46 (amino acids (aa) 404–441, T9450, Merck), 1:1,000 AT8 (pS202/pT205 Tau, MN1020, Thermo Fisher), 1:1,000 4-repeat tau (aa 275–291, catalogue no. 05-804, Merck), 1:500 3-repeat tau (aa 267–316, catalogue no. 05-803, Merck) and 1:1,000 C-terminal domain TMEM106B (Merck, catalogue no. SAB2106778). The PVDF membranes were incubated with primary antibodies at 4 °C overnight. The membranes were washed five times with TBS-T for 5 min, followed by incubation with secondary antibody for 40 min at room temperature, then washed five times in TBS-T for 5 min. PVDF membranes with ECL reagent (Lumigen) were imaged on an iBright 1500 (Thermo Fisher).

### Single-particle cryo-EM structure of sarkosyl-insoluble tau

The purification of sarkosyl-insoluble tau was as previously described[58]. In brief, 0.435 g of postmortem brain (cingulate gyrus) tissue was homogenized in 20 vol (v/w) of homogenization buffer (10 mM Tris-HCl (pH 7.4), 0.8 M NaCl, 1 mM EGTA, 10% w/v sucrose). The homogenate

was brought to 2% w/v sarkosyl, incubated for 30 min at 37 °C and then centrifuged at 10,000$g$ for 10 min, at 4 °C. The supernatant was retained and centrifuged at 100,000$g$ for 25 min at 4 °C. The sarkosyl-insoluble pellets were resuspended in 700 µl g$^{-1}$ extraction buffer (per gram of tissue) and centrifuged at 5,000$g$ for 5 min at 4 °C. The supernatant was diluted threefold in 50 mM Tris-HCl, pH 7.4, containing 0.15 M NaCl, 10% w/v sucrose and 0.2% w/v sarkosyl, and spun at 166,000$g$ for 30 min at 4 °C. The pellets were resuspended in 300 µl g$^{-1}$ EM buffer (20 mM Tris-HCl, pH 7.4, 100 mM NaCl).

The sample (4 µl) was applied to Quantifoil R1.2/1.3 (300 mesh) grids after a 60 s plasma cleaning step (Tergeo, Pie Scientific). Grids were blotted and plunge-frozen in liquid ethane using a Vitrobot Mark IV (FEI) with the chamber maintained at close to 100% humidity and 6 °C. The cryo-EM dataset was collected using EPU v.3.0 (Thermo Fisher) at the Astbury Biostructural Laboratory (University of Leeds) using a Titan Krios electron microscope (Thermo Fisher) operated at 300 kV with a Falcon4i detector in counting mode. A nominal magnification of ×96,000 was set yielding a pixel size of 0.83 Å. A total of 10,860 videos were collected with a nominal defocus range of −1.5 to −2.7 µm and a total dose of roughly 44 e$^-$/Å$^2$ over an exposure of 4 s, corresponded to a dose rate of roughly 7.6 e$^-$/pixel s$^{-1}$.

The raw EER videos were initially compressed and converted to TIFF using RELION v.4.0[59], regrouped to give 38 frames with a dose per frame of 1.2 e$^-$/Å$^2$. The TIFF stacks were aligned and summed using motion correction (MotionCorr2 v.1.2.1)[60] in RELION (Extended Data Fig. 1c) and contrast transfer function (CTF) parameters were estimated for each micrograph using CTFFIND v.1.14[61]. Tau fibrils from roughly 100 micrographs were picked manually and used to train a picking model in crYOLO v.1.9.6[62] for automated picking with an inter-box spacing of 3× layers (roughly 14 Å). Next, 321,041 segments were extracted 2× binned with roughly 560 Å$^2$ box dimensions. Two rounds of two-dimensional (2D) classification were performed to remove picking artefacts, with all classes corresponding to fibrils kept (Extended Data Fig. 1d) yielding 279,590 segments for further processing. An initial 3D template was generated from a PHF-like 2D class average and an estimated helical twist from measured crossover lengths (roughly 80 nm) using the relion_helix_inimodel2d command[63]. The first 3D classification was run using all of the 2× binned segments, with a sampling of 1.8° and strict high-resolution limit of 6 Å (Extended Data Fig. 1e), from which two classes presented tau PHF folds (72% of segments in total, the more ordered class containing 24% of segments was selected for further processing) and one class presented a tau SF fold (14% of segments). Each subset was extracted unbinned (336 pixel$^2$, 276 Å$^2$) and further classified with 0.9° sampling, without the high-resolution limit and with local searches of the helical twist (Extended Data Fig. 1f). The helical rise was set to 2.4 Å for the PHF subset and 4.8 Å for the SF subsets, on the basis of known structural data. The SF subset map improved to show a backbone fold that was identical to published tau SF structures[24], but did not contain enough segments to refine to a high-resolution structure. The PHF subset resolved to give a 3.0 Å (gold-standard, Fourier shell correlation (FSC) of 0.143) refined map (Extended Data Fig. 1g–i) after CTF refinement and Bayesian polishing, with a sharpening value of −57 Å$^2$ applied during postprocessing. The final refined helical rise and twist values were 2.405 Å and 179.44°, respectively (Supplementary Table 8).

## Model building of postmortem donor tau PHF cryo-EM structure

A published tau PHF fibril structure (Protein Data Bank (PDB) 5o3l)[24] was docked into the refined cryo-EM map and one chain was adjusted to fit into the density using real-space refine in Coot v.0.8.9.2[64]. The chain was duplicated and docked into the density to create five layers of dimeric tau polypeptide chains. The model was real-space refined using Phenix v.1.17.1[65] with noncrystallographic symmetry restraints applied to limit inter-chain divergence. The quality of the final model was assessed

using MolProbity v.4.5.2[66]. The final model was near identical to the template model, barring slight side chain adjustments and discrepancies in pixel size and/or magnification, with a root mean-squared deviation of 0.67 Å between a chain of each model for 73 Cα positions (Supplementary Table 8). The model was therefore not deposited to the PDB, as several identical models of tau PHF are already present[24].

## MX04-labelling and high-pressure freezing freeze−thawed postmortem acute brain slices

Flash-frozen postmortem brain samples were thawed at room temperature for 5 min, then placed in ice-cold carboxygenated NMDG buffer (93 mM NMDG, 2.5 mM potassium chloride, 1.2 mM sodium hydrogen carbonate, 20 mM HEPES, 25 mM glucose, 5 mM sodium ascorbate, 2 mM thiourea, 3 mM sodium pyruvate, 10 mM magnesium sulfate heptahydrate, 0.5 mM calcium chloride dihydrate, pH 7.4, 300−315 mOsmol)[56]. Then 100−200 µm slices of postmortem AD and control donor brain were sliced along the horizontal or coronal plane (speed 0.26 mm s$^{-1}$) using a vibratome (catalogue no. VT1200S, Leica) in ice-cold carboxygenated NMDG buffer (roughly 30 min). Next, postmortem AD and non-demented control acute brain slices were incubated in carboxygenated NMDG buffer, to which 15 µM MX04 was diluted for 1 h at room temperature before slices were washed three-times in carboxygenated NMDG for 5 min each. Grey matter biopsies (2 mm diameter) were incubated in cryoprotectant (5% w/v sucrose and 20% w/v dextran 40,000 in NMDG buffer) for 30 min at room temperature. Then, 100-µm-deep wells of the specimen carrier type A (Leica, catalogue no. 16770152) were filled with cryoprotectant, and the tissue biopsies were carefully placed inside to avoid tissue damage. They were then covered with the flat side of the lipid-coated specimen carrier type B (Leica, catalogue no. 16770153) and HPF (roughly 2,000 bar at −188 °C) using a Leica EM ICE.

### Cryo-FM

HPF samples were imaged using a cryogenic-fluorescence microscope (Leica EM Thunder) with a HC PL APO ×50/0.9 NA cryo-objective, Orca Flash 4.0 V2 sCMOS camera (Hamamatsu Photonics) and a Solar Light Engine (Lumencor) at −180 °C. A DAPI filter set (excitation and bandwidth 365 and 50, dichroic 400; emission and bandwidth, 460 and 50) was used to detect MX04-labelled amyloid. A rhodamine filter set (excitation and bandwidth 546 and 10, dichroic 560; emission and bandwidth, 525 and 50) was used as a control imaging channel. The images were acquired with a frame size of 2,048 × 2,048 pixels. Tile scans of HPF carriers were acquired with 17% laser intensity for 0.1 s. z stacks of ultrathin cryo-sections were acquired with 30% intensity and an exposure time of 0.2 s. Images were processed using Fiji ImageJ.

### Cryo-ultramicrotomy

HPF sample carriers were transferred to a cryo-ultramicrotome (Leica EM FC7, −160 °C) equipped with trimming (Trim 20, T399) and CEMOVIS (Diatome, cryo immuno, catalogue no. MT12859) diamond knives. A trapezoid stub of tissue measuring 100 × 100 × 60 µm was trimmed, which contained the target amyloid. Cryo-sections (70 nm thick) were then cut at −160 °C with a diamond knife (Diatome, cryo immuno, catalogue no. MT12859) and adhered onto a glow discharged (Cressington glow discharger, 60 s, 1 × 10$^{-4}$ mbar, 15 mA) 3.5/1, 300 mesh Cu grid (Quantifoil Micro Tools) using a Crion electrostatic gun and gold eyelash micromanipulators.

### Cryo-CLEM of cryo-sections

The location of amyloid plaques in tissue cryo-sections was assessed by cryogenic-fluorescence microscopy on the basis of MX04 fluorescence (excitation 370 nm, emission 460−500 nm). Grid squares that showed a signal for MX04 were selected for cryoET. The alignment between cryo-FM images and electron micrographs were carried out using a MATLAB script[67,68], in which the centres of ten holes in the carbon foil

surrounding the region of interest were used as fiducial markers to align the cryo-FM and cryo-EM images.

## Cryo-CLEM and cryo-FIB-SEM of lift-out lamellae

Carriers containing HPF samples for cryo-FIB-SEM lift-out were transferred to a cryo-ultramicrotome (Leica EM FC7, −160 °C) equipped with a Trim 45 T1865 diamond knife. The surface of the carrier and HPF tissue within were trimmed to remove surface ice contamination and the top layer of vibratome-damaged tissue[69]. To achieve this, three 300-μm-wide steps were trimmed back on four sides of the carrier: the outermost step was trimmed roughly 30 μm deep, the next was trimmed roughly 20 μm back and the innermost step was trimmed roughly 10 μm back. The resulting protruding square of tissue was then trimmed roughly 2–5 μm back to achieve a smoother less contaminated surface for cryo-FIB-SEM and lift-out.

The cryo-CLEM workflow was performed on a ZEISS Crossbeam 550 FIB-Scanning Electron Microscope equipped with the Quorum cryo-system, ZEISS Cryo-accessories tool kit and Omniprobe 350 cryo-micromanipulator (Oxford Instruments) and operated at 30 keV. For cryo-FM a Zeiss LSM (laser scanning microscope) 900 based on an upright Axio Imager stand equipped with AiryScan and Linkam cryo-stage was used.

At the Quorum prep desk, the HPF sample carrier (Leica Microsystems) was mounted on the corresponding Zeiss universal sample holder (USH, Zeiss cryo-accessories tool kit). First, the Zeiss USH was placed on the Zeiss adaptor for the Linkam cryo-stage used for performing cryo-FM. The assembly was transferred into the Linkam cryo-stage in liquid nitrogen using the Zeiss transfer box. A LM ZEN Connect project was acquired using the plugin ZEN Connect of the ZEN Blue v.3.6 software (Zeiss Microscopy). One overview image of the sample and the holder was acquired with a ×5, 0.2 NA C-Epiplan Apochromat air objective with an Axiocam 503. MX04, reflection (for correlation with the EM images) and a control channel were imaged with 385, 511 and 567 nm LEDs, respectively, in combination with a quad-band filter (Excitation BP 385 ± 15 nm, BP 469 ± 19 nm, BP 555 ± 15 nm, BP 631 ± 16.5 nm, Emission QBP 425 ± 15 nm, 514 ± 15 nm, 592 ± 12.5 nm, and 709 ± 50 nm). The image was acquired with a frame size of 2.79 × 2.10 mm and 1,936 × 1,460 pixels; 5, 3 and 60% LED power and 50, 6 and 300 ms exposure time for MX04, reflection and control, respectively.

For sample quality assessment, a large 3D volume (1.27 × 1.27 × 0.108 mm, 2,824 × 2,824 pixels, 49 z slices) was scanned with a 10×, 0.4 NA C-Epiplan Apochromat air objective in confocal mode. MX04 (0.04% laser power, 405 nm laser, detection window 410–546 nm, pixel dwell time 0.74 μs) and the reflection (0.01% laser power, 640 nm laser, detection window 630–700 nm, pixel dwell time 0.74 μs) were imaged.

Regions of interest were scanned with a 100×, 0.75 NA LD EC Epiplan-Neofluar air objective (125.15 × 125.15 × 19.8 μm and 1,140 × 1,140 pixels, 37 z slices, pixel dwell time 1.8 μs) with the following settings: MX04 (0.2% laser power, 405 nm laser, detection window 410–544 nm) and reflection (0.05% laser power, 640 nm, detection window 639–700 nm). A linear deconvolution was run on all confocal images (10× and 100×) using the Zeiss LSM plus plugin.

Airyscan z stacks were acquired from the regions of interest using the Airyscan 2 detector and the ×100 objective detailed above. MX04 was detected using the 405 nm laser line (1.5% laser power, 405 nm laser, pixel dwell time 2.18 μs, detection window 400–650 nm, image size 62.31 × 62.31 × 13.65 μm and 946 × 946 pixels, 36 z slices). Airyscan images were processed using the Airyscan joint deconvolution (jDCV) plugin of the Zeiss ZEN blue software. Cryo-CLEM alignment was performed with maximum projections of confocal and Airyscan image z stacks prepared in Zeiss ZEN blue. After finalizing light microscopy, the USH connected to the Linkam adaptor was transferred into the Quorum prep box. The USH was detached from the Linkam adaptor and mounted on the ZEISS cryo lift-out sample holder with the USH in flat orientation and a mounted upright, standing half-moon Omniprobe grid clipped

into an AutoGrid (Thermo Fisher). Using the Quorum cryo-shuttle the sample holder was transferred into the Quorum prep chamber attached to the Crossbeam 550 FIB-SEM. The temperatures of the cryo-stage and anticontaminator in the main and prep chamber were set to −160 °C and −180 °C, respectively. The sample was sputter-coated with platinum for 45 s (5 mA current). After the sputter coating, the sample was transferred on the Quorum cryo-stage in the main chamber.

Cryo-FM-targeted cryo-FIB-SEM lift-out was carried out driving the Zeiss Crossbeam FIB-SEM stage within a ZEN Connect imaging project, in which cryo-FM, SEM and FIB images were correlated using surface features (cryoplaning markings and ice contamination) as fiducial markers[70] (Extended Data Fig. 11). At normal view (0° stage tilt), 5 mm working distance and 2.3 kV acceleration voltage an overview of the HPF carrier and a zoomed-in SEM image were acquired and loaded into a new SEM session of the existing ZEN Connect project (imaging parameter SE2, Everhart-Thornley) detector, 98 pA SEM current, 4,096 × 3,072 pixels, 800 ns dwell time, line average with 23 iterations and 800 and 420 nm pixel sizes, respectively. The SEM session was correlated with the already acquired cryo-FM session by using the reflection mode channel images for alignment.

As image navigation was desired in FIB mode, the stage was tilted to 54° allowing normal FIB view. The sample was brought into the coincidence point of SEM and FIB with a 5 mm working distance and an overview FIB image (imaging current 50 pA, SE2 detector, 2,048 × 1,536, 1,6 μs dwell time, pixel average, 300 nm pixel size) was taken for FIB session alignment with the former SEM/cryo-FM session. The coincidence point was fine adjusted to the region targeted for lift-out and the milling box for coarse cross-sectioning was positioned on the basis of the alignment between cryo-FM and FIB images. A roughly 80-μm-wide, 35-μm-high and 30-μm-deep trapezoidal cross-section was milled from the front side using a 30 nA FIB probe. As ice contamination, especially on top of region of interest, was observed, the sample holder was transferred back into the Quorum prep box for cleaning. Under cryogenic conditions the sample surface was cleansed by using a brush and by wiping using a swab. After another sputter coating in the Quorum prep chamber (see above for parameters), a cold deposition of platinum precursor was applied in the Crossbeam main chamber. For cold deposition, the distance between sample and gas injection capillary was about 3 mm and the gas reservoir valve was opened for 45 s. The gas reservoir temperature was about 28 °C (unheated gas reservoir state). Using the saved stage position, the cross-section at the region of interest was targeted, the coincidence point alignment was checked. An FIB image was taken and aligned with the former FIB session using the already milled cross-section as reference.

Next, a 30 nA FIB probe was used to mill a second corresponding cross-section from the back side and a roughly 60-μm-wide left side cut that left a roughly 80-μm-wide, 10-μm-thick, 30-μm-deep tissue chunk attached on to its right side. At the front side, the cross-section was further polished using a 15 nA FIB probe. The stage was tilted to 10° tilt for milling a roughly 80-μm-wide L-shaped undercut, leaving a small connection on the left side (7 nA FIB probe). As a lift-out tool, a roughly 5-μm-thick copper block was attached to the Omniprobe manipulator tip. The stage was at 10° tilt to allow access of the micromanipulator while bypassing the AutoGrid ring. Before lift-out, the roughly 5 μm copper block was attached to the right side of the tissue chunk using redeposition milling of copper material (three 2.5 × 5 μm milling windows with 700 pA FIB probe and 140 mC cm$^{-2}$ dose). Next, the tissue chunk was cut free from the left side to achieve lift-out. At 10° stage tilt the roughly 80-μm-wide chunk was attached by redeposition milling to a half-moon EM grid (Omniprobe) clipped into an AutoGrid and cut in half, leaving the distal roughly 40 μm chunk attached to the EM grid. The remaining proximal roughly 40 μm chunk attached to the Omniprobe was attached to a second location on the EM grid. The stage was tilted to 56° before two 8–10-μm-wide, roughly 15-μm-deep lamella windows were milled in each chunk half. At 56° stage tilt, the

angle between tissue chunk and/or grid plane and FIB beam is 2° allowing to bypass the outer AutoGrid ring for lamellae thinning. The lamellae were sequentially thinned from both sides to roughly 2 µm, then roughly 1 µm, then roughly 500 nm and finally 130 to 200 nm using 700, 300, 100 and 50 pA FIB probes, respectively. Each lamella window was framed with unmilled tissue at the left, right and bottom sides.

### Cryo-electron tomography

Cryo-electron tomography was performed using a Thermo Fisher 300 keV Titan Krios G2, X-FEG equipped with a Falcon4i detector and Selectris energy filter in the Astbury Biostructure Laboratory at the University of Leeds. A dose symmetric tilt scheme[71] was implemented using Tomo5.15 (Thermo Fisher) to collect tilt series from −60° to +60° in 2° increments with a 100 µm objective aperture and 5–6.5 µm defocus. Each tilt increment received roughly 2.3 s of exposure (fractionated into eight frames) at roughly 2 e⁻/Å² per tilt for a total dose of roughly 120 e⁻/Å² per tilt series with a pixel size of 2.38 Å.

In AD samples, locations were chosen for the collection of tilt series in three different ways: (1) the presence of MX04 fluorescence, (Extended Data Fig. 3a), (2) the appearance of dense filamentous structures that resembled amyloid filaments in medium magnification micrographs (Extended Data Fig. 3b) and (3) areas surrounding MX04 fluorescence or other tissue areas in which membrane compartment features could be seen in medium magnification micrographs (Supplementary Tables 1, 2 and 4). Control non-AD samples lacked both MX04 signal (Extended Data Fig. 2a,b) and filamentous structures in medium magnification micrographs. Consequently, tilt series locations were picked in areas with visible membrane compartments in medium magnification micrographs (Supplementary Table 3).

### PMI and freeze–thaw step on mammalian brain tissue and amyloid architecture

We previously reported the architecture of Aβ-plaques in the *App*[NL-G-F] mouse model by cryoET[30]. In contrast to postmortem tissues with a 6 h PMI and freeze–thaw step, *App*[NL-G-F] tissues were cardiac perfused in NMDG buffer[56], were not freeze thawed and had a much shorter PMI[30] (hereon referred to as *App*[NL-G-F]-HPF samples). To control for the effect of 6 h PMI and a freeze–thaw step on the molecular architecture of tissues, we prepared *App*[NL-G-F] mouse (c57b/l6 background) tissues for cryoET under similar conditions as postmortem AD donor tissues, with a 6 h PMI and freeze–thaw step (hereon referred to as *App*[NL-G-F]-PMI-FT-HPF samples). Animals were treated in accordance with the UK Animal Scientific Procedures Act (1986) and National Institutes of Health guidelines. Oversight and approval was provided by the University of Leeds Animal Welfare and Ethics Review Board and licensed by the UK Government Home Office.

To prepare *App*[NL-G-F]-PMI-FT-HPF, a 10-month-old male *App*[NL-G-F] mouse received intraperitoneal injection of 5 mg kg⁻¹ MX04 24 h before culling[30]. The carcass was left at room temperature for 6 h to mimic the postmortem delay before forebrain was collected and flash-frozen in liquid nitrogen. Next, forebrain was taken through the workflow for postmortem human tissue (Fig. 1c) including a roughly 5 min thaw step preceding acute slice preparation in ice-cold carboxygenated NMDG buffer, 1 h incubation in carboxygenated NMDG buffer with 15 µM MX04, 3 × 5 min washes in carboxygenated NMDG buffer, and a 30 minute incubation in cryoprotectant (5% w/v sucrose and 20% w/v dextran 40,000 in NMDG buffer) at room temperature before high-pressure freezing. A MX04-labelled β-amyloid plaque was located within HPF tissue by cryo-FM, from which roughly 70-nm-thick cryo-sections were collected and attached to EM grids. Tissue sections were imaged by cryo-FM to locate MX04-labelled β-amyloid plaques before tomographic tilt series were collected from 14 and 46 locations with and without MX04 cryo-CLEM, respectively. Data collection parameters and constituents of these tomograms are detailed in Supplementary Table 7.

All MX04-labelled β-amyloid plaque tomograms (14 out of 60) from *App*[NL-G-F]-PMI-FT-HPF cryo-sections contained β-amyloid fibrils arranged in parallel bundles or a mesh (Extended Data Fig. 5). This fibril architecture was indistinguishable from that previously reported for *App*[NL-G-F]-HPF plaques that were HPF directly without a 6 h PMI and freeze–thaw step[30] (Extended Data Fig. 5).

We next compared non-amyloid constituents of amyloid plaques and subcellular compartments of the tissue that surrounded amyloid plaques in *App*[NL-G-F]-PMI-FT-HPF versus previously published *App*[NL-G-F]-HPF[30]. (1) In contrast to *App*[NL-G-F]-HPF tomograms[30], microtubules were absent in *App*[NL-G-F]-PMI-FT-HPF tomograms. This was expected because microtubules rapidly depolymerize at cold temperatures or when a source of nucleotide triphosphates is compromised[72,73]. Microtubules were also absent in postmortem AD and non-demented donor tomograms. (2) In contrast to *App*[NL-G-F]-HPF, *App*[NL-G-F]-PMI-FT-HPF tomograms contained a subset of mitochondria with swollen cristae and a depleted mitochondrial matrix (Extended Data Fig. 6c,d and Supplementary Table 7). This architecture is indicative of necrotic and/or apoptotic respiratory collapse[74,75]. Similarly damaged mitochondria were also observed in postmortem AD and non-demented donor tomograms (Extended Data Fig. 6a,b and Supplementary Table 7). (3) In contrast to *App*[NL-G-F]-HPF, 2 out of 60 *App*[NL-G-F]-PMI-FT-HPF tomograms contained a membrane fragment, suggesting the PMI and/or freeze–thaw step may produce membrane fragments in brain tissue that contains amyloid filaments. Burst plasma membrane were also observed in 10 out of 88 postmortem AD tomograms but were absent in 64 non-demented control postmortem tomograms. (4) In contrast to *App*[NL-G-F]-HPF, 10 out of 60 *App*[NL-G-F]-PMI-FT-PMF tomograms contained a burst plasma membrane, suggesting the PMI and freeze–thaw step can disrupt membrane integrity in these mice (Extended Data Fig. 5). Burst plasma membranes were observed in 11 out of 80 postmortem AD tomograms but absent in 64 non-demented control postmortem tomograms.

### Tomographic reconstructions and subtomogram averaging

Subtomogram alignment and averaging of Aβ fibrils and tau filaments was initially performed on a per-tomogram basis to assess the relationship between amyloid structure and its subcellular context. Because of the dense architecture of in-tissue amyloid, only tomograms with axial (oriented in the *z* axis of the reconstructed tomogram) fibrils and filaments provided sufficient contrast to accurately pick subvolumes for subtomogram averaging.

Dose-fractionated video frames were imported into Warp (v1.1.0b)[76] for frame alignment and initial estimation of the CTF. Tilt series stacks were generated in Warp and imported into etomo IMOD (v.4.12.35)[77,78] for fine alignment using patch tracking. Lower quality aligned frames were excluded from stack generation before import. Fine aligned tilt series were then imported back into Warp and 3D-CTF corrected tomograms were reconstructed at a pixel size of 9.52 Å (bin4). See Supplementary Tables 5 and 6 for per-tomogram subtomogram averaging details of tau filaments and Aβ fibrils, respectively.

For tau filaments, subtomogram averaging was performed on nine cryoET volumes (seven from cryo-sections and two from lift-out lamellae). Between 55 and 278 tau filaments (24 × 100 × 24 box size) were manually picked from 4× binned (9.52 Å voxel size) tomographic reconstructions in 3dmod (IMOD v.4.12.35). Coordinates of each filament were represented as a two-point contour with respective 'head' and 'tail' model points positioned at the poles of each filament. The slicer function of 3dmod was used to rotate and translate each tau filament, ensuring different models were approximately centred along the filament axis. Using these model files as input, a script invoking the particle estimation for electron tomography (PEET) 'stalkInit' command was run in default mode to generate new, single-point model files (containing coordinates of the head, centroid and tail), initial motive lists and rotation axes files, to supply for alignment and averaging in

PEET (v.1.17.0a)[79,80]. To minimize alignment to the missing wedge and to verify the accuracy of model point coordinates, initial averages were generated by restricting rotational and translation searching from the centroid of the de novo reference filament axis. In subsequent PEET alignment iterations, rotational and translational searching was performed, using the updated centroid coordinates and orientations as input. Where visible improvements in map resolution were observed, createAlignedModel was run to generate new models, motive lists and rotation axes containing the updated locations and orientations of particles from the alignment. Cylindrical masks with blurred edges were applied to the stalkInit reference volume during subvolume alignment. StalkInit subtomogram averages were generated without applying symmetry or helical reconstruction methods. The helical twist of tau filaments in each tomogram was measured from PEET stalkInit averages.

To improve alignment further, centroid models were added every 1 voxel along the stalkInit filament axis (99,664 new model coordinates from the nine cryoET volumes) using the 'addModPts' command. AddModPts models were used as new inputs for subvolume alignment and averaging in PEET. CreateAlignedModel was also run after addModPts alignments, outputting new models and initial motive lists of the updated locations and orientations of particles.

After initial alignment iterations in PEET, coordinates and orientations were imported into RELION v.4.0[59] (toRelionCoords) for further refinement with Warp bin4 tomographic reconstructions[59]. This was achieved using the 'imodinfo' command with the '-l' option to output the model information to a text file. Helical prior information was generated using a custom python script (available from GitHub at https://github.com/jjenkins01/model2helicalpriors.git) to modify the STAR file to contain rlnHelicalTubeID, rlnHelicalTrackLength, rlnAngleTiltPrior (set to 0), rlnAnglePsiPrior (set to 0) and rlnAnglePsiFlipRatio (set to 0.5). This modified STAR file (generated from 'toRelionCoords' and custom python scripts) was used in Warp to generate subtomograms (24 × 24 × 24 box size, 9.52 Å) that were then extracted for helical 3D auto-refinement and classification in RELION (using helical twist estimated from PEET stalkInit averages and without C2 symmetry). Refined subtomograms were then re-extracted in Warp at pixel sizes of 4.76 Å (48 × 48 × 48 box size) and followed by helical 3D auto-refinement and classification. Final rounds of helical 3D auto-refinement and 3D classification were performed on subtomograms re-extracted in Warp at pixel sizes of 2.38 Å (96 × 96 × 96). Final averages were sharpened using RELION postprocessing and global map resolutions were estimated at a FSC of 0.143 between two independently refined half-maps. Refined particle coordinates and orientation from RELION 3D auto-refinement that reached subnanometre resolution (Fig. 3h) were imported into M (v.1.0.9)[81] for multi-particle refinements.

For Aβ fibrils, subtomogram averaging was performed using two cryo-section volumes collected from MX04-labelled β-amyloid plaques that contained axially oriented fibrils. The same manual picking and subtomogram averaging procedures were followed (using stalkInit) as described for tau filaments. Initially, 100 fibrils were picked from one tomogram, with fibrils represented as two-point 'head' and 'tail' models, positioned at opposite ends of each fibril. This model was used as input to the PEET 'stalkInit' command to generate new model files (head, centroid, tail), initial motive lists and rotation axes files, for alignment and averaging in PEET. The subtomogram average of 100 fibrils generated a featureless, smooth tube, with reducing detail observed with increasing fibril numbers used for averaging, a possible indication of particle heterogeneity (Extended Data Fig. 10). To assess fibril heterogeneity, the widths of individual fibrils were manually measured in IMOD. On the basis of average fibril width, three subpopulations of fibrils were picked (3–5 nm protofilament-like rods, 4–9 nm fibrils and 6–12 nm wide fibrils). Subtomogram averaging of 20 protofilament-like rods, 42 fibrils and 42 wide fibrils from two tomograms was performed with PEET as before (Supplementary Table 6 and Extended Data Fig. 10).

To compare the structural similarity of in situ subtomogram averages with ex vivo purified tau conformers, available AD tau filament atomic models (PDB 5o3l and 5o3t) were fitted into half-maps with EM placement v.1.2.2[37,38] in ChimeraX v.1.5[82,83] giving LLG scores of fit (Figs. 3h and 4h,j, Extended Data Fig. 9 and Supplementary Table 5). When running EM placement, the map resolution was set to the value determined using the 0.143 half-map FSC threshold. Theory and experience with single-particle cryo-EM data suggest that LLG scores of 60 or greater indicate a non-random fit, with higher values being observed for more accurate models and higher resolution maps[38]. The LLG score is related to how much of the information content of the map is explained by the model. Because the total information content of the portion of the map covered by the model depends both on its local (including anisotropic) resolution and its volume, an LLG score above the threshold can be reached by any combination of map quality with model size and quality. The confidence threshold itself (LLG 60) is not expected to vary with map resolution or issues arising from preferential orientations[38]. For reference, fitting a PHF atomic model[24] into a 3 Å resolution single-particle cryo-EM (Extended Data Fig. 1g) half-maps gave LLG 11,889 and correlation coefficient 0.66. Atomic model (PHF or SF)[24] versus subtomogram averaged map Fourier shell correlation was also calculated with Phenix v.1.17.0a[65], avoiding artefacts at the edges of the averaged map by restricting the comparison to a central sphere with a radius of 80 Å (Extended Data Fig. 13).

Initial subtomogram averaging (without helical symmetry in PEET) and after-helical averaging in RELION were assessed by visualizing Euler angles in ChimeraX using the ArtiaX v.0.4.7[84] indicating good overall agreement. Filaments were assigned a polarity orientation in which those with more than or equal to an 80% Z Euler angle were oriented in the same direction (Extended Data Fig. 12c–f).

## Preparation of cryoET figures

All tomographic slice figures were one voxel thick prepared from 4× binned (9.52 Å voxel size) tomographic reconstructions generated with MotionCorr2 v.1.2.1[60] aligned frames that were reconstructed with AreTomo v.1.3.0[85] and deconvolved with Isonet v.0.2[86]. Segmenting membranes within tomograms was performed in Dynamo v.1.1.532[87] using the manual surface modelling tool. Dynamo tables containing coordinate information of the tomogram membrane models were converted into CMM files and visualized in ChimeraX. The neural network-based tomogram segmentation pipeline in EMAN2 v.2.99[88] was used to segment tau filaments and Aβ fibrils. In Fig. 3c–h, regions of the raw tomographic volumes containing β-amyloid fibrils and tau filaments were trimmed using the IMOD rubber band tool and trimvol command. The raw tomographic density was then visualized in ChimeraX using the segger tool to colour each Aβ fibril/tau filament by connectivity.

## Annotation of constituents in tomographic volumes

The constituents of tomographic volumes of tissue cryo-sections from β-amyloid plaques, tau pathology, non-demented control and cryo-FIB-SEM lift-out of tau threads were detailed in Supplementary Tables 1, 2, 3 and 4, respectively. Constituents were initially identified by two curators independently. Next, two curators consolidated and verified each annotation. The boundary of intracellular and extracellular regions of tomographic volumes were determined by the presence of myelinated axons or by lipid bilayers enclosing intracellular organelles within each tomographic volume and the corresponding electron micrograph used for cryo-CLEM. The following constituents were identified: (1) amyloid fibrils or filaments were assigned on the basis of MX04 cryo-CLEM labelling and rod shape: fibrils and filaments were cross-checked for their absence in tomographic volumes reconstructed from non-demented control brain donors; (2) subcellular compartments, defined by plasma membrane containing membrane-bound organelles or a higher tomographic density than

the extracellular space (excluding amyloid fibrils), that is, consistent with the higher concentration of proteins in the cell cytoplasm compared to the extracellular space; (3) mitochondria, defined by the double membrane including outer membrane and inner mitochondrial cristae; (4) putative F1FO-ATPases were identifiable within the inner mitochondrial membrane (for example, see Fig. 2c); (5) damaged mitochondria, double membrane-bound compartments with swollen cristae and a less densely packed mitochondrial matrix than other mitochondria to the extent that parts of the mitochondria seem empty (Extended Data Fig. 6); (6) myelinated axon, defined as five or more layers of 6–8 nm membrane lipid bilayer enclosing a subcellular compartment[89]; (7) open membrane sheets, defined by roughly 5-nm-thick membrane lipid bilayer sheets in interstitial space that did not form closed compartments; (8) burst compartment, membrane compartments that are open in the $x$–$y$ plane of the tomogram; (9) extracellular vesicles were defined as membranes that were closed and situated in extracellular spaces; (10) C-shaped vesicles were defined as cup-shaped membrane within the lumen of a vesicle; (11) multilamellar bodies were defined as 40–250 nm vesicle or subcellular compartment wrapped in three or more concentric rings of 4.5–6 nm membrane lipid bilayer; (12) extracellular droplets, 30–250 nm amorphous and smooth spheroidal droplets[35,36] that were situated in extracellular locations; (13) extracellular cuboidal particles, 27–200-nm-diameter particles that were situated in extracellular locations (these particles contained striated layers of higher tomographic density: 2.5–2.8 nm apart, Extended Data Fig. 7); (14) vesicles were defined as closed spheroidal membranes; (15) F-actin, defined as roughly 7-nm-diameter filaments composed of a helical arrangement of globular subunits[90] (F-actin was only observed intracellularly in a minor subset of both AD and non-demented control donor tomographic volumes); (16) ribosomes: spherical 25–30-nm-diameter dense particles; (17) microtubules, 25-nm-diameter tubes with 13 tubulin subunits and (18) knife damage, tissue cryo-sections contained regions in which the sample seems compressed, sometimes leaving a crevasse in the tissue that were readily identified as holes within the tissue[39].

## Statistics and reproducibility

All cryo-CLEM (Figs. 1g and 4b and Extended Data Figs. 3a,b and 11), tomographic slice images (Figs. 1h, 2a,b, 3 and 4e,f and Extended Data Figs. 3c–e, 4, 7 and 8a) and electron micrographs (Fig. 4d and Extended Data Figs. 1c and 3a,b) were representative of datasets from a single AD (42 β-amyloid plaque cryo-section, 25 tau pathology cryo-section, 13 tau pathology lift-out lamella tomograms, see Supplementary Tables 1, 2 and 4, respectively) and non-demented control (64 cryo-section tomograms, Supplementary Table 3) postmortem brain donor. All immunohistochemical imaging (Fig. 1a and Extended Data Fig. 1a), immunoblot (Fig. 1b and Extended Data Fig. 1b), immunofluorescence (Fig. 1c and Extended Data Fig. 2a) and cryo-FM imaging (Figs. 1e,f and 4c and Extended Data Fig. 2b–d) experiments were representative of 3–4 technical replicates of these donor tissues. tomographic slices in Extended Data Figs. 6 and 7 were representative of cryoET datasets of HPF (23 cryo-section tomograms, $n = 2$ biological replicates)[30] and freeze thawed (60 cryo-section tomograms, $n = 1$ biological replicates, see Supplementary Table 7) $App^{NL-G-F}$ knock-in mice.

## Reporting summary

Further information on research design is available in the Nature Portfolio Reporting Summary linked to this article.

## Data availability

Subtomogram average maps have been deposited in the Electron Microscopy Data Bank (EMDB) under accession codes EMD-50148 (CS1, Extended Data Fig. 9c), EMD-50152 (CS2, Fig. 3g), EMD-50153 (CS3, Extended Data Fig. 9b), EMD-50155 (CS4, Extended Data Fig. 9d),

EMD-50156 (CS5, Extended Data Fig. 9a), EMD-50157 (CS6, Extended Data Fig. 9e), EMD-50159 (CS7, Extended Data Fig. 9f), EMD-50160 (LOL1 PHF, Fig. 4h), EMD-50161 (LOL1 SF, Fig 4h) and EMD-50162 (LOL2 SF, Fig. 4j). The cryo-EM map and dose-fractionated video frames of sarkosyl-extracted tau PHF from a postmortem AD donor has been deposited to the EMDB with the accession code EMD-18990 and the Electron Microscopy Public Image Archive (EMPIAR) with the accession code EMPIAR-12045, respectively. Dose-fractionated video frames and tomograms associated with CS1-CS7 and LOL1-LOL2 subtomogram average maps have been deposited in the EMPIAR under accession code EMPIAR-12082. Tomographic datasets of postmortem AD brain tissue, non-demented control and $App^{NL-G-F}$ knock-in mice have been deposited under accession codes EMPIAR-12091, EMPIAR-12088 and EMPIAR-12092, respectively.

## Code availability

Source code for generating helical priors used in relion helical reconstruction is available from GitHub at https://github.com/jjenkins01/model2helicalpriors.git.

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

**Acknowledgements** M.A.G.G. was funded by a Medical Research Council (MRC) DiMeN DTP studentship (grant MR/N013840/1). R.A.W.F. holds a UKRI Future Leader Fellowship (grant MR/V022644/1) and a University of Leeds Academic Fellowship. S.E.R. holds a Royal Society Professorial Fellowship (grant RSRP\R1\211057). M.W. was funded by Wellcome and MRC (grants WT 204963 and MR/T011149/1). The Astbury Biostructure Laboratory Titan Krios microscopes were funded by the University of Leeds and Wellcome (grants 108466/Z/15/Z and 221524/Z/20/Z). The Leica EM ICE, UC7 ultra/cryo-ultramicrotome and cryo-CLEM systems were funded by Wellcome (grant 208395/Z/17/Z). R.J.R. was funded by Wellcome (grant 209407/Z/17/Z). We are grateful to the donors of postmortem tissue, clinical records and the Netherlands Brain Bank. We thank R. Hughes and S. Boxall (University of Leeds Bioimaging Facility) for support with fluorescence imaging experiments. We thank A. Turner for helpful discussion of amyloid structural biology. We thank I. Rigo and M. Reay for technical support. We thank J. Whitwell, C. McLean and G. Baxter for computational support. We thank R. Thompson, E. Hesketh, L. Aspinall, J. White and O. Degtjarik for help maintaining and setting up the Astbury Biostructure Laboratory (ABSL) cryo-EM facility, including high-pressure freezing, cryo-CLEM and Titan Krios microscopes.

**Author contributions** J.J.M.H. and T.H.J.M. arranged donor tissues and performed neurohistopathological characterization of postmortem tissue. N.F. and M.A.G.G. performed immunofluorescence and biochemical characterization of postmortem tissue, sarkosyl-insoluble extracts and tissue sample preparation. T.J.O., M.A.G.G., N.F. and M.G. carried out cryo-FM. T.J.O., M.A.G.G., N.F. and R.A.W.F. carried out cryo-ultramicrotomy. A.S. performed cryo-FIB-SEM lift-out. M.A.G.G. and Y.H. collected cryo-EM and cryoET data. M.A.G.G., T.J.O., J.J. and R.A.W.F. reconstructed cryoET data. M.A.G.G. and N.F. performed cryo-CLEM analysis. M.A.G.G., N.F. and R.A.W.F. annotated constituents of tomographic data. J.J., M.A.G.G. and N.F. carried out segmentations. J.J. and M.A.G.G. performed subtomogram averaging. M.W. performed single-particle cryo-EM data collection and helical reconstruction. N.A.R. and S.E.R. supervised M.W. R.R. wrote EM placement software. R.A.W.F. and S.E.R. supervised M.A.G.G. R.A.W.F. devised and supervised the entire project. All authors contributed to writing the paper.

**Competing interests** The authors declare no competing interests.

**Additional information**
**Correspondence and requests for materials** should be addressed to René A. W. Frank.

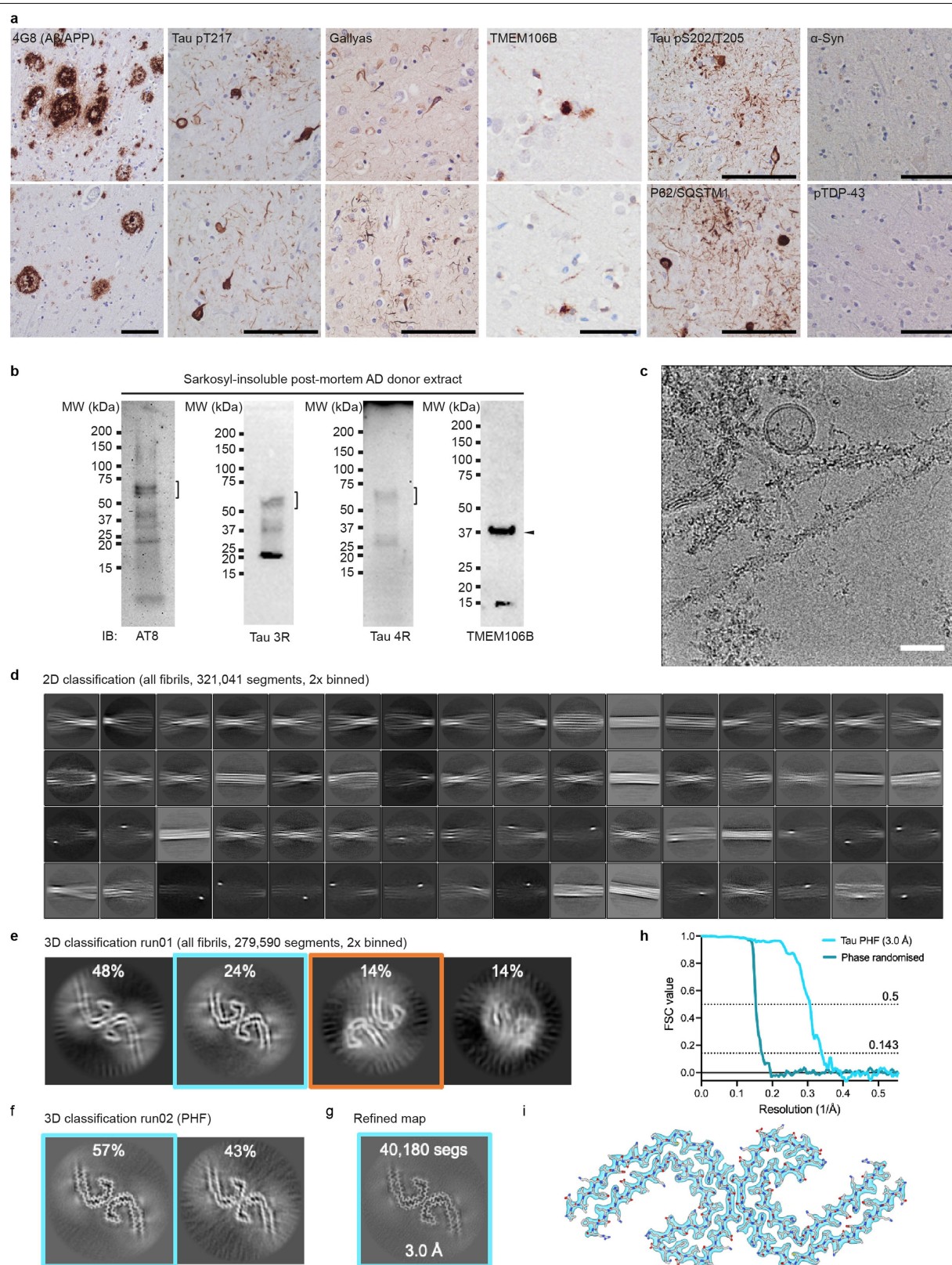

**Extended Data Fig. 1** | See next page for caption.

**Extended Data Fig. 1 | Neuropathological, biochemical, and cryoEM characterisation of post-mortem AD donor. a**, Formalin-fixed paraffin embedded tissue from the mid-temporal gyrus of the AD case assessed for amyloid pathology. Representative pairs of images showing from left to right, β-amyloid deposits detected with 4G8, tau pathology assessed with Tau phospho-S217 antibody, Gallyas silver staining to detect amyloid deposits, TMEM106B (C-terminal domain) inclusions, tau phosphoS202/T205 (AT8), and P62/SQSTM1 (a marker of aggregated protein) and the absence of α-synuclein and phospho-TDP-34 inclusions. Scale bar, 50 μm. **b**, Immunoblot detection of sarkosyl-insoluble fraction of AD post-mortem brain, from left to right, of phospho-Tau (AT8), three-repeat (3 R) tau, four-repeat (4 R) tau, and TMEM106B (C-terminal domain). Brackets, full-length tau bands. Arrowhead, proteolysed TMEM106B. **c**, Example micrograph from the cryoEM dataset of sarkosyl-insoluble amyloid purified from post-mortem AD donor. Scale bar, 50 nm. **d**, The most populated 2D class averages following classification of all fibril segments. Most of the resolvable classes resembled PHF and SF. **e-f**, Central slices of output maps from 3D classifications, annotated with segment distributions as a percentage within each run. Selected classes taken for further processing are highlighted in coloured boxes. Classification yielded 40,180 PHF segments (cyan) that produced a 3.0 Å resolution map. SF segments (orange) were less abundant and could not be resolved to high resolution. **g**, 2D projection of refined, sharpened tau PHF cryoEM map shown. **h**, Fourier shell correlation resolution estimation of final map resolution. **i**, Atomic model in map was indistinguishable from several deposited tau PHF cryoEM models, including the template selected for model building (PDB: 5o3l)[24]. See Supplementary Data Table 8.

**Extended Data Fig. 2 | Immunofluorescence profile and cryoFM targeting of MX04-labelled amyloid pathology of post-mortem AD donor tissue.**
**a**, Related to Fig. 1c. Confocal fluorescence microscopy of AD post-mortem donor brain and non-demented control brain. Images from left to right, amyloid (MX04), Aβ/APP (4G8), phospho-Tau (AT8), and merged. Cyan arrowhead, β-amyloid plaque. Open orange arrowhead, tau thread. Scale bar, 20 µm. **b**, CryoFM image of high-pressure frozen post-mortem brain biopsy from AD (left) and non-demented control (right) donor. Scale bar, 0.5 mm. Cyan, MX04 fluorescence. Red, autofluorescence detected with excitation and emission of 546 nm and 585 nm, respectively. Yellow rectangle, regions shown as close-ups below (lower left and right, respectively). Lower left and right panels, putative β-amyloid plaque and tau tangles, respectively. Closed cyan arrowhead,

putative β-amyloid plaque. Closed orange arrowhead, putative tau tangle. Open orange arrowhead, putative tau thread. Scale bar, 50 µm. **c**, CryoFM targeting of cryo-ultramicrotomy, related to Fig. 1f. Top to bottom, cryoFM image of planchette containing MX04-labelled high-pressure-frozen tissue, stereomicroscope image of the planchette during trimming with a cryo-ultramicrotome, alignment of cryoFM and stereomicroscope images, and close-up image of trimmed planchette with cryoFM image. White arrowhead, trapezoid stub of tissue targeted for the collection of cryo-sections. Orange arrowhead, MX04-labelled amyloid within tissue stub. **d**, Left and right, MX04 cryoFM image of tissue cryo-section containing β-amyloid plaque and tau tangles, respectively. White arrowhead, MX04-labelled amyloid pathology. Scale bar, 50 µm. Red and cyan, same as in **b**.

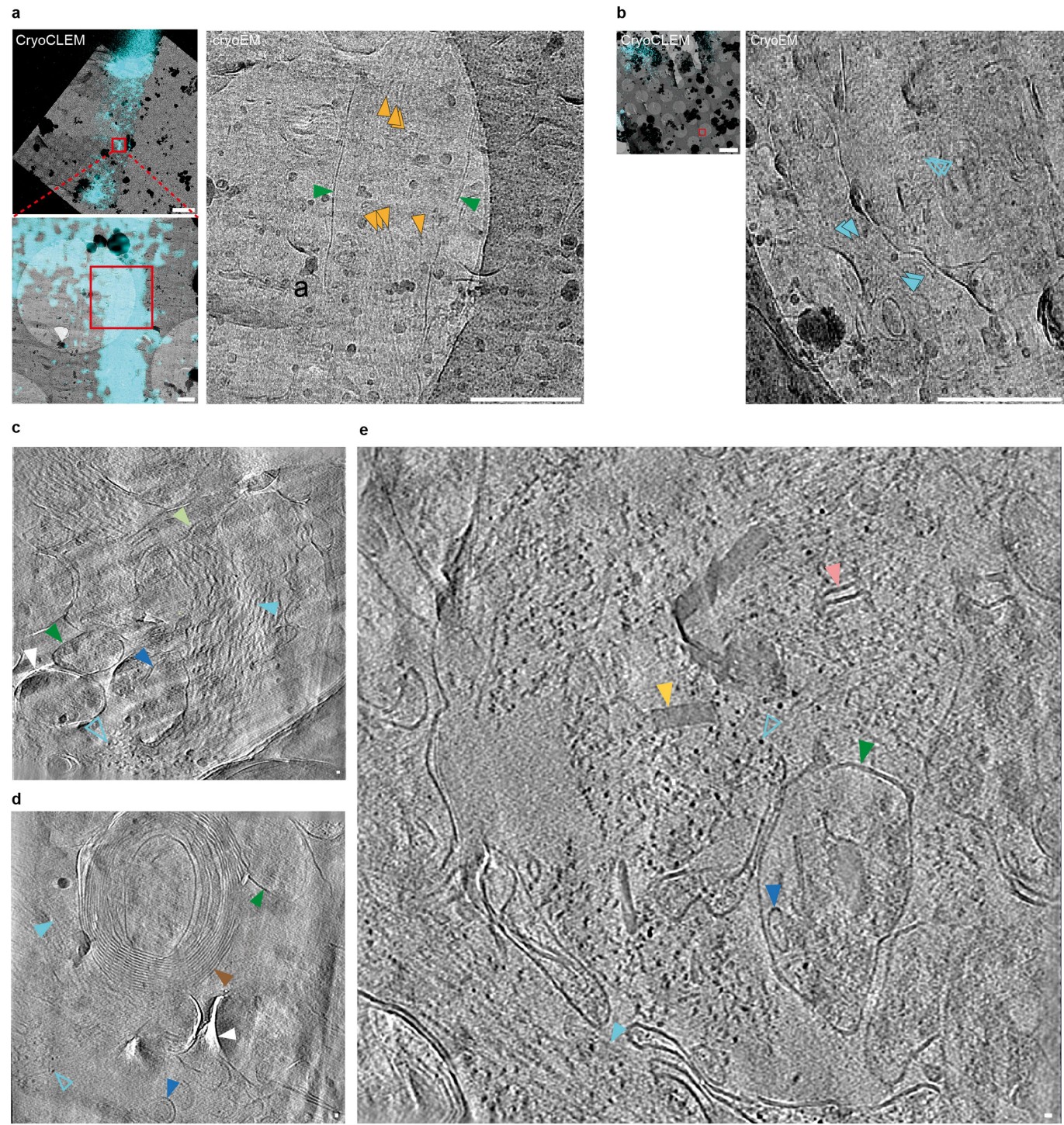

**Extended Data Fig. 3 | Cryo-CLEM and cryoET of MX04-labelled post-mortem AD brain. a**, CryoCLEM of MX04-labelled tau inclusion within AD post-mortem brain cryo-section. Top left, aligned cryoFM image (cyan, MX04) with cryoEM image. Red rectangle, area shown in close-up. Scale bar, 5 μm. Bottom left, close-up. Red rectangle, region shown, area shown in close-up. Right, close-up. Orange arrowheads, putative tau filaments. Green arrowhead, putative plasma membrane of neurite. Scale bar, 500 nm. See also cryoET data in Extended Data Fig. 8 and Supplementary Video 6). **b**, CryoCLEM of AD post-mortem brain cryo-section showing unlabelled amyloid deep in the tissue below the depth of MX04 penetration. Left, MX04 (cyan) cryoFM image aligned with cryoEM image showing MX04 only labels top ~15 μm of 100 μm thick tissue biopsy. Red rectangle, region in cryo-section corresponding to 27 μm deep within the

tissue biopsy. Scale bar, 5 μm. Right, close-up of left medium magnification cryoEM image showing region from which cryoET data were collected (see **e**). Cyan arrowhead, putative Aβ fibrils. Scale bar, 500 nm. **c-e**, Tomographic slices of β-amyloid plaque pathology in post-mortem AD brain. Filled and open cyan arrowheads, fibril in the *x-y* plane and axially (*z*-axis) of the tomogram, respectively. Brown arrowhead, myelinated axon. Dark green arrowhead, subcellular compartment. Blue arrowhead, intracellular membrane bound organelle. Yellow arrowhead, extracellular cuboidal particle. Pink arrowhead, extracellular vesicle. White arrowhead, knife damage. Scale bar, 10 nm. **c**, β-amyloid plaque pathology. Related to Fig. 1g, see also Supplementary Video 2. **d**, Same as **c** but with amyloid pathology adjacent to myelinated axon. **e**, Same as **c**, but related to **b** (also see Supplementary Video 3).

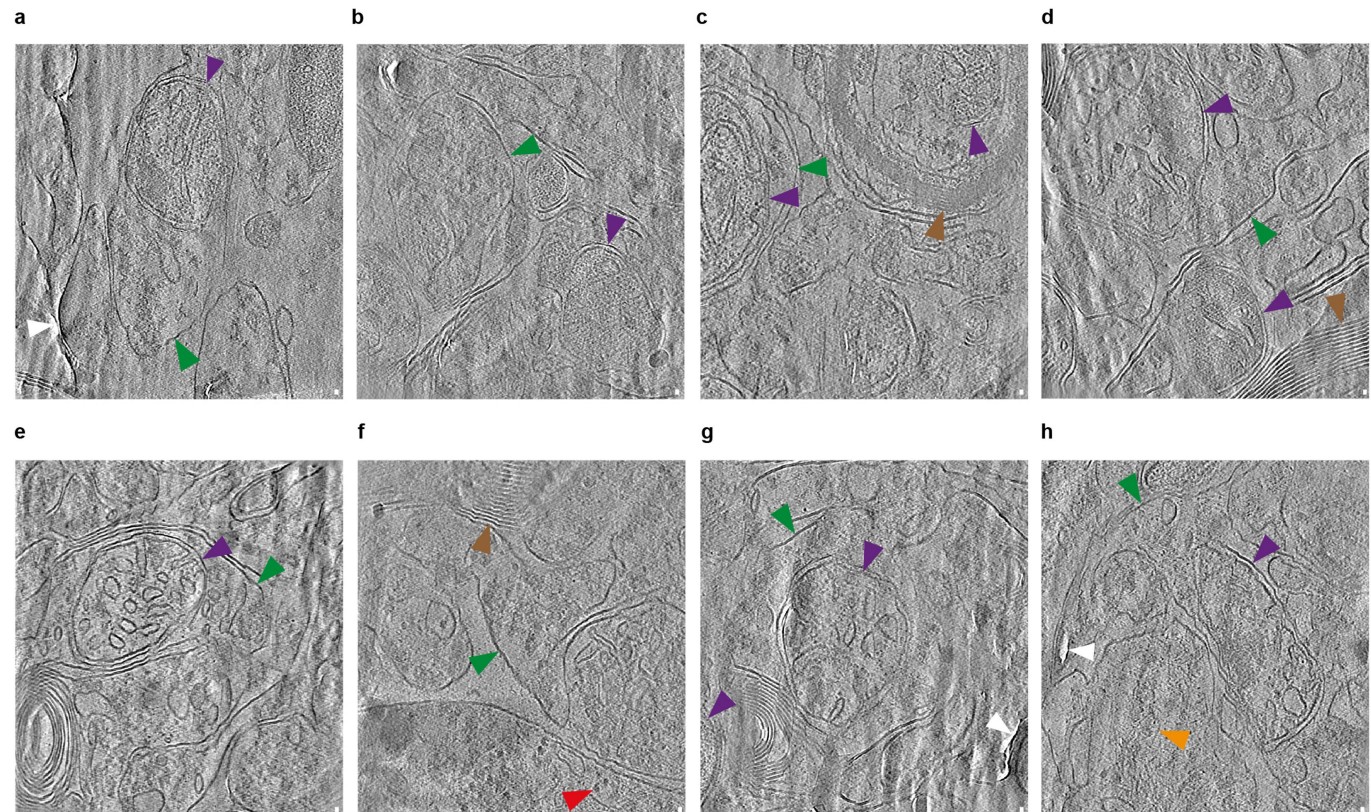

**Extended Data Fig. 4 | In-tissue cryoET of cryo-sections from non-demented control post-mortem brain donor.** Panels **a-h** show tomographic slices from 8 out of 64 tomograms in the dataset. No amyloid was observed in non-demented control post-mortem donor cryo-sections (see Supplementary Data Table 3). Green arrowhead, plasma membrane enclosing subcellular compartment.

Purple arrowhead, mitochondrion. Brown arrowhead, myelin. Red arrowhead, ribosomes. Orange arrowhead, actin. White arrowhead, knife damage. Criteria for identifying constituents are described in **Methods**. Scale bar, 10 nm. See Supplementary Videos 7–9.

$App^{NL-G-F}$ HPF

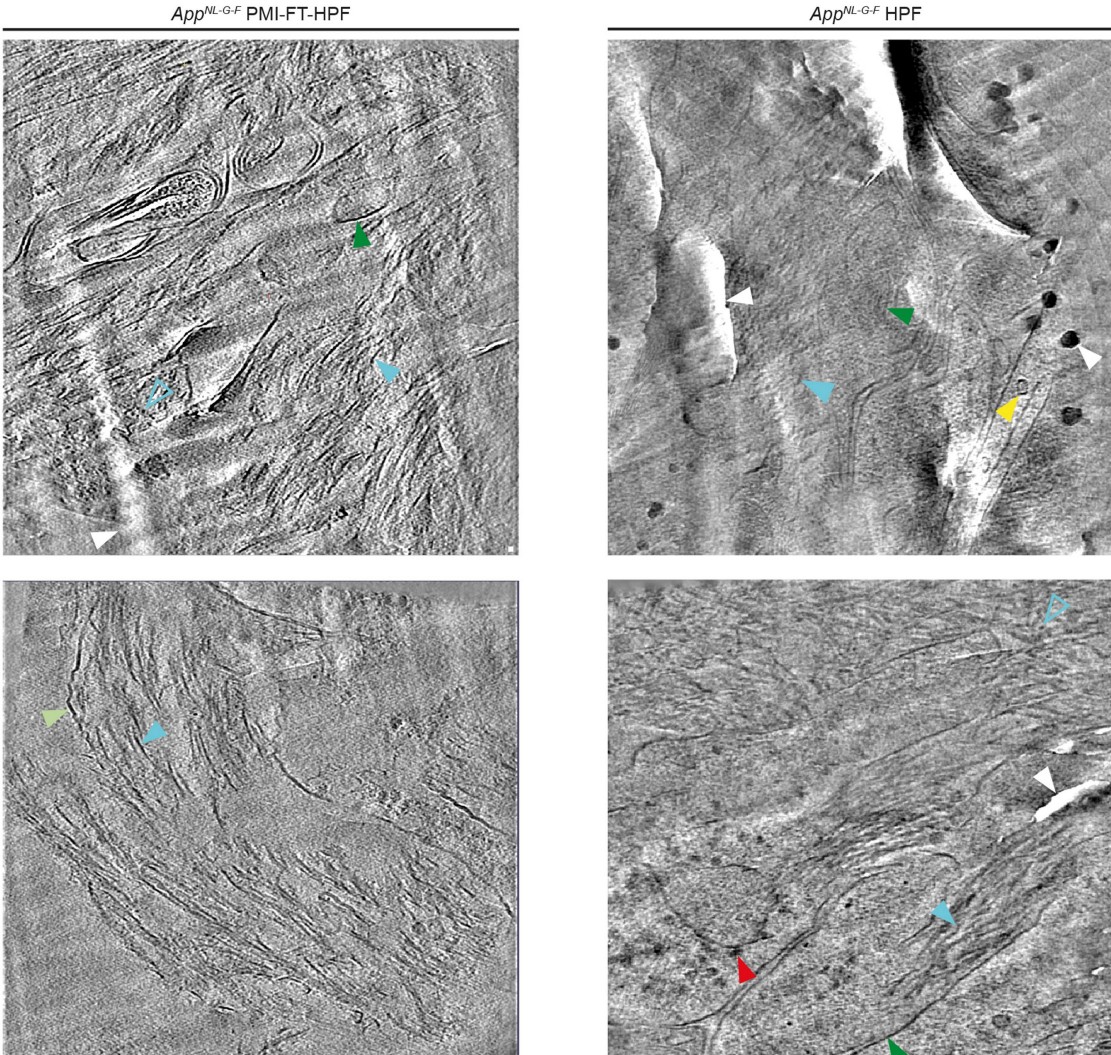

**Extended Data Fig. 5 | CryoET comparison of in-tissue MX04-labelled β-amyloid plaque in $App^{NL-G-F}$-PMI-FT-HPF and previously published $App^{NL-G-F}$-HPF.** *Left panels*, tomographic slices from an $App^{NL-G-F}$ mouse brain that underwent a 6 h post-mortem interval and freeze-thaw step ($App^{NL-G-F}$-PMI-FT-HPF) before vitrification by high pressure freezing and cryo-section preparation. *Right panels*, tomographic slices from $App^{NL-G-F}$ cortex that was immediately vitrified by high pressure freezing, with no post-mortem interval or freeze-thaw step ($App^{NL-G-F}$-HPF, from Leistner et al.)[30]. Filled and open cyan arrowheads, fibril in the *x-y* plane and axially (*z*-axis) of the tomogram, respectively. Dark green arrowhead, plasma membrane enclosing subcellular compartment. Light green arrowhead, burst plasma membrane. Yellow arrowhead, microtubule. Red arrowhead, ribosome. White arrowhead, knife-damage and surface ice contamination. Scale bar, 10 nm.

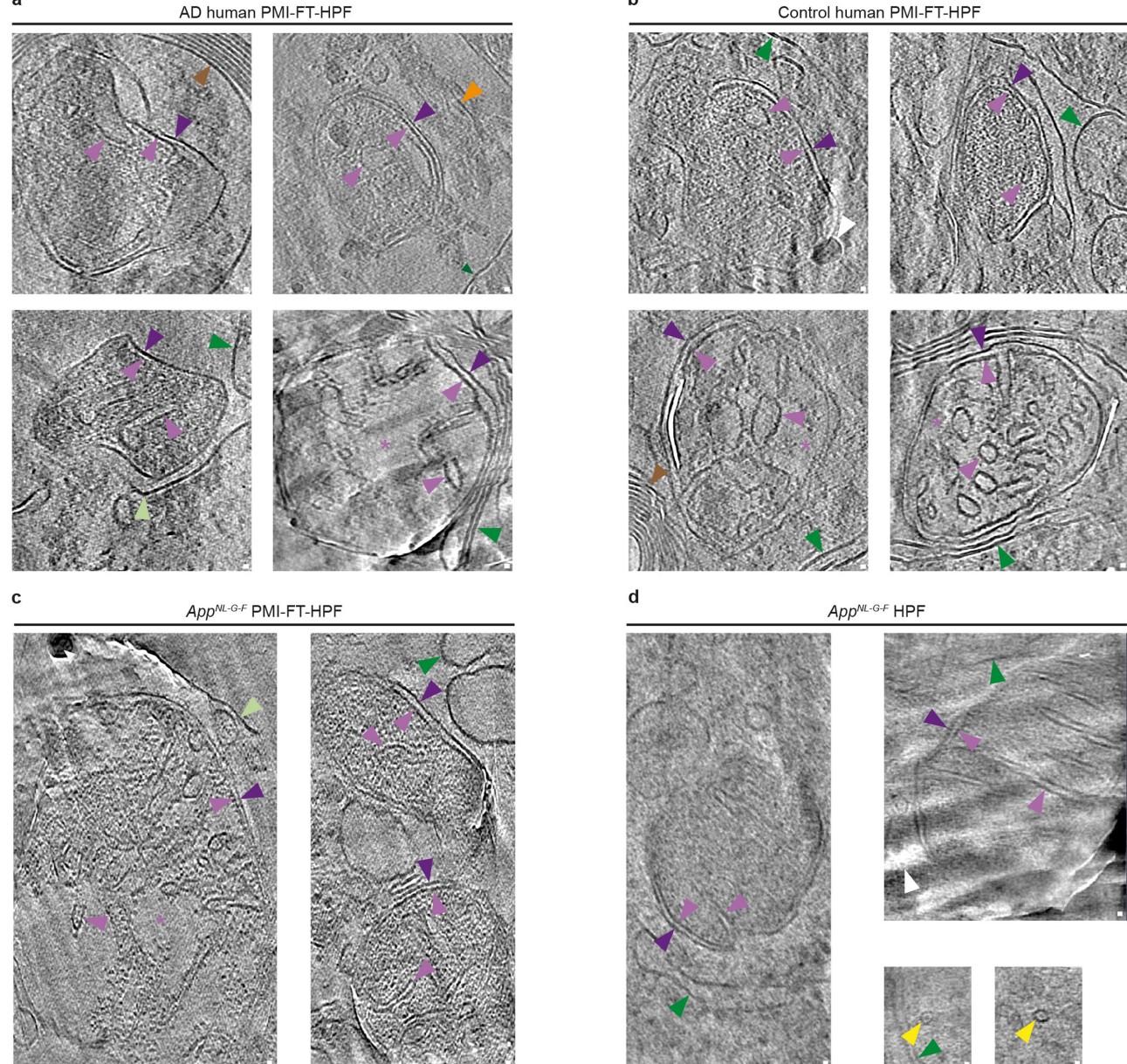

**Extended Data Fig. 6 | Tomographic slices showing damaged mitochondria in brain tissues that have undergone post-mortem interval and freeze-thaw step. a**, Post-mortem AD brain with PMI and freeze-thaw step that preceded high pressure freezing (PMI-FT-HPF). **b**, Post-mortem non-demented control brain (Control human PMI-FT-HPF). **c**, Mouse model of β-amyloidosis brain (*App^NL-G-F*) prepared with PMI-FT-HPF. **d**, Mouse model of β-amyloidosis (*App^NL-G-F*). Sample prepared without post-mortem interval and freeze-thaw step

(*App^NL-G-F*-HPF) (see Leistner et al. for details)[30]. Dark purple arrowhead, outer mitochondrial membrane. Light purple arrowhead, inner mitochondrial membrane. Light purple asterisk, diluted mitochondrial matrix. Dark green arrowhead, sub-cellular membrane compartment. Light green arrowhead, burst membrane. Orange arrowhead, actin filament. Yellow arrowhead, microtubule. Brown arrowhead, myelin sheath. White arrowhead, knife damage or surface ice contamination. Scale bar, 10 nm.

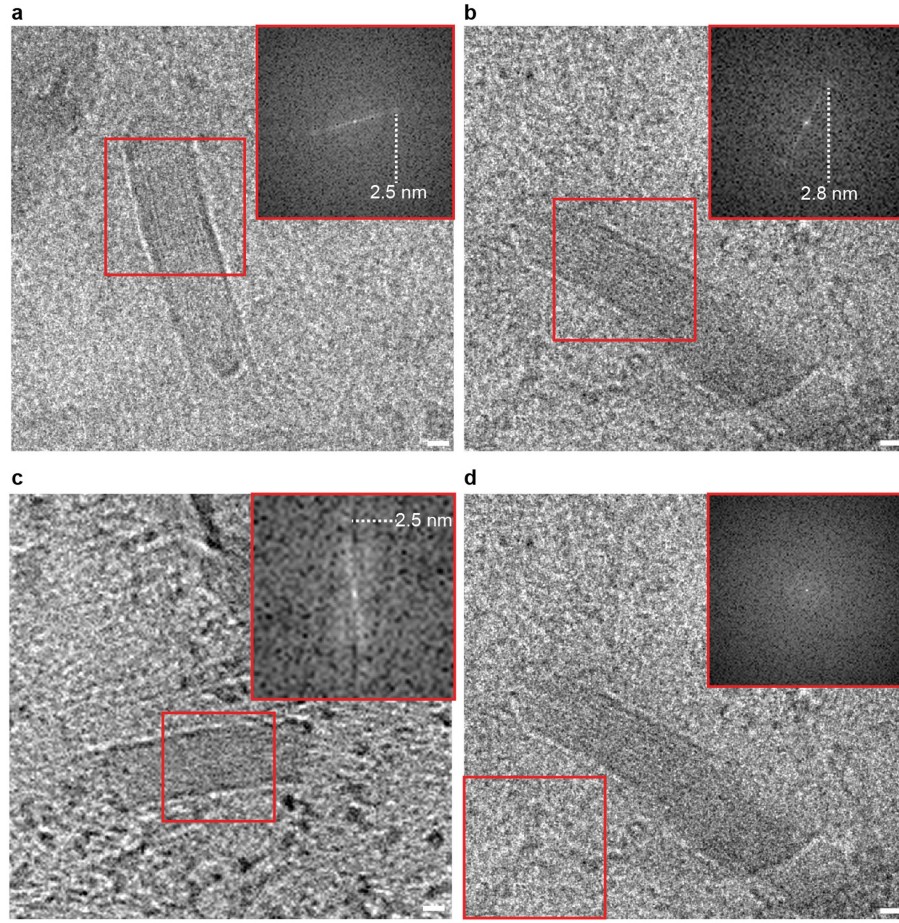

**Extended Data Fig. 7 | Extracellular cuboidal particles have ordered striations. a** and **b**, cryoEM images (single tomographic tilt, 2.4 Å pixel size) of an extracellular cuboidal particle in an Aβ plaque (see Extended Data Fig. 3e). Red square, subregion analysed by fast Fourier transform shown in insets with 2.5 nm or 2.8 nm peak, respectively. **c**, Tomographic slice (9.6 Å voxel size) showing extracellular cuboidal particle. **d**, Same as **b** but with no peak in control subregion. Scale bar, 10 nm.

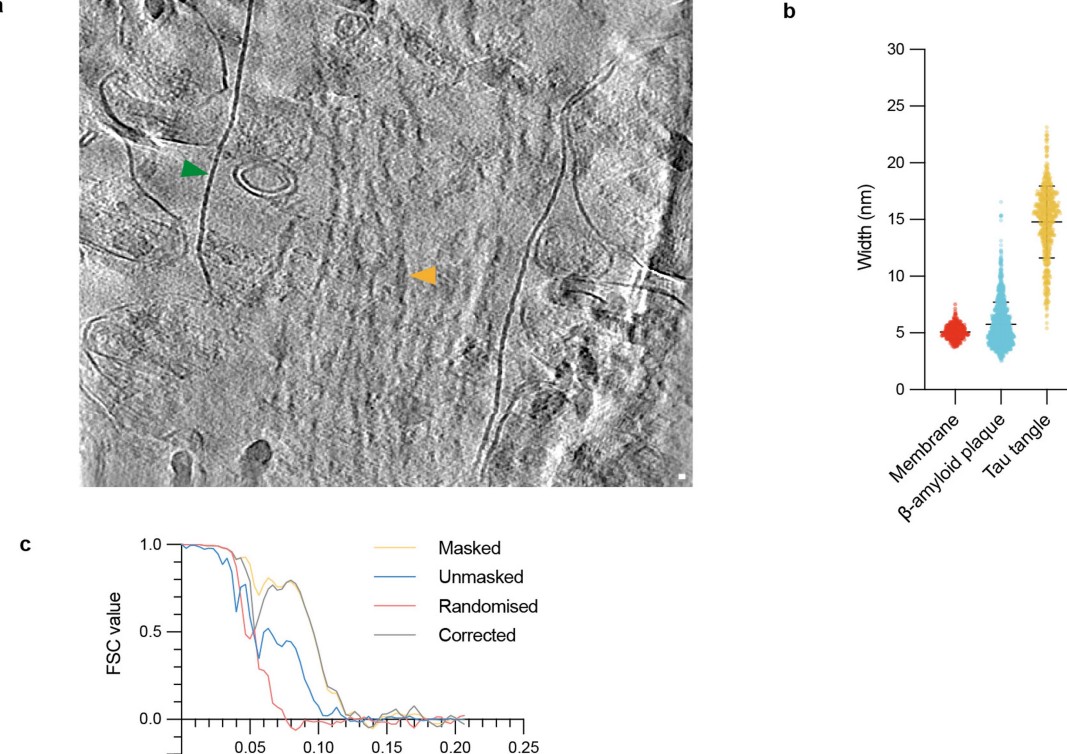

**a**

**b**

**c**

**Extended Data Fig. 8 | In-tissue cryoET of MX04-labelled tau inclusion cryo-section from AD post-mortem brain donor, amyloid width measurements, and FSC plot (related to Fig. 3h). a**, Tomographic slice through dystrophic neurite. Orange arrowhead, tau filament. Green arrowhead, plasma membrane of neurite. Scale bar, 10 nm. Related to Extended Data Fig. 3a and see Supplementary Video 6. **b**, Scatterplot showing the width distribution of lipid membrane (used as an internal control for width measurements, n = 296) and fibrils from in-tissue cryo-sections of β-amyloid plaques (n = 1360) and tau tangles (n = 561), respectively. Middle and top/bottom black bars indicate mean and one standard deviation. **c**, Resolution estimation of subtomogram averaging of tau subvolumes from a single tissue cryo-section by gold-standard Fourier shell correlation (FSC).

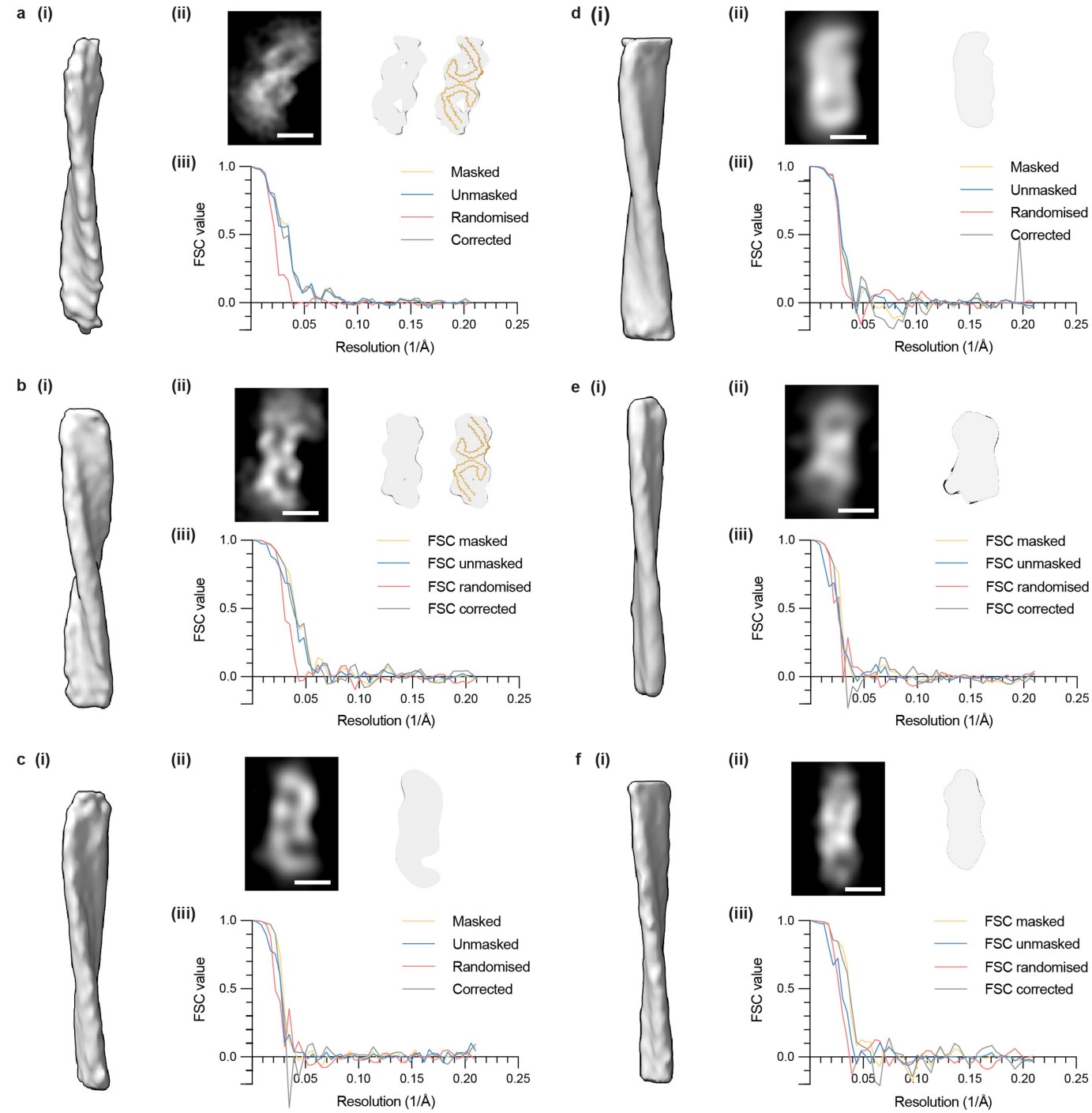

**Extended Data Fig. 9 | Lower-resolution subtomogram averages of tau filament clusters within tissue cryo-sections situated in six distinct locations. a-b**, Subtomogram averaged tau clusters composed of PHFs. **(i)** Surface rendering of subtomogram averaging of tau filaments (stalkInit, see Methods). **(ii)** Helical averaging of tau filament subvolumes from one AD cryo-section tomogram. Left, slice (2.38 Å thick) through averaged subvolume. Scale bar, 5 nm. Middle and right, subtomogram average shown without and with ex vivo purified tau PHF atomic model (yellow, PDB 5o3l)[24] fitted in map

with EM placement (LLG > 60)[37,38], respectively. Each protofilament in the PHF shown as yellow $C_\alpha$ trace. **(iii)** Graph showing resolution estimation of subtomogram averaging of tau subvolumes from a single tissue cryo-section by gold-standard Fourier shell correlation (FSC). See Supplementary Data Table 5 for EM placement[37,38] LLG and CC of PHF and SF[24]. **c-f**, Same as **a-b** but for subtomogram averaged tau clusters in which the tau filament ultrastructural polymorph was not resolved (EM placement LLG < 60)[37,38].

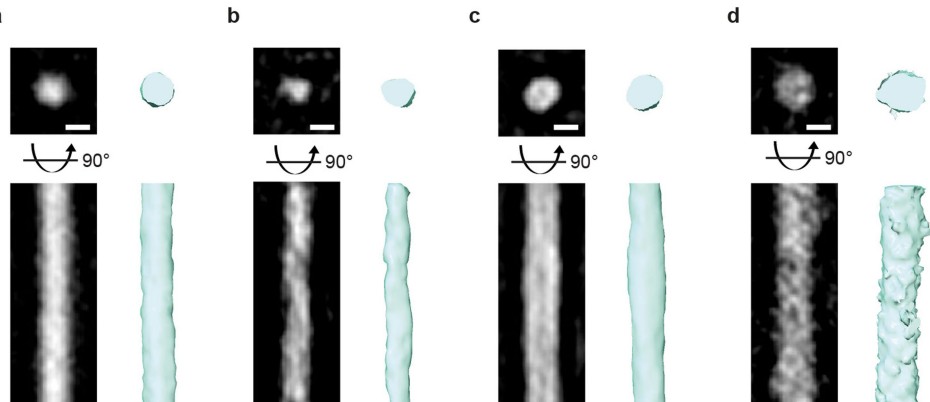

**Extended Data Fig. 10 | Subtomogram averaging of Aβ fibrils from MX04-labelled β-amyloid plaque cryo-sections.** Pairs of panels showing, top and side view of fibril. Left, slice (9.52 Å thick) through averaged subvolume. Right, surface rendering of subtomogram average (stalkInit, see Methods). Scale bar, 5 nm. **a**, Subtomogram averaging of 100 fibrils (of all widths) from one tomogram. **b**, Subtomogram averaging of 20 protofilament-like rods (3–5 nm diameter) from one tomogram. **c**, Subtomogram averaging of 42 fibrils (4–9 nm diameter) from one tomogram. **d**, Subtomogram averaging of 42 thick fibrils (6–12 nm diameter) from a different tomogram.

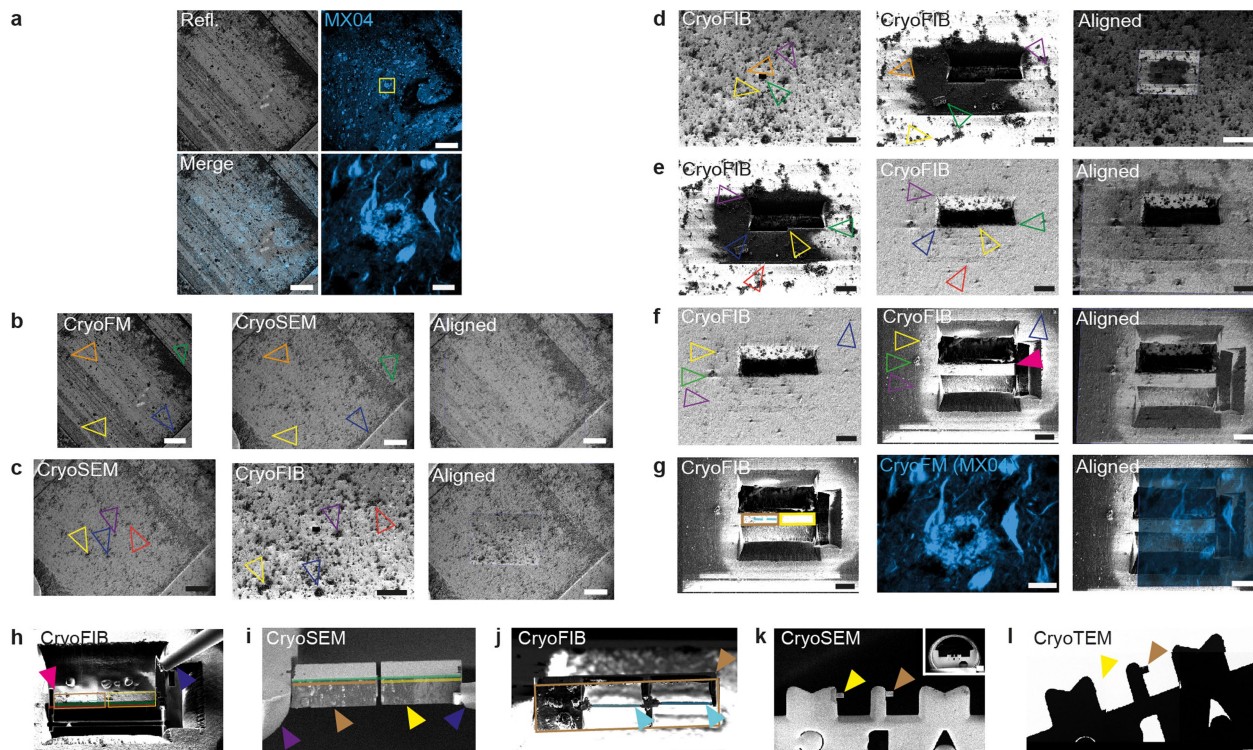

**Extended Data Fig. 11 | Correlated cryo-FM-FIB-SEM lift-out targeting of MX04-labelled tau pathology in high-pressure frozen post-mortem AD brain. a**, CryoFM of MX04-labelled amyloid pathology targeted for cryoFIB-SEM lift-out showing in top left, top right, and lower left, reflection, MXO4 fluorescence and merged overview, respectively. Scale bar, 200 μm. Yellow rectangle in top right, closeup in bottom right. Scale bar, 20 μm. **b-c**, Alignment of cryoFM, FIB and SEM images to target lift-out in ZEN Connect. Images on left were aligned with middle image using fiducials indicated by open arrowheads. **b**, Alignment between cryoSEM normal overview and cryoSEM normal high-magnification image of MX04-labelled lift-out target. Scale bar, 200 μm. **c**, Alignment between cryoSEM normal high-magnification and cryoFIB normal image. Scale bar, 200 μm. Left and right, scale bar, 200 μm. Middle, scale bar, 100 μm. **d**, Alignment between cryoFIB normal image and cryoFIB high-magnification after cryoFIB milling trench. Left and right, scale bar, 100 μm. Middle, scale bar, 20 μm. **e**, Alignment between cryoFIB images before and after surface cleaning, sputter coating and cold deposition of platinum precursor. Scale bar, 20 μm. **f**, Alignment of cryoFIB images before and after trenches were milled in front, behind and to the right side to prepare tissue chunk for targeted lift-out. Scale bar, 20 μm. **g**, Images showing the result of alignments (from **a** to **h**) to target MX04-labelled tau for cryoFIB-SEM lift-out. Left, cryoFIB image of tissue chunk prepared for lift-out. Middle, confocal cryoFM of MX04-labelled amyloid plaque and tau tangles. Brown and yellow rectangles, left and right regions of serial chunk lift-out, respectively. Cyan line, location of lamella. Scale bar, 20 μm. **h-l**, CryoFIB-SEM lift-out of

MX04-labelled post-mortem AD brain. Lines and rectangles tracing the tissue chunk during cryoFIB cuts and thinning were used for cryoCLEM. Red semi-transparent line, length of tissue chunk before lift-out. Green semi-transparent line, length of tissue chunk during lift-out. Brown rectangle, left tissue chunk. Yellow rectangle, right tissue chunk (lost during sample transfer from cryoFIB-SEM to cryoEM). Cyan line, region targeted for cryoFIB milling lamellae within tissue chunk. **h**, CryoFIB image of tissue chunk after final left side cut to detach tissue chunk. Magenta closed arrowhead, tissue chunk. Blue closed arrowhead, lift-out tool (copper block linking micromanipulator needle) attached to right side of tissue chunk. Scale bar, 10 μm. **i**, CryoSEM images showing attachment of the left tissue chunk to EM grid and cryoFIB cut between the left and right tissue chunks. Purple closed arrowhead, EM grid. Brown closed arrowhead, left tissue chunk. Yellow closed arrowhead, right tissue chunk. Blue closed arrowhead, lift-out tool. Scale bar, 10 μm. **j**, CryoFIB normal view image (56° stage tilt) of the tissue chunk after cryoFIB thinning to produce two 130–200 nm thick, ~8 μm wide, ~15 μm deep lamellae windows. Cyan closed arrowhead, lamella. Brown closed arrowhead, tissue chunk. Scale bar, 10 μm. **k**, CryoSEM showing serial attachment of the left and right tissue chunks. Yellow and brown closed arrowheads, left and right tissue chunks, respectively. Scale bar, 50 μm. Inset, overview image showing clipped half-moon Omniprobe EM grid. Scale bar, 0.5 mm. **l**, CryoEM overview showing left and right cryoFIB-milled tissue chunk attachment positions, left was lost and right remained during transfer cryoFIB to Krios TEM, respectively. Arrowheads, same as **k**. Scale bar, 50 μm.

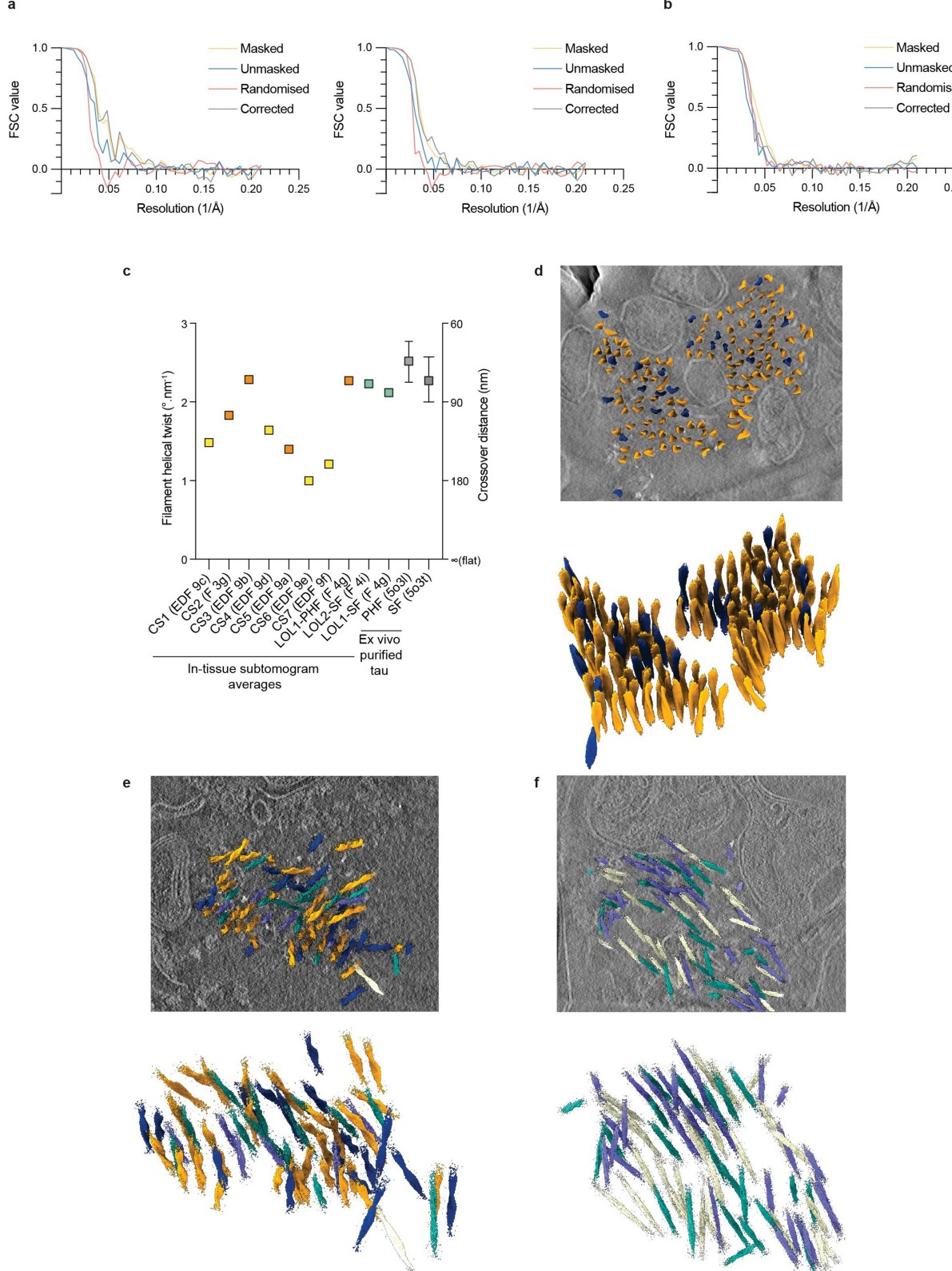

**Extended Data Fig. 12** | See next page for caption.

**Extended Data Fig. 12 | Related to Fig. 4h–j and spatially restricted variation in helical twist and polarity orientation of in-tissue tau filament clusters. a**, Left and right, resolution estimation of subtomogram averaging of 52 PHF and 19 SF in a single tissue cryoFIB-SEM lift-out lamella tomogram by gold-standard Fourier shell correlation (FSC), respectively. **b**, Resolution estimation of subtomogram averaging of 64 tau SF in a single tissue cryoFIB-SEM lift-out lamella tomogram by Fourier shell correlation (FSC). **c**, Filament helical twist (left y-axis) and cross-over distance (right y-axis) of in-tissue tau filament clusters in different locations of AD post-mortem brain from seven cryo-section (CS1-7) and two lift-out lamella tomograms, one of which (LOL1) was composed of two ultrastructural polymorphs, related to maps shown in Fig. 3g, Fig. 4g & i and Extended Data Fig. 9a–f. Orange and cyan data points, tau cluster subtomogram averages that fitted the atomic models of PHF or SF using EM placement (see Supplementary Data Table 5), respectively. Yellow data points, tau cluster subtomogram averages in which the ultrastructural polymorph was unresolved. For reference, the helical twist/crossover distances calculated for AD PHF and SF are shown in grey (data point and whiskers, cryoEM and range of twists observed by negative stain EM from Fitzpatrick et al.[24], respectively). See also Supplementary Data Table 5. **d**, Filament subtomogram average map (see Fig. 3h) mapped back into the raw tomographic volume with PHF filaments coloured by polarity orientation. Orange and indigo filaments with upwards and downwards polarity orientation, respectively. Top and bottom, volume viewed from top and rotated ~45°, with and without tomographic slice, respectively. Scale bar, 10 nm. **e**, Same as **b**, but related to Fig. 4h. Cyan and purple, SF filaments with upwards and downwards polarity orientation, respectively. **f**, Same as **b** and **c**, but related to Fig. 4j.

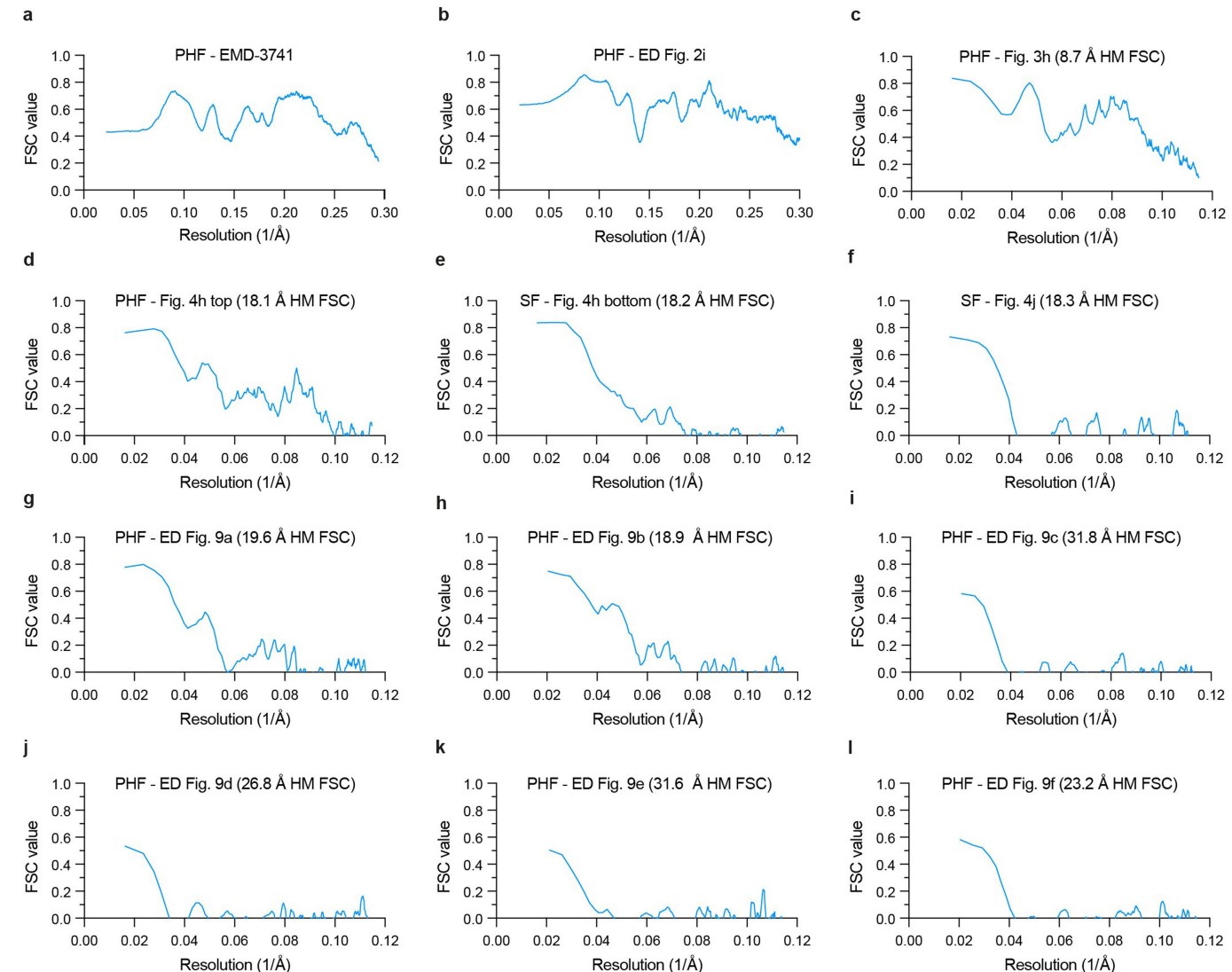

**Extended Data Fig. 13 | CryoEM atomic model versus subtomogram map Fourier shell correlation. a**, For reference, Fourier shell correlation of previously reported single-particle cryoEM structure of ex vivo sarkosyl-extracted PHF (PDB: 5o3l) and map (EMDB-3741)[24]. **b**, Fourier shell correlation of PHF (PDB 5o3l) and cryoEM map of ex vivo sarkosyl-extracted PHF from donor (Extended Data Fig. 1i). **c-l**, Fourier shell correlation of PHF (PDB 5o3l) or SF (PDB 5o3t) atomic model[24] versus subtomogram average maps corresponding to Fig. 3h PHF, Fig. 4h PHF, Fig. 4h SF, Fig. 4j SF and Extended Data Fig. 9a–f, respectively. Resolution estimates based on Fourier shell correlation of half-maps annotated in parenthesis. See also Supplementary Data Table 5.

# Reporting Summary

## Statistics

For all statistical analyses, confirm that the following items are present in the figure legend, table legend, main text, or Methods section.

| n/a | Confirmed | |
|---|---|---|
| ☐ | ☒ | The exact sample size (*n*) for each experimental group/condition, given as a discrete number and unit of measurement |
| ☐ | ☒ | A statement on whether measurements were taken from distinct samples or whether the same sample was measured repeatedly |
| ☒ | ☐ | The statistical test(s) used AND whether they are one- or two-sided *Only common tests should be described solely by name; describe more complex techniques in the Methods section.* |
| ☒ | ☐ | A description of all covariates tested |
| ☒ | ☐ | A description of any assumptions or corrections, such as tests of normality and adjustment for multiple comparisons |
| ☐ | ☒ | A full description of the statistical parameters including central tendency (e.g. means) or other basic estimates (e.g. regression coefficient) AND variation (e.g. standard deviation) or associated estimates of uncertainty (e.g. confidence intervals) |
| ☒ | ☐ | For null hypothesis testing, the test statistic (e.g. *F*, *t*, *r*) with confidence intervals, effect sizes, degrees of freedom and *P* value noted *Give P values as exact values whenever suitable.* |
| ☒ | ☐ | For Bayesian analysis, information on the choice of priors and Markov chain Monte Carlo settings |
| ☒ | ☐ | For hierarchical and complex designs, identification of the appropriate level for tests and full reporting of outcomes |
| ☒ | ☐ | Estimates of effect sizes (e.g. Cohen's *d*, Pearson's *r*), indicating how they were calculated |

*Our web collection on statistics for biologists contains articles on many of the points above.*

## Software and code

Policy information about availability of computer code

| Data collection | CryoET: Tomo v5.15 (Thermo Fisher) ; cryoEM: EPU v3 (Thermo Fisher) |
|---|---|
| Data analysis | CryoET and cryoCLEM data processing:<br>MotionCorr2 v1.2.1, IMOD v4.12.35, PEET v1.17.0a, Warp/M v1.10b, RELION v4.0, em_placement v1.2.2, Dynamo v1.1.532, AreTomo v1.3.0,EMAN v2.99, Isonet v0.2, ChimeraX v1.5, Phenix v1.2.1, Zeiss ZEN Blue v3.6, Matlab R2019a, ImageJ v2.0.0-rc-49/1.51d, ArtiaX v0.4.7.<br>CryoEM data processing:<br>MotionCorr2 v1.2.1, RELION v4.0, crYOLO v1.9.6, Coot v0.8.9.2, Phenix v1.17.1, CTFFIND v1.14, Molprobity v4.5.2. |

For manuscripts utilizing custom algorithms or software that are central to the research but not yet described in published literature, software must be made available to editors and reviewers. We strongly encourage code deposition in a community repository (e.g. GitHub). See the Nature Portfolio guidelines for submitting code & software for further information.

## Data

Policy information about availability of data

All manuscripts must include a data availability statement. This statement should provide the following information, where applicable:

- Accession codes, unique identifiers, or web links for publicly available datasets
- A description of any restrictions on data availability
- For clinical datasets or third party data, please ensure that the statement adheres to our policy

A data availability statement is included. Subtomogram average maps have been deposited in the Electron Microscopy Data Bank (EMDB) under accession codes EMD-50148 (CS1, Extended Data Fig. 9c), EMD-50152 (CS2, Fig. 3g), EMD-50153 (CS3, Extended Data Fig 9b), EMD-50155 (CS4, Extended Data Fig 9d), EMD-50156 (CS5, Extended Data Fig 9a), EMD-50157 (CS6, Extended Data Fig 9e), EMD-50159 (CS7, Extended Data Fig 9f), EMD-50160 (LOL1 PHF, Fig. 4h), EMD-50161 (LOL1 SF, Fig 4h), EMD-50162 (LOL2 SF, Fig. 4j). The cryoEM map of sarkosyl-extracted tau PHF from post-mortem AD donor has been deposited to the EMDB with the accession code EMD-18990.
Dose-fractionated movie frames and tomograms associated with CS1-CS7 and LOL1-LOL2 subtomogram average maps have been deposited in the Electron Microscopy Public Image Archive (EMPIAR) under accession code EMPIAR-12082. Tomographic datasets of post-mortem AD brain tissue, non-demented control and AppNL-G-F knockin mice  have been deposited under accession codesEMPIAR-12091, EMPIAR-12088 and EMPIAR-12092, respectively.

## Research involving human participants, their data, or biological material

Policy information about studies with human participants or human data. See also policy information about sex, gender (identity/presentation), and sexual orientation and race, ethnicity and racism.

| | |
|---|---|
| Reporting on sex and gender | Post-mortem Alzheimer's disease donor brain tissue:  Sex: Female. Age at death: 70 years.<br>Post-mortem non-demented control donor brain tissue: Sex: Male. Age at death: 90 years. |
| Reporting on race, ethnicity, or other socially relevant groupings | This study did not include socially constructed or socially relevant variables. |
| Population characteristics | Between 70 and 90 year's old. Neuropathological diagnosis of Alzheimer's disease. |
| Recruitment | Samples were selected on the basis of neuropathological examination and brain tissue availability, which are unlikely to have impacted the results. |
| Ethics oversight | This study was performed at Netherlands Brain Bank, Amsterdam University Medical Centres (location VUmc), and the University of Leeds. This study was approved by both VUmc and the University of Leeds Research Ethics Committee. Informed consent was obtained from the patients' next of kin. |

Note that full information on the approval of the study protocol must also be provided in the manuscript.

# Field-specific reporting

Please select the one below that is the best fit for your research. If you are not sure, read the appropriate sections before making your selection.

☒ Life sciences  ☐ Behavioural & social sciences  ☐ Ecological, evolutionary & environmental sciences

For a reference copy of the document with all sections, see nature.com/documents/nr-reporting-summary-flat.pdf

# Life sciences study design

All studies must disclose on these points even when the disclosure is negative.

| | |
|---|---|
| Sample size | No sample size calculation was performed.  Since population-based variation was not investigated in this study, a single post-mortem AD and non-demented control donor brain sample (mit-temporal gyrus) was used for cryoET. Cingulate cortex from the same donor was used for cryEM of sarkosyl-insoluble tau. A single 10 months-old, male  App^NL-G-F/NL-G-F (Saito et al., 20214) mouse was used to confirm the effect of post-mortem interval. |
| Data exclusions | No tomograms were excluded from any of the datasets. For helical averaging of cryoET and cryoEM data, pre-established common image classification procedures (S.H.W. Scheres, J. Struc. Biol. 180: 519-530, (2012)) were employed to select particle images with the highest resolution content in the cryo-EM reconstruction process. Details of the number of selected images are given in Extended Data Table 5 and 8. |
| Replication | All attempts at replication were successful. At least three independent repeats per experiment where representative data is shown. |
| Randomization | Not applicable applicable to this study. Samples were allocated to experimental group on the basis of neuropathological examination. For subtomogram averaging, subvolumes were divided  into two random halves for gold standard Fourier shell correlation estimates of resolution. The same is true of for the averaging of cryoEM data, where the particles were randomly divided into half sets during refinement in RELION. |

| Blinding | Identification of the molecular and organelle constituents of tomograms was performed blind by two independent curators. For all other experiments, investigators were not blinded to allocation during experiments and outcome assessment. The perceived risk of detection/performance bias was deemed negligible. |
|---|---|

# Reporting for specific materials, systems and methods

We require information from authors about some types of materials, experimental systems and methods used in many studies. Here, indicate whether each material, system or method listed is relevant to your study. If you are not sure if a list item applies to your research, read the appropriate section before selecting a response.

## Materials & experimental systems

| n/a | Involved in the study |
|---|---|
| ☐ | ☒ Antibodies |
| ☒ | ☐ Eukaryotic cell lines |
| ☒ | ☐ Palaeontology and archaeology |
| ☐ | ☒ Animals and other organisms |
| ☒ | ☐ Clinical data |
| ☒ | ☐ Dual use research of concern |
| ☒ | ☐ Plants |

## Methods

| n/a | Involved in the study |
|---|---|
| ☒ | ☐ ChIP-seq |
| ☒ | ☐ Flow cytometry |
| ☒ | ☐ MRI-based neuroimaging |

## Antibodies

| Antibodies used | Immunohistochemistry primary antibodies: 1:800 anti-pTauSer202/Thr205 clone AT8 (MN1020, ThermoFisher Scientific, Waltham, MA, USA), 1:1000 anti-amyloid beta clone 4G8 (#800710, Biolegend, San Diego CA, USA), 1:6400 anti-pTau-Thr217 (#44-744, ThermoFisher Scientific), 1:1000 anti-P62-lck (#610833, BD Biosciences, CA), 1:500 anti-alpha-synuclein (phospho-S129, clone EP1536Y) (ab51253, Abcam, Cambridge, UK), 1:6000 anti-pTDP-43 Ser409/410 (TiP-PTD-M01, Cosmo Bio USA, CA, USA), 1:1000 anti-TMEM106B (C terminal, SAB2106773, Sigma-Aldrich). |
|---|---|
| | Immunohistochemistry secondary antibodies: Envision mouse/rabbit HRP (K4065, DAKO, Glostrup, Denmark). |
| | Immunoblot primary antibodies: 1:2000 anti-Tau clone Tau46 (aa 404-441, T9450, Merck), 1:1000 anti-pTauSer202/Thr205 clone AT8 (MN1020, ThermoFisher), 1:1000 anti-4-repeat Tau (aa 275-291, 05-804, Merck), 1:500 anti-3-repeat Tau (aa 267-316, 05-803, Merck), 1:1000 anti-C-terminal domain TMEM106B (SAB2106778, Merck). |
| | Immunoblot secondary antibodies: 1:20000 anti-mouse-HRP (170-6516, Bio-rad), 1:20000 anti-rabbit-HRP (172-1019, Bio-rad). |
| | Immunofluorescence primary antibodies: 1:750 anti-amyloid beta clone 6E10 (803001, Biolegend), 1:750 anti-amyloid beta clone 4G8 (800710, Biolegend), 1:750 anti-pTauSer202/Thr205 clone AT8 (MN1020, Thermofisher). |
| | Immunofluorescence secondary antibodies: 1:1000 anti-mouse IgG2B AF-633 (A21126, Thermo Fisher), 1:1000 anti-mouse IgG1 AF-568 (A21124, Thermo Fisher). |

| Validation | Immunohistochemistry primary antibodies: Anti-pTauSer202/Thr205 clone AT8: Validated extensively in the literature (e.g. Braak and Braak, Acta Neuropathol 82:239 (1991)) for immunohistochemical detection in human tissues. Anti-amyloid beta clone 4G8: Validated extensively in the literature (e.g. Forny-Germano L, et al. J Neurosci. 34:13629 (2014)) for immunohistochemical detection in human tissues. Anti-pTau-Thr217: Validated by Hart de Ruyter F, et al. Acta Neuropathol 145(2):197-218 (2023) for immunohistochemical detection in human tissues. Anti-P62-lck: Validated extensively in the literature (e.g. Kovacs G, et al. Neuropathology and Applied Neurobiology. 39:166-78 (2013)) for immunohistochemical detection in human tissues. Anti-alpha-synuclein (phospho-S129): Validated extensively in the literature (e.g. Lashuel et al. NPJ Parkinson Dis, 8:136 (2023); Delic et al., J Comp Neurol. 526:1978 (2018)) for immunohistochemical detection in human tissues. Anti-pTDP-43 Ser409/410: Validated extensively on human tissues in the literature (e.g. Kovacs et al., Neuropathology and Applied Neurobiology. 39:166 (2013)) for immunohistochemical detection in human tissues. Anti-TMEM106B: Validated by Perneel J, et al., Acta Neuropathologica 145:285 (2022) for immunohistochemical detection in human tissues. |
|---|---|
| | Immunohistochemistry secondary antibodies: Anti-mouse/rabbit HRP (DAKO, Glostrup, Denmark): Validated in many other studies (e.g. Hart de Ruyter et al., Acta Neuropathol 145:197 (2023); Boon et al., Acta Neuropathol 140:811 (2020). |
| | Immunoblot primary antibodies: anti-Tau clone Tau46 : Validated by Mawal-Dewan et al., J. Biol. Chem. 269:30981 (1994) for detection of human tau. Anti-pTauSer202/Thr205 clone AT8: Validated against human pS202/pT205 tau in (Mercken et al., Acta Neuropathol. 84, 265-272 (1992)) for detection of human tau. Anti-4-repeat Tau: Validated by Ercan et al., Molecular Neurodegeneration 12:87 (2017) for detection of human tau. Anti-3-repeat Tau: Validated by Ercan et al., Molecular Neurodegeneration 12:87 (2017) for detection of human tau. Anti-C-terminal domain TMEM106B: Validated by Perneel J, et al., Acta Neuropathologica 145:285 (2022) for detection of human TMEM106B  Anti-amyloid beta clone 6E10: Validated in earlier study using wild-type control (Leistner et al., Nat. Commun. 14:2833 (2023)) for detection of human beta-amyloid. Anti-amyloid beta clone 4G8: Validated in many earlier study (e.g. Poduslo et al., Biochem. 43:6064 (2004)) for detection of human beta-amyloid. |
| | Immunoblot secondary antibodies (anti-mouse-HRP and anti-rabbit-HRP):  Validated in manufacturer's (Bio-rad) website: https://www.bio-rad.com/en-uk/product/hrp-ap-conjugates?ID=cc14540b-25dd-4bbf-a17a-8c3c52116d9d): "Blotting-grade HRP conjugates (horseradish peroxidase) and AP conjugates (alkaline phosphatase) produce specific results, eliminating false positives in western blotting immunoassays: Double affinity-purified blotting-grade antibodies are isolated by affinity chromatography and further purified by cross-adsorption against an unrelated species to eliminate nonspecific immunoglobulins". |
| | Immunofluorescence primary antibodies: Anti-pTauSer202/Thr205 clone AT8: Validated by Dehkordi et al. Nat. Aging 1:1107 (2021). Anti-amyloid beta clone 4G8: Validated in many earlier study (e.g. Poduslo et al., Biochem. 43:6064 (2004)). |
| | Immunofluorescence secondary antibodies (anti-mouse IgG2B AF-633, anti-mouse IgG1 AF-568): Validated extensively in the literature (e.g. Leistner et al., Nat. Commun. 14:2833 (2023). |

# Animals and other research organisms

| | |
|---|---|
| Laboratory animals | 10 month old  App^NL-G-G/NL-G-F knockin mouse on a c57b/l6 background (Saito et al., Nat. Neurosci. 17:661 (20214). Housing conditions: 20–24°C , 12 hour day/night cycle, 45-65% relative humidity. |
| Wild animals | The study did not involve wild animals. |
| Reporting on sex | Only male animals were used in the study. Sex based analysis is not necessary because differences in the architecture of amyloid plaques is not expected to be significantly different in the male versus female App. |
| Field-collected samples | Study did not involve field-collected samples. |
| Ethics oversight | Oversight and approval was provided by the University of Leeds Animal Welfare and Ethics Review Board and licensed by the UK Government Home Office. |

Note that full information on the approval of the study protocol must also be provided in the manuscript.

# Plants

| | |
|---|---|
| Seed stocks | This study did not include plants. |
| Novel plant genotypes | This study did not include plants. |
| Authentication | This study did not include plants. |

