## [Peer Review file · Nature]

Manuscript Title: CryoET of β -amyloid and tau within post-mortem Alzheimer's disease brain

Reviewer Comments & Author Rebuttals

Reviewer Reports on the Initial Version:

Referees' comments:

Referee #1 (Remarks to the Author):

This paper describes exciting results on the use of cryo-electron tomography (cryo-ET) on high-pressure frozen brain tissue samples from an individual with Alzheimer's disease. This is a technological feat and the first ever report on such a study. As such, it could herald a new era of structural cellular biology of diseased human brain tissue. Therefore, in general, we are enthusiastic about this study, and would like to see it published in Nature eventually. However, there are a few major concerns that would need to be addressed before publication:

1) Performing cryo-ET on fresh brain tissue that is high-pressure frozen with a minimal postmortem delay would represent a fabulous step forward in our understanding of the cellular aspects of Alzheimer's pathology. However, this manuscript does not yet represent that milestone. It possibly borders on the misleading that this manuscript refers 30 times to the brain tissue using the word "fresh", whereas in fact the brain tissue was flash-frozen at -80C, then thawed and high-pressure frozen again. The only indirect reference to this crucial step is the following sentence in the Methods section on page 10: "Fresh, flash-frozen post-mortem AD and non-demented donor post-mortem brain were cryopreserved at -80°C and provided a source of tissue for these studies". Perhaps we have misunderstood this, and fresh brain tissue was indeed high-pressure frozen straight from the six-hour postmortem delays mentioned in the main text on page 3). In that case, this entire first point can be ignored. However, given the obvious logistical problems in high-pressure freezing fresh human brain tissue with minimal postmortem delays, we suspect that a crucial thaw-freeze cycle has been completely ignored in this manuscript. If this is the case, this step should be highlighted in Figure 1d, a detailed description of this step (at what temperature was thawing performed, how much time did thawing take, etc) should be added to the Methods section, and the possible effects of a thaw-freeze cycle should be openly discussed in the Results section. Reported postmortem delays in the main text should be the total amount of time the sample was out of the -80C freezer between death and high-pressure freezing. Since the same authors have recently reported a cryo-ET study on (indeed freshly high-pressure frozen) brain tissue from an amyloid-beta mouse model, they could perhaps do a control experiment that explores the effect of thaw-freezing on the mouse brain tissue. Differences between the tomograms of freshly high-pressure frozen mouse brain tissue and mouse brain tissue subjected to an additional thaw-freeze cycle could then be extrapolated to better understand the relevance of the presented results on the human tissue.

2) Even if the thaw-freeze cycle has some suboptimal effects on the preservation of the brain tissue, the same would also happen in the non-demented control brain. Therefore, the differences between AD and the control brain could still be relevant for our understanding of disease. Therefore, more data should be presented on the control brain. Currently, there are only 4 representative images in Extended Data Figure 5, some minimal statistics in Extended Data Table 3, plus one or two brief sentences in the Results section that mention the non-demented control. A revised version of the manuscript should make a more detailed comparison between the control and AD brains. What was the immunohistochemistry of the control brain like? Was CLEM performed on the control brain? If so, were there any areas of MX04 fluorescence in the control brain? And were they targeted for tomogram collection in the same way as the AD brain, or were tomograms taken at random positions?

3) Some of the assumed tau filaments reside extracellularly, whereas tau aggregation in disease primarily happens intracellularly. Extracellular tau aggregates are mainly thought to be ghost tangles that remain after the original neuron in which they formed has died. Do the authors think this is what has happened with the extracellular tau filaments observed? Or are these extracellular filaments perhaps a consequence of the thaw-freeze cycle? A discussion on this topic should be added to the manuscript.

4) The subtomogram averaging leads to relatively low-resolution structures for the amyloid-beta and tau filaments. For amyloid-beta filaments, basically no information is added by subtomogram averaging. Statements in the discussion that the averaged helical models for amyloid-beta were “broadly consistent” with ex vivo structures, but “additional amyloid species were also apparent” should therefore be removed. For tau, better results are described. One tomogram of the cryo-sections leads to an average structure with two C-shaped protofilaments that are consistent with PHFs (Figure 3j). For the lift-outs, the same is true for Fig 4i and possibly also Fig 4k. This is an exciting result, that PHFs can be recognized directly in human brain tissue samples! However, the authors then push their interpretation too far in stating that the structure in Fig4k better fits a CTE filament. At these resolutions, the differences between Fig 3j, Fig 4i and Fig4k are not clear. Why would Fig4i and Fig3j be the same, but Fig4k be different? Artefacts in the averaging procedure at low resolution provide a more convincing argument that they, in fact, probably belong to the same class of filaments. If the authors wish to maintain this claim, they should either drastically improve the resolution of the subtomogram averages (e.g. to the level individual beta-sheets are separated, i.e. at ~1nm). Or, they should show the presence of CTE filaments in this brain by complementary methods, like cryo-EM structures from sarkosyl insoluble fractions. Otherwise, this claim should be removed from the manuscript.

5) Regarding the identification of filament types, it is unclear to us why one would present the fits of tau structures from PSP, GGT and AGD in Extended Data Figures 8-10. Those are different diseases, from which this individual did not suffer, and these filaments are thus not expected in this brain. Why not compare to alpha-synuclein filaments then too? One relevant comparison that is not done, would be TMEM106B filaments, as these have been shown to be present in many human brains of advanced age. We’re not the sure the age of the non-demented control is given in the manuscript, but one might even see TMEM106B filaments in that case too? Their presence could be revealed by Western blots of sarkosyl insoluble fractions. In general, without subtomogram averages of sufficient resolutions, the

only indication that these filaments are in fact tau and amyloid-beta, as opposed to any other type of filament in the brain tissue comes from the interpretation of the MX04 fluorescence signal as amyloid plaques and tau tangles. The limitations of this type of lower-resolution images on the assignment of specific filament types should be openly discussed in the paper.

6) One conclusion is that the cross-over distances of tau PHFs in the human brain differ from those observed in sarkosyl insoluble fractions. Perhaps the authors could comment on the effects of the missing wedge, which lead to smearing of density in the direction of the filaments, on the estimation of cross-over distances?

Minor points:

1. How do authors prepare the brain slices? In the Methods session, please indicate which planes of the brain were imaged for all the tomograms, are they the same or different? It will help readers understand their relative orientations and position in the brain tissue.
2. Page 5, line 166, there is no Figure2e.
3. Page 5, line 170, outside of a myelinated axon should be figure 2c,d?
4. Method, page 12, line 463, ECTA should be EDTA?
5. Reference 51 is not right.
6. Page 28, line 1015, figure legend tau PHF (PDB 5osl), this PDB code is wrong. Please also check for other figure legends.
7. Extended data figure 2. Legend. b. Typo error. there are two "cyan arrowhead, b-amyloid plaque.
8. Extended data figure 7, legend. b, CTE is not plotted in b, but shown in the figure legend.
9. Extended data figure 9, typo, CTE type 2 and 2.
10. Extended data figure 15. Legend, line 317, a should be b, line 319, b should be c.

Referee #2 (Remarks to the Author):

This study is a technical tour de force in which the authors label AD plaques in fresh postmortem human brain tissue prior to vitrification. They then use fluorescence-guided cryosectioning and FIB milling to enable tilt series collection on both beta-amyloid and Tau fiber bundles. Subtomogram averaging and classification is used to show that filaments within each bundle adopt variable conformations. It is clearly written and the figures are easy to interpret.

The findings are both original and significant (given the sample origin), and the methods applied are state-of-the-art from a cryomicroscopy perspective. Given the evidence provided I believe the conclusions are sensible and do not over-reach. I do not have major concerns and think the study should be published. I believe the readership of Nature will be quite interested in the methods employed and the results.

Referee #3 (Remarks to the Author):

The manuscript entitled "Title: In situ cryo-electron tomography of β -amyloid and tau in post-mortem Alzheimer's disease brain" by Gilbert et al. presents the cryo EM tomography at nm resolution of both tau and Ab(1-42) deposits in a Alzheimer's disease brain (Fixation methods are deliberately avoided, leading to post-mortem tissue that closely resembles its native state upon high-pressure freezing) representing structural biology in vivo. This is not only a very impressive proof of concept study from a technological point of view but also an important contribution in the understanding of a structure activity relationship of protein amyloids. Several interesting and important findings are indicated such as the alignment of fibrils, that amyloid inclusions coexist with vesicles and damaged mitochondria as well as ApoE-like entities, the branching of fibrils indicating secondary nucleation, distinct properties from distinct sites (i.e. site-resolved heterogeneity), a correlation between the AD-purified tau cryo EM structures with the tomography with almost an order less resolution (and as such unfortunately, but not surprisingly, rather far away from atomic resolution). The combination of the two by mapping cryo EM structures into the tomography density is interesting while not entirely conclusive in particular for the Ab(1-42).

There are the following open points to be considered:

- (i) The study confirmed the appearance of the plaques ($A\beta$ fibrils, branched fibrils, and protofilament-like rods interlaced with non-amyloids structures), which are similar to their earlier studies with the FAD model. Is the fibril content also similar to earlier studies?
- (ii) In total 9 tomograms containing Tau fibrils (7 from cryo-sections and 2 from lift-out) were utilized separately with subtomogram averaging (Ex D.F.9,10,15). How can the authors explain the differences in subtomogram averages prepared by two different methods on the same sample?
- (iii) The reviewer recognizes that producing data with this workflow is very challenging. However, solving a structure from only two tomograms with 20 or 42 fibrils is not allowing sufficient statistics and resolution.
- (iv) The mapping of the atomic resolution structure onto the EM tomograms: The mapping is of qualitative nature. Can one quantify the mapping? In addition, while the PHF fits best it does not fit well (good, but not well). This might be explained by the standard deviation of the EM tomograms. Furthermore, the same polymorph is fit into the EM density of Figure 3i and 4j albeit both tomograms look really different (one with a hole, the other one not). This might be explainable again by the

accuracy and precision of the tomography, but may be also not and should be explained quantitatively by showing the standard deviation to be expected (may be by taking subensembles and show them side by side). Furthermore, the correlation between the cryo EM structure and the tomography slice in Figure 3k shows only some resemblance.

The resolution of the Ab(1-42) are even worse and it is unclear whether any subtyping can be made in particular since there might be also other proteins that are bound to the fibril surface as indicated by the tomogram of extended Figure 11d. Furthermore, the twist rate taking from the cryo EM structure may not be present as such. Thus, the strong statement on polymorph heterogeneity is questioned.

(v) The authors discuss that they observed parallel alignment of fibrils, which are all mostly vertically oriented. However, how does the missing wedge and therefore the anisotropic resolution in tomography affect this finding? Would horizontally aligned fibrils be detected at similar contouring thresholds?

(vi) Of great curiosity is the parallel arrangement of the fibrils. Can this be rationalized? does it follow a membrane close by? Do the fibrils also have similar length or is this an artifact? Do the Ab(1-42) fibrils have a common origin going outward with branching?

(vii) Use of twist rate to state polymorph difference: There are several reports that show that the twist rate has often nothing to do with a distinct polymorph. Most often fibril change the twist rate over length and Mezzenga demonstrated that the twist rate can be manipulated by salt only without request of another polymorph following polymer chemistry argumentation. Hence, while at this low resolution interesting to see, a different twist rate of the fibrils has very little information content unfortunately.

Minor points:

(a) In the abstract, the authors claim to have “determined the structure” of amyloid and tau (Line 28). This is a misleading statement, as “determining the structure” usually refers to higher-resolving work where the secondary structure details are recognized. This wording should be toned down.

(b) The authors also claim that the tau fibril heterogeneity can be explained by the subcellular tissue environment (Line 37/38). However, this is not shown anywhere (only that there is a difference between tau fibril clusters that are spatially distributed).

(c) In line 85-90, the authors claim to have studied the 3D architecture of plaques and tangles. This wording should be toned down, as the authors have only studied a slim lamella from the tissue, not the entire plaques or tangles.

(d) Please chose one form of notation i.e. cryo-ET, cryo-FM.

(e) The projection thickness of tomographic slices has not been defined, please add the related information.

(f) In line 396, please clarify that there are two sets of samples for each donor, frozen at -80C and fresh. The sentence reads as if the fresh sample was frozen.

(g) It is stated that the Ab(1-42) in vitro structure differs from the brain derived material. This is not the case albeit often stated. The solid state NMR structure is very similar to the type 1 structure having 50% the N-terminal residues folding a b-strand as in the case of the cryo EM structure and 50% not folded.

(h) In Ex D.F.2-c cyan arrows are not easy to detect in cyan background. Also, in the same figure Tau threads labelled by open orange arrowheads are not easily seen.

(i) In Ex D.F.2, please keep the same color code for the same feature at different figure panels if possible.

(j) In Supp., in line 50 "Cyan arrowhead, β -amyloid plaque." is repeated twice.

(k) In Ex D.F.3-b right panel, please remove open cyan arrowheads, as it wasn't explained in the legend what they are pointing to.

(l) In Ex D.F.3-b, how can it be confirmed that these unlabelled fibrils are truly amyloids and not neurofilaments or vimentin?

(n) In Ex D.F.4, for an easier read instead of many colorful arrowheads, abbreviated terms can be used. For example, "EV or ExV" for extracellular vesicles.

(m) In Ex D.F.7-b, CTE-1 data is not plotted on the graph.

(o) In Ex D.F.7-b, the helical twist was redefined by Scheres (Scheres S. H. W. (2020). Acta crystallographica. 76(Pt 2), 94–101.) as $4.75 \text{ Angstroms (rise)} * 180 \text{ degree/crossover distance}$. For a 750 Angstroms crossover distance, helical twist can be estimated as -1.14 degrees, which then is per rung and not per nm. In the graph, twist is specified in degrees per nm, which is unusual.

(p) In ex. D.F.7.-c: How was classification done? PCA is a dimension-reduction method, after which one of several possible classification algorithms can be applied. K-means, HA, or others are possible examples. Which was used here, and how was it done?

(q) In Ex D.F.8., please add the thickness of the tomographic slices/projections at three positions.

(r) In Ex D.F.10, the resolution is not sufficient to allow significant docking experiments. The wording in lines 202-205 should be toned down.

(s) In Ex D.F.10-f, this polymorph does not fit any ex vivo model. This opens a range of potential interpretations for its nature and origin. Could you please discuss it?

(t) In Ex D.F.14, green, brown and yellow rectangles are difficult to see.

Author Rebuttals to Initial Comments:

Referee #1 (Remarks to the Author):

This paper describes exciting results on the use of cryo-electron tomography (cryo-ET) on high-pressure frozen brain tissue samples from an individual with Alzheimer's disease. This is a technological feat and the first ever report on such a study. As such, it could herald a new era of structural cellular biology of diseased human brain tissue. Therefore, in general, we are enthusiastic about this study, and would like to see it published in Nature eventually. However, there are a few major concerns that would need to be addressed before publication:

We thank the reviewer for their careful reading of our manuscript and for recognising our manuscript as the first in situ cryoET of human brain and technological feat (first cryoCLEM-guided cryoFIB-SEM of human brain). We have addressed each question and comment below (see also reference list).

1) Performing cryo-ET on fresh brain tissue that is high-pressure frozen with a minimal post-mortem delay would represent a fabulous step forward in our understanding of the cellular aspects of Alzheimer's pathology. However, this manuscript does not yet represent that milestone. It possibly borders on the misleading that this manuscript refers 30 times to the brain tissue using the word "fresh", whereas in fact the brain tissue was flash-frozen at -80C, then thawed and high-pressure frozen again. The only indirect reference to this crucial step is the following sentence in the Methods section on page 10: "Fresh, flash-frozen post-mortem AD and non-demented donor post-mortem brain were cryopreserved at -80°C and provided a source of tissue for these studies". Perhaps we have misunderstood this, and fresh brain tissue was indeed high-pressure frozen straight from the six-hour postmortem delays mentioned in the main text on page 3). In that case, this entire first point can be ignored. However, given the obvious logistical problems in high-pressure freezing fresh human brain tissue with minimal postmortem delays, we suspect that a crucial thaw-freeze cycle has been completely ignored in this manuscript. If this is the case, this step should be highlighted in Figure 1d, a detailed description of this step (at what temperature was thawing performed, how much time did thawing take, etc) should be added to the Methods section, and the possible effects of a thaw-freeze cycle should be openly discussed in the Results section. Reported postmortem delays in the main text should be the total amount of time the sample was out of the -80C freezer between death and high-pressure freezing.

The reviewer is correct, as stated in the methods section of our submitted manuscript that fresh post-mortem tissue samples were flash-frozen in liquid nitrogen and that this was the source of tissue that was subsequently thawed and prepared for high-pressure freezing. In our revised manuscript we have expanded the **Methods** section entitled 'MX04-labelling and high-pressure freezing freeze-thawed post-mortem acute brain slices', accordingly. We have also amended **Fig. 1d** schematic to explicitly state this step of the work-flow as requested.

Since the same authors have recently reported a cryo-ET study on (indeed freshly high-pressure frozen) brain tissue from an amyloid-beta mouse model, they could perhaps do a control experiment that explores the effect of thaw-freezing on the mouse brain tissue. Differences between the tomograms of freshly high-pressure frozen mouse brain tissue and mouse brain tissue subjected to an additional thaw-freeze cycle could then be extrapolated to better understand the relevance of the presented results on the human tissue.

The reviewer suggests a control in which we demonstrate the effect of a 'post-mortem interval (PMI)' and freeze-thawing of tissue using *App*^{NL-G-F} mouse brain by comparing such a sample directly with our previously reported cryoET study in which tissues were prepared without a 'post-mortem' interval or a freeze-thaw step (Leistner et al., 2023). We have now done this, culling mice and leaving them at room temperature for 6 hours before dissecting the forebrain and freezing it in liquid nitrogen to simulate the conditions experienced by post-mortem human AD brain. We then processed these mouse brains following the same protocol we developed for human post-mortem tissues (preparing high-pressure frozen samples, cryo-sections, cryoFM imaging, cryoCLEM and cryoET). We collected 19 and 41 tomograms on and around methoxy-04-labelled β -amyloid, respectively. We compared these to our earlier publication on cryoET of ultra-fresh (i.e. lacking this post-mortem interval and freezing-thawing step) *App*^{NL-G-F} mouse brain (see **ED Fig. 6-7** and **ED Table 7**). Below are key findings:

a) The architecture of amyloid in mouse brain that had undergone a 6 hour delay before freeze-thawing (c.f. ~6 hours PMI in human samples) appeared indistinguishable from tissue that was high-pressure frozen directly (see **ED Fig. 6**). In particular, we observed parallel bundles and a mesh of β -amyloid fibrils. This suggests that the post-mortem interval and freeze-thawing of the tissue does not have a major effect on the in-tissue architecture of the amyloid. This is reassuring since almost all cryoEM studies of purified amyloid (including (Falcon et al., 2018; Fitzpatrick et al., 2017; Yang et al., 2022) have been performed using brain tissue that has a similar or longer post-mortem interval and has been freeze-thawed before preparing amyloid for cryoEM.

b) As expected, damaged mitochondria were apparent in 'PMI-6h freeze-thaw' *App*^{NL-G-F} tomograms (see **ED Fig. 7** in revised manuscript) but absent from the ultra-fresh *App*^{NL-G-F} tissues used in our earlier publication (Leistner et al., 2023). In our current manuscript, we described damaged mitochondria present in both cognitively normal and AD post-mortem brain cryoET datasets. This is consistent with a post-mortem interval or freeze-thawing causing mitochondrial damage.

c) As expected, microtubules were absent in the '6 h PMI freeze-thaw' *App*^{NL-G-F} tomograms compared to ultra-fresh *App*^{NL-G-F} tissues used in our earlier publication, which reported numerous microtubules (see **ED Fig. 7 & 8** in our revised manuscript). Microtubules are known to depolymerise when starved of nucleoside triphosphates or freeze-thawed (Mitchison, 1995; Pollard, 1976). In accordance, we did not observe microtubules in non-demented and AD post-mortem tomograms of samples that had a 6 h PMI and freeze-thaw step.

d) Differences arose in membrane integrity when comparing *App*^{NL-G-F} tissue with a 6 h PMI and freeze-thaw step to *App*^{NL-G-F} tissue that was directly vitrified. We observed open membrane sheets in two tomograms, and ruptured plasma membrane in 10 of 60 tomograms from the '6 h PMI freeze-thaw' *App*^{NL-G-F} dataset. We observed open membrane sheets in 10 tomograms, and ruptured plasma membrane in 11 tomograms out of the 80 tomograms collected from human AD post-mortem brain. Ruptured plasma membranes were not observed in our earlier publication of ultra-fresh *App*^{NL-G-F} mice, nor in non-demented post-mortem human brain. We describe in our revised manuscript that amyloid plaques in post-mortem AD brain contained fragments of membrane intermingled with amyloid fibrils that were absent in our recently published paper on mouse in-tissue cryoET (Leistner et al., 2023).

We have summarised the above points in the **Results** and **Methods** sections of our revised manuscript. Additionally, since submitting our manuscript a preprint was made available from (Creekmore et al., 2023) showing post-mortem brain that has not undergone freeze thawing, but has a much longer post-mortem interval (12-18 hours). While this report was not able to identify macromolecules definitively as a consequence of the thickness of the lamella, the use of 1 M trehalose, and the lack of a cryoCLEM label, this preprint manuscript provided evidence that putative microtubules were intact when samples were not freeze-thawed, in accordance with our experiments testing the effect of the PMI and freeze-thaw step in *App*^{NL-G-F} brain tissues.

2) Even if the thaw-freeze cycle has some suboptimal effects on the preservation of the brain tissue, the same would also happen in the non-demented control brain. Therefore, the differences between AD and the control brain could still be relevant for our understanding of disease. Therefore, more data should be presented on the control brain. Currently, there are only 4 representative images in Extended Data Figure 5, some minimal statistics in Extended Data Table 3, plus one or two brief sentences in the Results section that mention the non-demented control. A revised version of the manuscript should make a more detailed comparison between the control and AD brains.

We agree with the reviewer that any suboptimal effects of freeze-thawing also apply to the non-demented control sample and that it is on the basis of comparisons of cryoET from AD brain

compared to non-demented control tissues that we are able to deduce which structures in the tissue are likely to be associated with disease.

The reviewer requested more data on the control brain and comparison with AD brain. We have now included more example tomographic slices from 14 non-demented control brain tomograms (**ED Fig. 6**) and a more in-depth analysis of key features conserved between control and AD post-mortem tissues, including damaged mitochondria with swollen cristae and dilute mitochondrial matrix, the presence of some ribosomes, and the absence of microtubules (**ED Fig. 7**). These are a likely effect of freeze-thaw cycle on 6 hr post-mortem human brain tissue which we now report in our revised **Results** section.

We would also like to highlight that it is important not to over interpret in-tissue cryoET data and only make like-for-like comparisons. Taking this cautious approach we have only used non-demented control tissue to a) confirm the specific MX04 labelling of AD pathology in high-pressure frozen tissue and b) to differentiate molecular and cellular features that are associated with AD, particularly non-amyloid constituents of plaques. In our revised manuscript we have now included a survey of all molecular constituents that could be reliably identified in the raw tomographic maps (see revised **ED Tables 1-4 and 7**). In particular, we only observed cuboidal extracellular droplets and open membrane sheets in post-mortem AD brain and not in post-mortem non-demented control brain.

What was the immunohistochemistry of the control brain like?

Immunohistochemistry of control brain confirmed the absence of A β or tau aggregates or any other MX04-labelled amyloid. See **ED Fig. 3a** in our revised manuscript.

Was CLEM performed on the control brain? If so, were there any areas of MX04 fluorescence in the control brain? And were they targeted for tomogram collection in the same way as the AD brain, or were tomograms taken at random positions?

The non-demented control brain was also labelled with the cryoCLEM label, MX04, by the same procedures as AD donor tissue (see **ED Fig. 3b**). Non-demented control brain tissue was from grey matter in the same cortical brain region as the AD donor tissue, but did not contain MX04-labelled amyloid (**ED Fig. 3b**, right panel). In the absence of MX04-labelled amyloid, non-demented control tomograms were collected at as many positions as possible throughout the tissue cryo-sections (dataset of 64 tomograms). This encompassed intracellular and extracellular locations that are comparable to those collected from AD donor samples. In the **Methods** section

'MX04-labelling and high-pressure freezing freeze-thawed post-mortem acute brain slices' of our revised manuscript, we have included these important details.

3) Some of the assumed tau filaments reside extracellularly, whereas tau aggregation in disease primarily happens intracellularly. Extracellular tau aggregates are mainly thought to be ghost tangles that remain after the original neuron in which they formed has died. Do the authors think this is what has happened with the extracellular tau filaments observed? Or are these extracellular filaments perhaps a consequence of the thaw-freeze cycle? A discussion on this topic should be added to the manuscript.

The reviewer correctly states in point '2' above that we have compared AD to non-demented control donor tissue tomograms to establish what structures were likely associated with disease versus those that could be a consequence of freeze-thawing. In the non-demented control cryoET dataset along with '6 h PMI freeze-thaw' *App*^{NL-G-F} dataset, there was no evidence of membranes being completely lost whereby intracellular organelles such as mitochondria would be found entirely extracellularly. Therefore, it is unlikely that freeze-thawing caused tau filaments and intracellular organelles such as the mitochondrion in this tomogram to reside in extracellular locations (see **Fig. 2** in revised manuscript). Therefore, these data are most consistent with showing the in-tissue molecular architecture of ghost tangles. We have included a discussion of this point in **Results** section of our revised manuscript as requested.

It is also worth noting that subtomogram averaging of these extracellular filaments reached 8.7 Å resolution (see response below), revealing the backbone trace of tau and confirming that these are undoubtedly PHF (see **Fig. 3j** in our revised manuscript and our response to point 4 below).

4) The subtomogram averaging leads to relatively low-resolution structures for the amyloid-beta and tau filaments. For amyloid-beta filaments, basically no information is added by subtomogram averaging. Statements in the discussion that the averaged helical models for amyloid-beta were "broadly consistent" with ex vivo structures, but "additional amyloid species were also apparent" should therefore be removed.

We agree with the reviewer and we have removed these statements as requested.

For tau, better results are described. One tomogram of the cryo-sections leads to an average structure with two C-shaped protofilaments that are consistent with PHFs (Figure 3j). For the lift-

outs, the same is true for Fig 4i and possibly also Fig 4k. This is an exciting result, that PHFs can be recognized directly in human brain tissue samples!

We thank the reviewer and share their excitement. See improvement in subtomogram averaging described below!

However, the authors then push their interpretation too far in stating that the structure in Fig4k better fits a CTE filament. At these resolutions, the differences between Fig 3j, Fig 4i and Fig4k are not clear. Why would Fig4i and Fig3j be the same, but Fig4k be different?

Artefacts in the averaging procedure at low resolution provide a more convincing argument that they, in fact, probably belong to the same class of filaments. If the authors wish to maintain this claim, they should either drastically improve the resolution of the subtomogram averages (e.g. to the level individual beta-sheets are separated, i.e. at ~1nm). Or, they should show the presence of CTE filaments in this brain by complementary methods, like cryo-EM structures from sarkosyl insoluble fractions. Otherwise, this claim should be removed from the manuscript.

The reviewer raised a question that the resolution of our subtomogram averaging may be insufficient to determine the fold of filaments in our tomograms and they suggest that we must obtain higher (<1 nm) resolution. We agree, and we have now implemented an improved subtomogram averaging pipeline to these cryoET data using Warp-PEET-Relion-M. This has increased the resolution (to 8.7 Å) such that we can now resolve the backbone trace of each tau molecule in the filament from a single cryo-section tomogram containing 136 filaments. Importantly, subtomogram helical averaging in Relion was able to determine the polarity of each filament, which otherwise could result in averaging filaments with opposite polarity, which we suspect arose with subtomogram averaging using PEET alone. The revised manuscript contains a detailed description of this new subtomogram averaging pipeline in the **Methods** section and the results of this improved subtomogram averaging are shown in **Fig 3i** and **ED Fig. 11**.

We also applied this new pipeline to each cluster of tau filaments in our dataset, on per-tomogram basis, which improved the resolution of all, but two, of the tau filament clusters, including the cryoFIB-SEM liftout tomograms. The data in the liftout tomograms was of sufficient quality for Relion to estimate the polarity of each filament sub-volume, which then gave a 25 Å resolution map of straight filament (see **Fig. 4h-k** in our revised manuscript). Importantly, this also confirmed the reviewer's suggestion that the CTE-like cluster was incorrect. We thank the referee for encouraging us to improve our subtomogram averaging pipeline.

The variation in resolution of subtomogram averaging performed on a per tomogram basis (8.7-33 Å resolution) seemed to correlate with a number of variables in the data, including copy number

of filaments, the orientation of filaments in the tomogram (the more axial the filament the better), cryo-section/liftout lamella thickness, and tomogram quality (number of tilt increments). Additionally, we used 'EM placement' (Millán et al., 2023; Read et al., 2023) to score the fit of atomic models (PHF versus SF) into in-tissue cryoET average maps, which gave an absolute score (log-likelihood gain, LLG) suggesting a better fit for either PHF or SF in our subtomogram averages. In our revised manuscript, we have included EM place LLG scores in the **Results** section (see also right panels of **Fig. 3i, 4j** and **4k** in revised manuscript).

Overall, in our revised manuscript we now interpret the fold of a tau filament subtomogram average in which β -sheets were resolved. For lower resolution tau cluster subtomogram averages, we suggest PHF versus SF ultrastructural polymorphs on the basis of EM placement fitting of atomic models.

The reviewer also suggested performing purification of sarkosyl-insoluble tau to obtain a helically averaged cryoEM structure from this donor. This we have done with a small (~500 mg) block of tissue from a different brain region (cingulate gyrus) from the same donor that provided middle temporal gyrus used for in-tissue cryoET. We obtained a 3 Å resolution helically averaged structure of PHF and a low-resolution average for SF. See **ED Fig. 2b-f** in our revised manuscript. These data confirm that the AD case is comparable to previous AD cases used for structural investigations of ex vivo, sarkosyl-extracted tau (Fitzpatrick et al, 2017 and Falcon et al., 2018). We did not identify the CTE fold in the sarkosyl-insoluble sample, as expected given the new cryoET/subtomogram averaging results at higher resolution discussed above.

5) Regarding the identification of filament types, it is unclear to us why one would present the fits of tau structures from PSP, GGT and AGD in Extended Data Figures 8-10. Those are different diseases, from which this individual did not suffer, and these filaments are thus not expected in this brain. Why not compare to alpha-synuclein filaments then too? One relevant comparison that is not done, would be TMEM106B filaments, as these have been shown to be present in many human brains of advanced age. We're not the sure the age of the non-demented control is given in the manuscript, but one might even see TMEM106B filaments in that case too? Their presence could be revealed by Western blots of sarkosyl insoluble fractions. In general, without subtomogram averages of sufficient resolutions, the only indication that these filaments are in fact tau and amyloid-beta, as opposed to any other type of filament in the brain tissue comes from the interpretation of the MX04 fluorescence signal as amyloid plaques and tau tangles. The limitations of this type of lower-resolution images on the assignment of specific filament types should be openly discussed in the paper.

We agree with the reviewers that PSP, GGT and AGD are different diseases from AD and are not expected in this donor brain. We have thus removed the Extended Data Figures showing docking into non-AD tauopathy structures of the tau filament. Instead, to assess quantitatively the similarity of subtomogram averaged in-tissue tau with atomic models of sarkosyl-extracted tau, we have

used EM place tool (as described above). We have included a brief discussion of the limitations of lower-resolution cryoET maps in our revised manuscript, as requested.

The reviewer also raises the question of whether TM106B filaments were present in our donor sample. We have performed TMEM106B (C-terminal epitope) immunohistochemical staining of tissue and TMEM106B immunoblots of sarkosyl insoluble fractions as requested by the reviewer, which revealed the presence of TM106B. However, we were unable to detect sufficient TM106B filaments in our cryoEM averaging to obtain a structure. This is consistent with TM106B being a minor population and not sufficient for cryoEM. On the basis of subtomogram averaging (**Figs. 3j, 4i, 4k** and **ED Fig. 11** of our revised manuscript), our cryoET dataset did not contain TMEM106B fibrils, which is consistent with scarcity of this amyloid in ex vivo, sarkosyl-extracted amyloid (**ED Fig. 2b-f**).

6) One conclusion is that the cross-over distances of tau PHFs in the human brain differ from those observed in sarkosyl insoluble fractions. Perhaps the authors could comment on the effects of the missing wedge, which lead to smearing of density in the direction of the filaments, on the estimation of cross-over distances?

We agree with the reviewer that the resolution of cryoET is anisotropic (less in Z than in X or Y) because of the missing wedge. As a consequence, we found that in tomograms with tau filaments oriented on the missing wedge ('axial'), the helical twist (rather than crossovers per se) of each individual tau filament could be directly observed (see **Fig. 3g-h** in our revised manuscript containing raw tomographic density of filaments showing the twist of individual filaments). In contrast, filaments oriented on the X-Y plane (in-plane filaments), the cross-section of the filament is worse because of the 'smearing' effect of the missing wedge. This is also shown in two movies, 'axial' tau filaments in **ED Movie 5** versus 'in-plane' filaments in **ED Movie 6**.

To increase accuracy in filament twist estimates we subtomogram averaged each 'axial' cluster of tau filaments independently. First, individual filaments (not segments of filaments) were aligned and averaged in PEET (stalkinit). These whole filament averages were used to obtain an estimate of twist (degrees per nanometre, see **Fig. 3i, 4h, 4j** and **ED Fig. 11** in our revised manuscript) from which crossover distance (filament length to rotate 180°) was calculated (see **ED Table 5**). These twist estimates were used for further subtomogram averaging in Relion and showed that filament clusters were different to each other (see **ED Fig. 17**). Some tau filament clusters were also different to that previously reported for cryoEM structures of sarkosyl-insoluble tau (Fitzpatrick et al, 2017, Falcon et al, 2018). We have amended the subtomogram averaging **Methods** section to clarify the above points.

As an additional control, we have also confirmed that the pixel size at the nominal magnification (53,000x) of our Krios was correctly calibrated using a cryoET dataset of a sample of known structure (ribosome) collected with the same settings as that we used our AD tissue.

Minor points:

We thank you the reviewer for spotting the minor points below, all of which we have now amended accordingly.

1. How do authors prepare the brain slices? In the Methods session, please indicate which planes of the brain were imaged for all the tomograms, are they the same or different? It will help readers understand their relative orientations and position in the brain tissue.

To prepare samples for cryoET acute slices were prepared from a block containing middle temporal gyrus. Acute slices were cut along the plane (horizontal or coronal) containing all layers of the cortex as well as underlying white matter. We have amended the **Methods** section entitled 'MX04-labelling and high-pressure freezing freeze-thawed post-mortem acute brain slices' in our revised manuscript to include this information.

Thank you for spotting the minor errors below.

2. Page 5, line 166, there is no Figure2e.

This has been amended, accordingly.

3. Page 5, line 170, outside of a myelinated axon should be figure 2c,d?

This been amended. accordingly.

4. Method, page 12, line 463, ECTA should be EDTA?

Amended accordingly.

5. Reference 51 is not right.

Amended accordingly.

6. Page 28, line 1015, figure legend tau PHF (PDB 5osl), this PDB code is wrong. Please also check for other figure legends.

Amended accordingly.

7. Extended data figure 2. Legend. b. Typo error. there are two “cyan arrowhead, b-amyloid plaque.

Amended accordingly.

8. Extended data figure 7, legend. b, CTE is not plotted in b, but shown in the figure legend.

Amended accordingly.

9. Extended data figure 9, typo, CTE type 2 and 2.

Amended accordingly.

10. Extended data figure 15. Legend, line 317, a should be b, line 319, b should be c.

Amended accordingly.

Referee #2 (Remarks to the Author):

This study is a technical tour de force in which the authors label AD plaques in fresh

postmortem human brain tissue prior to vitrification. They then use fluorescence-guided cryosectioning and FIB milling to enable tilt series collection on both beta-amyloid and Tau fiber bundles. Subtomogram averaging and classification is used to show that filaments within each bundle adopt variable conformations. It is clearly written and the figures are easy to interpret.

The findings are both original and significant (given the sample origin), and the methods applied are state-of-the-art from a cryomicroscopy perspective. Given the evidence provided I believe the conclusions are sensible and do not over-reach. I do not have major concerns and think the study should be published. I believe the readership of Nature will be quite interested in the methods employed and the results.

We thank the reviewer for recognising that our manuscript is a technical tour de force and the original and significant insights that it provides. In our revised manuscript, we have further improved the resolution of our subtomogram averages revealing the polypeptide backbone trace of the core of PHFs from a cluster of 136 filaments in a single tomogram.

Referee #3 (Remarks to the Author):

The manuscript entitled "Title: In situ cryo-electron tomography of β -amyloid and tau in post-mortem Alzheimer's disease brain" by Gilbert et al. presents the cryo EM tomography at nm resolution of both tau and Ab(1-42) deposits in a Alzheimer's disease brain (Fixation methods are deliberately avoided, leading to post-mortem tissue that closely resembles its native state upon high-pressure freezing) representing structural biology in vivo. This is not only a very impressive proof of concept study from a technological point of view but also an important contribution in the understanding of a structure activity relationship of protein amyloids. Several interesting and important findings are indicated such as the alignment of fibrils, that amyloid inclusions coexist with vesicles and damaged mitochondria as well as ApoE-like entities, the branching of fibrils indicating secondary nucleation, distinct properties from distinct sites (i.e. site-resolved heterogeneity), a correlation between the AD-purified tau cryo EM structures with the tomography with almost an order less resolution (and as such unfortunately, but not surprisingly, rather far away from atomic resolution). The combination of the two by mapping cryo EM structures into the tomography density is interesting while not entirely conclusive in particular for the Ab(1-42). There are the following open points to be considered:

We thank the reviewer for appreciating the results reported in our manuscript and for their careful reading and comments on the text and figures. We have addressed each comment below. Importantly, in response to all reviewers we have implemented a new subtomogram averaging pipeline that has greatly improved the resolution of our subtomogram averages of tau filaments,

which we report in our revised manuscript. We have addressed each question and comment below (see also reference list).

(i) The study confirmed the appearance of the plaques (A β fibrils, branched fibrils, and protofilament-like rods interlaced with non-amyloids structures), which are similar to their earlier studies with the FAD model. Is the fibril content also similar to earlier studies?

In broad terms, yes. In the **Discussion** section of our revised manuscript we have included references to earlier studies discussing key similarities and differences related to the architecture of A β fibrils and tau tangles. In so doing we have avoided a comparison of every feature observed in our cryoET dataset because the tissue sample in earlier classical EM experiments is altered by the method of preparation compared to cryoET methods and therefore detailed like-for-like comparisons are unsafe. In particular, it is not possible to determine if detailed differences between cryoET volumes and earlier 2D conventional EM images are a consequence of sample preparation or the sample itself. Below we have summarised the key difference between sample preparation for conventional resin-embedded EM and the cryoET study reported in our manuscript.

Earlier electron microscopy studies of β -amyloid plaques over the last ~60 years employed methods involving chemical cross-linking (shrinking the tissue), washing in organic solvents (denaturing proteins and removing much of the cytoplasm), post-fixing in osmium salts (further denaturing proteins), embedding in plastic, and heavy metal staining the remaining contents of the tissue. Tissues were then sectioned and 2D electron microscopy images collected. These data (key references include (Kidd, 1964; Terry et al., 1964)) provided important morphological evidence that amyloid plaques were composed of fibrils and filaments.

In-tissue cryoET reported here used sample preparation methods that are entirely different by avoiding steps that shrink the sample, denature proteins or wash away constituents of the tissue. Instead, tissue samples were cryo-preserved, cryo-sectioned or cryoFIB-milled before cryoEM imaging of macromolecules, organelles and subcellular compartments in their native state rather than a heavy metal stain and reconstructing the 3D volume of the tissue from tilted series cryoEM micrographs. Of course, there are some differences caused by freeze thawing as we discuss in response to Referee 1 above.

(ii) In total 9 tomograms containing Tau fibrils (7 from cryo-sections and 2 from lift-out) were utilized separately with subtomogram averaging (Ex D.F.9,10,15). How can the authors explain the differences in subtomogram averages prepared by two different methods on the same sample?

The reviewer asks how we can explain the differences in subtomogram averages prepared by cryo-sectioning versus cryoFIB-SEM liftout lamella preparation. We did not observe any systematic difference in the structure of tau filaments from cryo-section versus liftout lamellae tomograms. Each tau filament cluster (one per tomogram) showed differences in structure (summarised below).

Nine tomograms had a sufficiently high copy number of tau filaments for subtomogram averaging (7 cryosections and 2 liftout lamellae). Importantly, in our revised manuscript we now report an improvement in subtomogram averaging using a Warp-PEET-Relion-M pipeline) that produced subtomogram averages up to 8.7 Å resolution.

Initial averaging was performed using whole filaments (not segments of filaments) in PEET (stalkinit) suggested structural heterogeneity in which filaments in the same cluster were more similar than filaments in different clusters (**Fig. 3i, 4h, 4j** and **ED Fig. 11**). Therefore, we initially averaged filaments on a per cluster basis. This showed that most clusters (4/7 cryosection tomograms) were consistent with the PHF ultrastructural polymorph (**ED Fig. 11**). A subset of clusters (3/7 cryosections) were not consistent with either PHF or SF ultrastructural polymorph and the subtomogram average was not of sufficient quality to identify the fold of these two clusters of MX04-labelled amyloid (**ED Fig. 11e-f**). All tau clusters, including PHFs, varied in their twistedness (**ED Fig. 17**). It was striking that this variability of twist was spatially restricted.

Subtomogram averaging of tau filament clusters in liftout lamellae suggested a cluster composed of both PHF and SF in one cluster and SF filaments only in another cluster.

While the best map that revealed the backbone trace of tau filaments came from subtomogram averaging a cluster of 136 filaments in a single cryo-section tomogram, the quality of subtomogram averaged maps obtained from every other cluster of tau filaments varied (8.7-32.8 Å resolution). This variation in resolution could be attributed to a number of parameters in the data, including copy number of filaments, the orientation of filaments in the tomogram (the more axial the filament, the better), cryo-section/liftout lamella thickness, and tomogram quality (number of tilt increments) (see **ED Table 5**). We have revised the manuscript to include a brief discussion of what was resolved by subtomogram averaging of our cryoET dataset to address this important point raised by the reviewer.

(iii) The reviewer recognizes that producing data with this workflow is very challenging. However, solving a structure from only two tomograms with 20 or 42 fibrils is not allowing sufficient statistics and resolution.

We agree with the reviewer. We performed subtomogram averaging of tau filaments from single tomograms containing 64-278 filaments, which was successful in resolving internal features of tau filaments including helical twist, and the fold of the protein. In contrast, A β fibrils are much thinner and smoother and thus a greater challenge for subtomogram averaging. We also noted the apparent heterogeneity of beta-amyloid in each tomogram, which was different to tau and is the major insight from these limited subtomogram averages. We have toned down the description of A β fibril averages in the **Results** and **Discussion** sections of the revised manuscript, accordingly.

(iv) The mapping of the atomic resolution structure onto the EM tomograms: The mapping is of qualitative nature. Can one quantify the mapping? In addition, while the PHF fits best it does not fit well (good, but not well). This might be explained by the standard deviation of the EM tomograms. Furthermore, the same polymorph is fit into the EM density of Figure 3i and 4j albeit both tomograms look really different (one with a hole, the other one not). This might be explainable again by the accuracy and precision of the tomography, but may be also not and should be explained quantitatively by showing the standard deviation to be expected (may be by taking subensembles and show them side by side). Furthermore, the correlation between the cryo EM structure and the tomography slice in Figure 3k shows only some resemblance.

The reviewer asks a good question, whether the fitting of atomic models to lower resolution in situ subtomogram averaged maps can be quantified. We have now done this using a recently developed computational tool, 'EM placement' (Millán et al., 2023; Read et al., 2023), which uses half-maps from subtomogram averaging to calculate a log likelihood gain (LLG) and cross correlation coefficient for the fit of an atomic model into the average map. These quantitative scores account for the fit of the atomic model, the resolution of the map and provide an absolute quantitative score, enabling comparison of the score of different models into the same map. In our revised manuscript we have included EM placement scores to give a likelihood-based score for comparing subtomogram average maps to atomic models of ex vivo/sarkosyl-extracted tau filaments. The result of EM placement are shown in **Fig 3j (right panel)**, **4i (right panel)**, **4k (right panel)**. See also **ED Fig. 11**.

The resolution of the Ab(1-42) are even worse and it is unclear whether any subtyping can be made in particular since there might be also other proteins that are bound to the fibril surface as indicated by the tomogram of extended Figure 11d. Furthermore, the twist rate taken from the cryo EM structure may not be present as such. Thus, the strong statement on polymorph heterogeneity is questioned.

We agree. Given the low number of fibrils and difficulty in aligning these relatively thin and smooth fibrils, we have toned down our description of the A β fibril subtomogram averages in our revised manuscript, accordingly.

(v) The authors discuss that they observed parallel alignment of fibrils, which are all mostly vertically oriented. However, how does the missing wedge and therefore the anisotropic resolution in tomography affect this finding? Would horizontally aligned fibrils be detected at similar contouring thresholds?

The reviewer is correct that cryoET has anisotropic resolution. An expected consequence of this anisotropic resolution is that vertical filaments have higher contrast than horizontal tau filaments at the same threshold. Nonetheless, both horizontally and vertically aligned fibrils were clearly observed in our tomograms (for horizontal see **ED Fig. 10a** and **ED Movie 6**; for vertical see **Fig. 3** and **ED Movie 5**). Tau filaments were all parallel with each other, whether observed vertically or horizontally. In contrast, β -amyloid plaques contained regions in which fibrils were in a mesh (a mixture of both horizontal and vertical orientations). In the **Results** section of our revised manuscript we have included a citation of the figure showing horizontally arranged tau filaments to clarify this point.

(vi) Of great curiosity is the parallel arrangement of the fibrils. Can this be rationalized? does it follow a membrane close by?

We agree with the reviewer that the parallel arrangement of tau filaments is interesting. In some tomograms it is apparent that a cluster of filaments sits within the narrow neurite and the filament axis is parallel to the neurite axis. This could be rationalised as the confined space of the neurite influencing the arrangement of filaments. The **Discussion** section of our revised manuscript now includes these important points. Additionally, in three tau cluster tomograms the resolution of subtomogram averaging was sufficient to determine the polarity orientation of each filament within these parallel clusters, which we now describe and discuss in our revised manuscript.

Do the fibrils also have similar length or is this an artifact? Do the Ab(1-42) fibrils have a common origin going outward with branching?

The volume of tissue captured in each cryoET reconstruction was $1 \times 1 \times 0.1 \mu\text{m}$. Most tau filaments are expected to be much longer than this, extending several microns (Kidd, 1964; Yagishita et al., 1981). Consequently, in-tissue cryoET provided volumes sampling more than a thousand tau filaments but not their entire length and our cryoET dataset did not contain the ends/termini of tau filaments. Therefore, we could not measure their length or determine if tau filaments vary in length.

The volume of cryoET reconstructions of β -amyloid plaques showed the ends of some, but not all, fibrils. As with tau filaments, A β fibrils were longer than could be encompassed in cryosections or liftout lamella tomograms. Therefore, cryoET data are not suitable for addressing the question of whether or not A β fibrils have a common origin. To increase clarity, we have slightly amended the **Discussion** section of our revised manuscript from "could offer an explanation for the focal concentration of A β observed in β -amyloid plaque" to "could contribute to the high local concentration of A β that characterises β -amyloid plaques".

(vii) Use of twist rate to state polymorph difference: There are several reports that show that the twist rate has often nothing to do with a distinct polymorph. Most often fibril change the twist rate over length and Mezzenga demonstrated that the twist rate can be manipulated by salt only without request of another polymorph following polymer chemistry argumentation. Hence, while at this low resolution interesting to see, a different twist rate of the fibrils has very little information content unfortunately.

We agree with the reviewer that twist does not necessarily imply a different polymorph. We thank the reviewer for drawing our attention to the Mezzenga paper, which showed that salt concentration alters the helical twist apparent by AFM of in vitro prepared A β peptide. Such studies are fascinating but of course cannot determine what is the ultrastructural polymorph under these varying conditions. Additionally, ex vivo sarkosyl-insoluble purifications of tau filaments (in the same buffer) showed slight variations of helical twist (Fitzpatrick et al., 2017) within the same buffer. We have included a brief discussion of varying helical twist in **Discussion** section of our revised manuscript.

Our cryoET data did not provide any evidence of twist varying along the length of filaments in AD tissue, but helicity was in most subcellular locations highly similar within filaments of the same tau cluster. In our revised manuscript we now report that for a subset of tomograms that gave higher resolution averages we could determine the polarity orientation of each filament in the tissue, that in several locations appeared to be non-random.

Minor points:

(a) In the abstract, the authors claim to have "determined the structure" of amyloid and tau (Line 28). This is a misleading statement, as "determining the structure" usually refers to higher-resolving work where the secondary structure details are recognized. This wording should be toned down.

In our revised manuscript, we now report improved subtomogram averaging that reached sub-nanometre resolution for a cluster of tau filaments in a tissue cryosection tomogram, in which we can resolve the polypeptide chain and secondary structural details of the filament.

(b) The authors also claim that the tau fibril heterogeneity can be explained by the subcellular tissue environment (Line 37/38). However, this is not shown anywhere (only that there is a difference between tau fibril clusters that are spatially distributed).

We agree with the reviewer. The subcellular tissue environment is one of several possible explanations for how differences of structure are spatially distributed. We have amended our discussion of this point, accordingly.

(c) In line 85-90, the authors claim to have studied the 3D architecture of plaques and tangles. This wording should be toned down, as the authors have only studied a slim lamella from the tissue, not the entire plaques or tangles.

We agree and we have toned down this sentence, accordingly.

(d) Please chose one form of notation i.e. cryo-ET, cryo-FM.

Thank you for spotting this. We have used acronyms cryoET for cryo-electron tomography and cryoFM for cryo-fluorescence and harmonized throughout the text.

(e) The projection thickness of tomographic slices has not been defined, please add the related information.

Thank you for spotting this. All tomographic slices shown in figures are one voxel thick from bin4 tomographic reconstructions (9.52 Å voxel size). We have included this information in the **Methods** section entitled 'Preparation of cryoET figures' of our revised manuscript. We have also added the thickness of the cryosection and tissue lamellae to **ED Tables 1-5** and **7** of our revised manuscript.

(f) In line 396, please clarify that there are two sets of samples for each donor, frozen at -80C and fresh. The sentence reads as if the fresh sample was frozen.

All experiments were performed with fresh tissue samples that were first flash frozen in liquid nitrogen before being thawed, sliced in an NMDG-containing artificial cerebrospinal fluid, labelled with MX04, and high-pressure frozen. We described these samples as 'fresh' following the

convention of neuropathological brain banks throughout the world that describe tissues prepared as fresh-frozen in contrast to tissues that are first chemically fixed with paraformaldehyde before being frozen. These tissues are comparable to those used for the preparation of sarkosyl-extracted amyloid (e.g. (Fitzpatrick et al., 2017)), except that tissues were used with short (~6 h) post-mortem interval. In our revised manuscript we have amended the main text, **Methods**, and **Fig. 1d** to clarify this point.

We have also included a new tomographic data set to control for the effect of the post-mortem interval and the freeze-thaw step (see response to Referee 1). These data confirm that the fibril architecture and non-amyloid constituents (extracellular vesicles and droplets) were similar with or without a 6 h PMI and the freeze-thaw step. In our revised manuscript we describe these data in the **Results** and **Methods** sections.

(g) It is stated that the Ab(1-42) in vitro structure differs from the brain derived material. This is not the case albeit often stated. The solid state NMR structure is very similar to the type 1 structure having 50% the N-terminal residues folding a b-strand as in the case of the cryo EM structure and 50% not folded.

We agree that at the secondary structural level, solid state NMR structures of in vitro prepared β -amyloid fibrils (Colvin et al., 2016; Wälti et al., 2016) are somewhat similar to the type I ex vivo, sarkosyl-extracted β -amyloid. However, a careful comparison was reported by (Yang et al., 2022): “when examined at the single-residue level, none of the Ab42 filaments assembled in vitro displayed the same side chain orientations and contacts or the same inter-protofilament packing as that observed in type I and type II filaments”. We inspected these atomic models ourselves: alignment of the backbone C α atoms of the NMR versus cryoEM structures confirmed that these structures are significantly different (7q4b versus 5kk3: 4.2 Å C α RMSD and 7q4b versus 2nac: 4.9 Å).

(h) In Ex D.F.2-c cyan arrows are not easy to detect in cyan background. Also, in the same figure Tau threads labelled by open orange arrowheads are not easily seen.

Thank you for spotting this. We have amended the cyan annotations with a black border to improve clarity. We have increased the size of panels showing tau threads in ED Fig 2c (now **ED Fig. 2b** in our revised manuscript).

(i) In Ex D.F.2, please keep the same color code for the same feature at different figure panels if possible.

Thank you for spotting this. We have checked for consistency of arrowhead annotations: Closed cyan: amyloid plaque; Closed orange: tau tangle; open orange: tau thread.

(j) In Supp., in line 50 “Cyan arrowhead, β -amyloid plaque.” is repeated twice.

Thank you for spotting this. We have removed this duplicated description for the supplementary figures.

(k) In Ex D.F.3-b right panel, please remove open cyan arrowheads, as it wasn't explained in the legend what they are pointing to.

Thank you for spotting this. We have added a description of the open cyan arrowhead to the figure legend.

(l) In Ex D.F.3-b, how can it be confirmed that these unlabelled fibrils are truly amyloids and not neurofilaments or vimentin?

The main figures showed a typical tomogram of MX04-labelled β -amyloid plaques. Since MX04 labelling was only for 1 hour, MX04 was only apparent $\sim 30 \mu\text{m}$ deep into acute brain slices. In **ED Fig. 4b**, we show a tomogram collected from a location deeper in the tissue slice that lacked MX04 labelling. Even in the absence of MX04 amyloid label for this particular tomogram, several lines of evidence suggest it is highly unlikely this tomogram is of intermediate filaments, including neurofilaments or vimentin: 1) CryoET of vimentin (Eibauer et al., 2021; Goldie et al., 2007) indicates these filaments are $\geq 12.7 \text{ nm}$ diameter. Negative stain EM indicates neurofilaments have 10-15 nm diameter (Troncoso et al., 1990). The diameter of fibrils in our tomogram are 3-8 nm diameter, which is too thin to be intermediate filaments. 2) The architecture of the extracellular filaments we show is also highly similar to Mx04-labelled β -am-amyloid plaque tomograms (**Fig. 1h**) and those identified in our previous cryoET study of *App*^{NL-G-F} mice (Leistner et al., 2023). In our revised manuscript, we have amended the legend of **ED Fig.4b** to include the above points. We have also revised the **Methods** section entitled ‘Cryo-electron tomography’ of our revised manuscript to include a description of how locations were selected for cryoET data collection.

(n) In Ex D.F.4, for an easier read instead of many colorful arrowheads, abbreviated terms can be used. For example, “EV or ExV” for extracellular vesicles.

We chose to annotate tomographic slices with arrowheads and not text labels because it was important to ensure that the structural feature in the tomogram was precisely pinpointed within the tomographic slice. We considered adding both arrowheads and text labels, but we have

avoided this because the addition of both would obscure more of the tomographic slice in the figure, particularly when there are many features to annotate.

(m) In Ex D.F.7-b, CTE-1 data is not plotted on the graph.

Thank you for spotting the error in this figure legend, which we have now corrected in the revised manuscript.

(o) In Ex D.F.7-b, the helical twist was redefined by Scheres (Scheres S. H. W. (2020). *Acta crystallographica*. 76(Pt 2), 94–101.) as $4.75 \text{ \AA} \times 180 \text{ degree/crossover distance}$. For a 750 Angstroms crossover distance, helical twist can be estimated as -1.14 degrees , which then is per rung and not per nm. In the graph, twist is specified in degrees per nm, which is unusual.

We agree that estimating twist by degrees per layer was established by Scheres (Fitzpatrick et al., 2017) because helical averaging of cryoEM data was able to resolve the layers/rungs that are $\sim 4.7 \text{ \AA}$ within amyloids. However, it would be misleading for us to report twist in degrees per rung because the in-tissue tomographic tilt series reported here were collected with (2.38 \AA pixel size; 4.76 \AA Niqvist) that cannot resolve the layers/rungs of an amyloid fibril. Helical twist in our cryoET data was thus estimated by measuring degrees rotation within a fixed length of subtomogram averaged filaments in each cluster (degrees per nm). Using these measurements we also calculated the commonly understood 'crossover distance' (distance for 180 degrees twist = length of the fibril in nm to rotate 180 degrees). Both 'degrees per nm' and 'crossover distance' are plotted in **ED Fig 17a** (see left and right y-axis) in our revised manuscript, respectively.

(p) In ex. D.F.7.-c: How was classification done? PCA is a dimension-reduction method, after which one of several possible classification algorithms can be applied. K-means, HA, or others are possible examples. Which was used here, and how was it done?

We thank the reviewer for spotting this omission. K-means was used and we have added this important detail to the **Results** section of our revised manuscript.

(q) In Ex D.F.8., please add the thickness of the tomographic slices/projections at three positions.

We have added the tomographic slice thickness information (1 voxel thick) of a bin1 tomogram (2.38 \AA voxel size) that is **ED Fig. 11** in our revised manuscript.

(r) In Ex D.F.10, the resolution is not sufficient to allow significant docking experiments. The

wording in lines 202-205 should be toned down.

We agree. With our improved subtomogram averaging pipeline (Warp-PEET-Relion-M) we have obtained higher resolution averages that now replace ED Fig 10 (see **ED Fig 11** in our revised manuscript). We have also implemented a quantitative method for fitting that gives an absolute score of fit based a likelihood score (Read *et al.*, 2023).

(s) In Ex D.F.10-f, this polymorph does not fit any ex vivo model. This opens a range of potential interpretations for its nature and origin. Could you please discuss it?

In our revised manuscript we have now also used 'EM placement' to obtain a quantitative score of fitting, which provided an inconclusive scores for 3 out of 9 clusters (see **ED Fig. 11**). We have therefore limited our discussion of these three MX04-labelled tau filament structures because the resolution obtained by subtomogram averaging these amyloid fibrils did not reach a sufficient resolution to resolve the fold.

(t) In Ex D.F.14, green, brown and yellow rectangles are difficult to see.

Thank you for spotting this. We have amended the opacity of these rectangles to improve clarity.

References

Colvin, M.T., Silvers, R., Ni, Q.Z., Can, T.V., Sergeyev, I., Rosay, M., Donovan, K.J., Michael, B., Wall, J., Linse, S., Griffin, R.G., 2016. Atomic Resolution Structure of Monomorphic A β 42 Amyloid Fibrils. *J Am Chem Soc* 138, 9663–9674.
<https://doi.org/10.1021/jacs.6b05129>

Creekmore, B.C., Kixmoeller, K., Black, B.E., Lee, E.B., Chang, Y.-W., 2023. Native ultrastructure of fresh human brain vitrified directly from autopsy revealed by cryo-electron tomography with cryo-plasma focused ion beam milling. *bioRxiv* 2023.09.13.557623.
<https://doi.org/10.1101/2023.09.13.557623>

- Eibauer, M., Weber, M.S., Turgay, Y., Sivagurunathan, S., Goldman, R.D., Medalia, O., 2021. The molecular architecture of vimentin filaments. *bioRxiv* 2021.07.15.452584. <https://doi.org/10.1101/2021.07.15.452584>
- Falcon, B., Zhang, W., Schweighauser, M., Murzin, A.G., Vidal, R., Garringer, H.J., Ghetti, B., Scheres, S.H.W., Goedert, M., 2018. Tau filaments from multiple cases of sporadic and inherited Alzheimer's disease adopt a common fold. *Acta Neuropathol* 136, 699–708. <https://doi.org/10.1007/s00401-018-1914-z>
- Fitzpatrick, A.W.P., Falcon, B., He, S., Murzin, A.G., Murshudov, G., Garringer, H.J., Crowther, R.A., Ghetti, B., Goedert, M., Scheres, S.H.W., 2017. Cryo-EM structures of tau filaments from Alzheimer's disease. *Nature* 56, 1–18.
- Goldie, K.N., Wedig, T., Mitra, A.K., Aebi, U., Herrmann, H., Hoenger, A., 2007. Dissecting the 3-D structure of vimentin intermediate filaments by cryo-electron tomography. *Journal of Structural Biology* 158, 378–385.
- Kidd, M., 1964. Alzheimer's disease — an electron microscopical study. *Brain* 87, 307–320. <https://doi.org/10.1093/brain/87.2.307>
- Leistner, C., Wilkinson, M., Burgess, A., Lovatt, M., Goodbody, S., Xu, Y., Deuchars, S., Radford, S.E., Ranson, N.A., Frank, R.A.W., 2023. The in-tissue molecular architecture of β -amyloid pathology in the mammalian brain. *Nat Commun* 14, 2833. <https://doi.org/10.1038/s41467-023-38495-5>
- Millán, C., McCoy, A.J., Terwilliger, T.C., Read, R.J., 2023. Likelihood-based docking of models into cryo-EM maps. *Acta Crystallogr. Sect. D* 79, 281–289. <https://doi.org/10.1107/s2059798323001602>
- Mitchison, T.J., 1995. Evolution of a dynamic cytoskeleton. *Philosophical Transactions Royal Soc Lond Ser B Biological Sci* 349, 299–304. <https://doi.org/10.1098/rstb.1995.0117>
- Pollard, T.D., 1976. The role of actin in the temperature-dependent gelation and contraction of extracts of *Acanthamoeba*. *The Journal of Cell Biology* 68, 579–601.
- Read, R.J., Millán, C., McCoy, A.J., Terwilliger, T.C., 2023. Likelihood-based signal and noise analysis for docking of models into cryo-EM maps. *Acta Crystallogr. Sect. D* 79, 271–280. <https://doi.org/10.1107/s2059798323001596>
- Terry, R.D., Gonatas, N.K., Weiss, M., 1964. Ultrastructural studies in Alzheimer's presenile dementia. *Am J Pathology* 44, 269–97.
- Troncoso, J.C., March, J.L., Häner, M., Aebi, U., 1990. Effect of aluminum and other multivalent cations on neurofilaments in vitro: An electron microscopic study. *J. Struct. Biol.* 103, 2–12. [https://doi.org/10.1016/1047-8477\(90\)90080-v](https://doi.org/10.1016/1047-8477(90)90080-v)

Wälti, M.A., Ravotti, F., Arai, H., Glabe, C.G., Wall, J.S., Böckmann, A., Güntert, P., Meier, B.H., Riek, R., 2016. Atomic-resolution structure of a disease-relevant A β (1–42) amyloid fibril. *Proc National Acad Sci* 113, E4976–E4984. <https://doi.org/10.1073/pnas.1600749113>

Yagishita, S., Itoh, Y., Nan, W., Amano, N., 1981. Reappraisal of the fine structure of Alzheimer's neurofibrillary tangles. *Acta Neuropathol* 54, 239–246. <https://doi.org/10.1007/bf00687747>

Yang, Y., Arseni, D., Zhang, W., Huang, M., Lövestam, S., Schweighauser, M., Kotecha, A., Murzin, A.G., Peak-Chew, S.Y., Macdonald, J., Lavenir, I., Garringer, H.J., Gelpi, E., Newell, K.L., Kovacs, G.G., Vidal, R., Ghetti, B., Ryskeldi-Falcon, B., Scheres, S.H.W., Goedert, M., 2022. Cryo-EM structures of amyloid- β 42 filaments from human brains. *Science* 375, 167–172. <https://doi.org/10.1126/science.abm7285>

Reviewer Reports on the First Revision:

Referee #1 (Remarks to the Author) + attachment

This manuscript has improved considerably from the original version, with the addition of new data, including important controls of a healthy human brain and a mouse brain undergoing similar freeze-thaw procedures, a more conservative interpretation of the subtomogram averaging results, and much better subtomogram averaging for some of the PHFs. It is also satisfying to see that erroneous statements about the presence of CTE filaments in the AD brain have been removed. We believe there now exists a clear path to publication. However, given the extent of the changes made in this revision, we do have several important points to consider for a second round of revision.

P 3 “CryoET was performed on fresh, freeze-thawed post-mortem brain samples ...”

-> As mentioned in our original review, these samples are not fresh and should not be called that way. In the rebuttal the authors seem to distinguish between “fresh” (by which they mean their freeze-thawed) and “ultra-fresh” samples (which would be high-pressure frozen straight away). The use of the adverb “ultra” should be avoided and the adjective “fresh” should be reserved for the future case where AD brain is directly high-pressure frozen with minimal post-mortem delay. (Would one call a lettuce that was frozen, then thawed and served on a plate still fresh? And if a company selling this as fresh would refer to the original lettuce as “ultra-fresh”, wouldn’t we feel cheated?)

Therefore, all references to “fresh” or “fresh-frozen” sample should be removed from the paper. It is our strong opinion that the paper cannot be accepted without removal of the word “fresh” in all instances except the description of the original, unfrozen brain with minimal post-mortem delay. If the authors wish to use a subjective to describe their sample, they could use “freeze-thawed” instead.

P3. “from an AD donor and a non-demented donor (post-mortem delay 6 h 10 h and 5 h 45 m, respectively”

-> We don’t understand why there are 3 values for 2 donors. Please explain.

P3. “CryoEM of sarkosyl insoluble aggregates revealed the donor brain contained PHF and SF, but not TMEM106B fibrils (Extended Data Fig. 2b-f)”

-> This is incorrect. Several 2D class averages in Extended Data Fig 2C correspond to TMEM106B filaments. We attached a screenshot where we’ve labelled these in red. In addition, the extraction procedures used by the authors should have resulted in amyloid-beta filaments too. These seem to be abundant in the brain from the cryoET experiments. Why did the authors not see these in their 2D class averages? Were these perhaps not picked by the autopicker? Were they lost in the extraction/purification?

P4. “A tissue block was thawed...”

-> It would be good to mention the delay between thawing and high-pressure freezing in the main text, explicitly mentioning this in the context of the total post-mortem delay.

P5. "Extracellular droplets in β -amyloid plaques..."

-> How do you know these are droplets? They don't look like droplets at all to us. Also, how can droplets, which assumedly are liquid in nature, be cuboidal? These results have been inserted in the revised version, without mention of this the rebuttal. Making claims about cuboidal droplets would require more evidence, and as such it may be better to remove these claims from the current manuscript.

P5. "A cluster of this amyloid was also found..."

-> Above, the authors have said that they cannot exclude that open membrane sheets arise from the freeze-thaw procedure. This seems at odds with the statement "suggesting the extracellular location of this tau filament cluster and mitochondrion did not arise during sample preparation."

P6. LLG and real-space cross-correlation coefficients are not very useful indicators of the fit of atomic models in the subtomogram average maps. Instead, or at the very least in addition, curves of the Fourier shell correlation between map and model should be included to the Extended Data, and the resolution at which they drop below 0.5 should be reported in the main text. These values will also provide possibly more realistic estimates for the resolutions of the subtomogram averages.

P7. "These in situ cryoET data indicate that tau filaments form clusters"

-> This was already observed above. Are the authors trying to say that tau filaments of a specific type (e.g. PHF or SF) cluster together?

P7. "All in-tissue PHF subtomogram averages were 10 to 45% less twisted than that of previously reported ex vivo PHFs (79-181 nm versus 71 nm cross over distance of in situ PHF filaments versus ex vivo PHF, respectively)."

-> This is incorrect, and we apologize for not having brought this up in our initial review. The authors mistakenly assume that a single refined twist value in the cryo-EM single-particle reconstruction reflects the twist of all filaments used for that map. Instead, filaments with varying twists are averaged together in a single reconstruction, because real-space averaging is only performed using the central ~10-25% of the particle box (which will typically amount to less than 100 Angstroms), and differently twisted filaments will only lead to minor differences in density within that region. In fact, it has been reported that also sarkosyl-extracted filaments vary in twist, as explicitly stated by Fitzpatrick et al Nature 547, 185-190 (2017): "PHFs had a longitudinal spacing between crossovers of 650-800 Å and a width of about 150 Å at the widest part and 70 Å at the narrowest. SFs were about 100 Å wide with crossover distances ranging from 700 to 900 Å". It is not entirely clear to us which of the CS entries in Ext Data Figure 17 corresponds to the 8.7A PHF map, but all the LOL entries agree with the twist ranges reported by Fitzpatrick et al. Only subtomogram average maps that have sufficient resolution to confidently dock in PHFs and SFs should be used for estimating twist angles. Consequently, there is probably not sufficient information to conclude that in situ tau PHFs have different twists than those extracted from tissue using sarkosyl and this statement should be removed from the manuscript. Also the following statements in the Discussion should be toned down: "However, these maps exhibited a marked

variability in their helical twist compared to ex vivo, sarkosyl-insoluble tau filaments²⁴. The similarity of filaments within the same cluster compared to those in different clusters suggest this variation in helicity is spatially restricted within each local subcellular environment.”

P7. “PCA classified the majority of filaments by subcellular location (Extended Data Fig. 17b-c), supporting the idea that each tau cluster comprises structurally similar filaments”

-> This PCA is cumbersome, as the separation into clusters may pick up on systematic differences in the reconstructed densities from the different tilt series images. It would thus be better to remove this statement and the PCA analysis from the paper. Also, the statement “the spatial segregation of distinct tau filament structures within different cellular contexts from a single brain region of an individual post-mortem AD donor” in the discussion section needs toning down.

As editorial side comments:

It would be helpful to have a tracked changes version for the second round of review.

Figure legends are extremely verbose, the extended data figures are many and quite a few are larger than one page, and sometimes contain redundant information (e.g. Ext Data Figures 11 & 12). A lot of work remains to be done to make these conform to Nature’s formatting guidelines. As, consequently, the final text is expected to change considerably from this version to the final one, part of this work could already be performed in the second round of revision.

C

2D classification (all fibrils, 321,041 segs, 2x binned)

Referee #3 (Remarks to the Author):

The revised manuscript entitled “In situ cryo-electron tomography of β -amyloid and tau in post-mortem Alzheimer’s disease brain” is greatly improved. There is however the following important remaining point to be considered.

For the model in the map fits and the validation the following theoretical framework was used:

Millán, C., McCoy, A. J., Terwilliger, T. C. & Read, R. J. Likelihood-based docking of models 447 into cryo-EM maps. *Acta Crystallogr. Sect. D* 79, 281–289 (2023). 448 7.

Read, R. J., Millán, C., McCoy, A. J. & Terwilliger, T. C. Likelihood-based signal and noise 449 analysis for docking of models into cryo-EM maps. *Acta Crystallogr. Sect. D* 79, 271–280 450 (2023).

In both publications the LLG and CC score is applied to maps with a resolution of 1.7 Å to 8.46 Å and in one case the resolution is estimated by the authors to 9 Å to 11 Å. LLG scores with values larger than 60 are considered to be a good correlation between model and map, but they have sometimes scores larger than 10'000. The question is what is a good LLG score for the fibril system with its strong directionality along the z-axis?

The following calculations for the LLG & CC approach is proposed:

- (i) To generate a baseline how good the "model to map fit" approach is, the full single particle maps of tau PHF should be used, therefore it is suggested to calculate for each of the maps in Extended Figure 2d, Extended Figure 2e and Extended Figure 2f the LLG and CC scores and specify for each of the calculations the complex correlation, σ_A .
- (ii) Instead of using the full single particle fibril maps (Extended Figure 2d-f) only a plane of the map (containing one single protein layer) for the fit should be used, that the goodness of the fit is not biased by the helical symmetry.
- (iii) Point (i) & (ii) would give a baseline and then it is suggested to compare the results with tempy2.
- (iv) For all the maps from cryoET data, which ranges from 8.4 Å to 32 Å, calculate LLG and CC scores (Figure 3i-j, Figure 4i-k, Extended Data Figure 11 a-f [already done] and 12 a-d).
- (v) Similar to point (ii), calculate for all the maps from cryoET data (Figure 3i-j, Figure 4i-k, Extended Data Figure 11 a-f and 12 a-d), therefore use only a fibril plane of the thickness of a single protein layer.
- (vi) For the cryoET map with 8.7 Å (Figure 3j) calculate a map to model fit using tempy2

Minor points of considerations are the following:

(a) Line 241: "super-resolution cryoFM" is this a typo? Otherwise, it would be great to have short paragraph in method section or Si, which specify the instrument and settings used for the super-resolution cryoFM and the resolution limit in respect to standard cryoFM.

(b) Figure 1: Please add the values of the scale bar to the figures for better readability

(c) Figure 3j: In the overlay of model to map (cross section view) please use only the c-alpha trace of the backbone

(d) Extended Data Figure 11: In the overlay of model and map (iii) (cross section view) please use only the c-alpha trace of the backbone

(e) Extended Data Figure 11b: the LLG values are -77.5 and CC: 0.41, therefore no model should be fitted into the map

(f) Extended Data Figure 11f: the LLG values are 144.3 and CC: 0.47, therefore the model should be fitted into the map

Author Rebuttals to First Revision:

Responses to referees' comments

Referee #1 (Remarks to the Author) + attachment

This manuscript has improved considerably from the original version, with the addition of new data, including important controls of a healthy human brain and a mouse brain undergoing similar freeze-thaw procedures, a more conservative interpretation of the subtomogram averaging results, and much better subtomogram averaging for some of the PHFs. It is also satisfying to see that erroneous statements about the presence of CTE filaments in the AD brain have been removed. We believe there now exists a clear path to publication. However, given the extent of the changes made in this revision, we do have several important points to consider for a second round of revision.

We thank the referee for carefully reading our revised manuscript and for appreciating the strengths of our new data. Most comments were related to our choice of syntax in the text and typographical errors, for which we apologise. We have amended with tracked changes the second revision of the manuscript to address each of these points as described in our responses below.

P 3 “CryoET was performed on fresh, freeze-thawed post-mortem brain samples ...”

-> As mentioned in our original review, these samples are not fresh and should not be called that way. In the rebuttal the authors seem to distinguish between “fresh” (by which they mean their freeze-thawed) and “ultra-fresh” samples (which would be high-pressure frozen straight away). The use of the adverb “ultra” should be avoided and the adjective “fresh” should be reserved for the future case where AD brain is directly high-pressure frozen with minimal post-mortem delay. (Would one call a lettuce that was frozen, then thawed and served on a plate still fresh? And if a company selling this as fresh would refer to the original lettuce as “ultra-fresh”, wouldn't we feel cheated?)

Therefore, all references to “fresh” or “fresh-frozen” sample should be removed from the paper. It is our strong opinion that the paper cannot be accepted without removal of the word “fresh” in all instances except the description of the original, unfrozen brain with minimal post-mortem delay. If the authors wish to use a subjective to describe their sample, they could use “freeze-thawed” instead.

We agree with the reviewer and we have amended the manuscript accordingly.

P3. “from an AD donor and a non-demented donor (post-mortem delay 6 h 10 h and 5 h 45 m, respectively”

-> We don't understand why there are 3 values for 2 donors. Please explain.

We apologise for the confusion - it was caused by a typographical error that has been amended to: "from an AD donor and a non-demented donor "6 h 10 m and 5 h 45 m, respectively".

P3. "CryoEM of sarkosyl insoluble aggregates revealed the donor brain contained PHF and SF, but not TMEM106B fibrils (Extended Data Fig. 2b-f)"

-> This is incorrect. Several 2D class averages in Extended Data Fig 2C correspond to TMEM106B filaments. We attached a screenshot where we've labelled these in red. In addition, the extraction procedures used by the authors should have resulted in amyloid-beta filaments too. These seem to be abundant in the brain from the cryoET experiments. Why did the authors not see these in their 2D class averages? Where these perhaps not picked by the autopicker? Were they lost in the extraction/purification?

The reviewer is correct that there is a minor subset of particles in our cryoEM dataset of sarkosyl-extracted AD brain that likely correspond to TMEM106B on the basis of ~200 nm crossover distance (compared to PHF and SF that are on average less than half this crossover distance). We did attempt to obtain a reconstruction of the putative TMEM106B subset of particles but there were relatively few (2%), and we were not able to solve a TMEM106B structure. We then used language that was misleading because we were referring to 'structures solved' rather than the dataset as a whole. We thank the reviewer for drawing our attention to this and we have amended the text to "CryoEM of sarkosyl insoluble aggregates resolved PHF and SF in the AD donor brain".

The reviewer also suggests that amyloid- β fibrils should also have been present in the sarkosyl-extracted amyloid sample. We only observed a small number of A β fibrils relative to tau filaments from this sarkosyl-extracted preparation. To do so we established purification of tau filaments from 0.5 g AD donor tissue, which is roughly 10-fold less than reported previously, because we did not have access to greater amounts of tissue from this donor. We tested two protocols Falcon et al, 2018 and Schweighauser et al., 2023. The latter reported purification of tau from Tau P301S transgenic mice that lacked β -amyloid. In our hands the Schweighauser et al. protocol yielded the greatest amount of tau. We agree with the reviewer's suggestion that amyloid beta fibrils may have been lost during extraction/purification using the Schweighauser et al. protocol.

We performed cryoEM of sarkosyl extracts to confirm at high resolution the fold and ultrastructural polymorph of the tau filaments present in this particular donor, as requested by the reviewer after the initial submission of the manuscript. We hope the reviewer is satisfied that by obtaining a 3 Å resolution structure of tau PHF from this donor, and 2D class averages showing the SF ultrastructural polymorph that the donor is well-characterised for the in-tissue cryoET that is the focus of our study.

P4. "A tissue block was thawed..."

-> It would be good to mention the delay between thawing and high-pressure freezing in the main text, explicitly mentioning this in the context of the total post-mortem delay.

We agree and we have amended the text to include these details in the Methods section describing the time interval of each step of the protocol, accordingly. In short, once the tissue is thawed (5 min), 100 μm slices are prepared in chilled buffer (30 min) and labelled with methoxy X04 (1 h). Incubated in cryoprotectant (30 min) and high-pressure frozen. These steps add ~2 h to the post-mortem interval.

P5. "Extracellular droplets in β -amyloid plaques..."

-> How do you know these are droplets? They don't look like droplets at all to us. Also, how can droplets, which assumedly are liquid in nature, be cuboidal? These results have been inserted in the revised version, without mention of this the rebuttal. Making claims about cuboidal droplets would require more evidence, and as such it may be better to remove these claims from the current manuscript.

The reviewer is correct that we included a description of the shape of a subset of the non-amyloid constituents of amyloid plaques in the results section and we apologise for neglecting to describe this in the rebuttal (**ED Fig. 9**, now **ED Fig 7** in second revision). We agree that 'cuboidal droplet' is not correct. These non-amyloid in-tissue constituents are 27-200 nm diameter particles with regularly spaced striation that are comparable to lipid droplets, lipoprotein particles and also resemble ApoE and premelanosomal protein-associated caps (a non-pathological amyloid found in the retina) found attached to extracellular vesicles (previously reported by Van Niel/Raposo, 2015, DOI: [10.1016/j.celrep.2015.08.057](https://doi.org/10.1016/j.celrep.2015.08.057)). In the second revision of our manuscript, we have amended the description of these particles by removing the term 'droplet' and instead describing these non-amyloid constituents as "extracellular cuboidal particles".

P5. "A cluster of this amyloid was also found..."

-> Above, the authors have said that they cannot exclude that open membrane sheets arise from the freeze-thaw procedure. This seems at odds with the statement "suggesting the extracellular location of this tau filament cluster and mitochondrion did not arise during sample preparation."

We agree that sample preparation can result in the presence of small fragments of open membrane sheets and burst plasma membrane. Small membrane fragments were rarely (2/60) observed in mouse model tissue that has undergone a PMI and freeze-thawing similar to post-mortem brain (App^{NL-G-F} PMI-FT). However, we did not observe mitochondria in extracellular

locations in *App*^{NL-G-F} PMI-FT tissues nor in non-demented control tissues. These data suggest that sample preparation resulted in some limited damage to membranes, but not to the extent that whole mitochondria reside extracellularly. Since the tau filament cluster is extracellular together with a mitochondrion that completely lacks any evidence of a plasma membrane, it is not clear that the extracellular location of mitochondria and the tau filament cluster arose by sample preparation.

We hope the referee accepts this is a measured interpretation of the data. We have amended this description in our second revision reflecting this cautious interpretation: “A cluster of this amyloid was also found extracellularly, located next to a damaged mitochondrion outside of a myelinated axon, and without any evidence of an enclosing plasma membrane in the vicinity (**Fig. 2c-d**). In contrast, damaged mitochondria, frequently observed in non-demented control and *App*^{NL-G-F}-PMI-FT-HPF tomograms (**Extended Data Fig. 6** and **Supplementary Data Table 3 & 7**), were all completely or partially enclosed by plasma membrane. Nonetheless, we could not determine definitively whether or not the extracellular location of this tau filament cluster and mitochondrion was a consequence of sample preparation.”

P6. LLG and real-space cross-correlation coefficients are not very useful indicators of the fit of atomic models in the subtomogram average maps. Instead, or at the very least in addition, curves of the Fourier shell correlation between map and model should be included to the Extended Data, and the resolution at which they drop below 0.5 should be reported in the main text. These values will also provide possibly more realistic estimates for the resolutions of the subtomogram averages.

We noticed that in ED Fig. 11 of the first revision of our manuscript the order of the LLG scores detailed in the figure legend did not match the order of the figure panels. We apologise for this error. In the second revision of our manuscript, we have corrected this (now **ED Fig. 9**) and LLG scores are listed in **Supplementary Data Table 5** in the second revision of our manuscript. We hope the reviewer now agrees that the LLG scores are consistent with the observed quality of the maps and provide a useful quantitative indicator of the fit of atomic models into the map. See also our response to referee #3 below.

We have additionally followed the reviewer’s suggestion of performing model versus map Fourier shell correlation. This was carried out using the method implemented in the Phenix mtriage program to compare the atomic model (PHF or SF) with subtomogram averaged maps, avoiding artefacts at the edges of the averaged map by restricting the comparison to a central sphere with a radius of 80 Å. To compare with subtomogram average maps, we also prepared model-map FSC for the high resolution, cryoEM structure of PHF (Fitzpatrick et al., 2017; PDB: 5o3l and map: EMD-3741) and our single-particle cryoEM map (**ED Fig. 2i** in second revision). For each plot we have also included an annotation of the gold-standard FSC resolution estimate in red (HM FSC):

This provides a graphical representation that is consistent with the LLG score obtained using EM placement (**Supplementary Data Table 5**). The above plots probably do not provide additional information to the LLG score and do not provide a realistic estimate of the resolution of the map. For example, the subtomogram average map that resolved the separation of β -sheets to 8.7 Å resolution in a single tomogram (**Fig. 3j**, now **Fig. 3h** in second revision) shows a cryoEM PHF model-map FSC above 0.5 to only 19 Å (top right panel). This may be in part because the cryoEM PHF model and subtomogram average maps relate to slightly different structures, including different twist (**ED Figure 13** in second revision, see also response to comment below) and an additional shell of density (as was also observed in unsharpened cryoEM maps in Fitzpatrick et al., 2017) that is not occupied by the model in this part of the map. Therefore, we propose not including these plots in our revised manuscript but if the reviewer would prefer that they are included we can add them to **ED Fig. 8 & 9**.

P7. “These in situ cryoET data indicate that tau filaments form clusters”

-> This was already observed above. Are the authors trying to say that tau filaments of a specific type (e.g. PHF or SF) cluster together?

We apologise this sentence was unclear. This summary sentence was intended to show that even when in close proximity, tau clusters can have distinct characteristics whereby a tau cluster comprised only of SFs is situated 1 μm apart from a mixed PHF/SF cluster in a neighbouring subcellular compartment. We have amended this sentence in the second revised manuscript to clarify this point: “These *in situ* cryoET data highlight the co-existence of multiple distinct ensembles of tau filaments (a mixed cluster of PHF/SF compared to an SF only cluster) organized within two neighbouring microscopic regions of pathology.”

P7. “All in-tissue PHF subtomogram averages were 10 to 45% less twisted than that of previously reported *ex vivo* PHFs (79-181 nm versus 71 nm cross over distance of *in situ* PHF filaments versus *ex vivo* PHF, respectively).”

-> This is incorrect, and we apologize for not having brought this up in our initial review. The authors mistakenly assume that a single refined twist value in the cryo-EM single-particle reconstruction reflects the twist of all filaments used for that map. Instead, filaments with varying twists are averaged together in a single reconstruction, because real-space averaging is only performed using the central ~10-25% of the particle box (which will typically amount to less than 100 Angstroms), and differently twisted filaments will only lead to minor differences in density within that region. In fact, it has been reported that also sarkosyl-extracted filaments vary in twist, as explicitly stated by Fitzpatrick et al Nature 547, 185–190 (2017): “PHFs had a longitudinal spacing between crossovers of 650–800 Å and a width of about 150 Å at the widest part and 70 Å at the narrowest. SFs were about 100 Å wide with crossover distances ranging from 700 to 900 Å”. It is not entirely clear to us which of the CS entries in Ext Data Figure 17 corresponds to the 8.7Å PHF map, but all the LOL entries agree with the twist ranges reported by Fitzpatrick et al. Only subtomogram average maps that have sufficient resolution to confidently dock in PHFs and SFs should be used for estimating twist angles. Consequently, there is probably not sufficient information to conclude that *in situ* tau PHFs have different twists than those extracted from tissue using sarkosyl and this statement should be removed from the manuscript.

We agree with the reviewer that the previously reported Fitzpatrick et al single-particle cryoEM of tau encompasses the structure of tau filaments with variable twist and that negative stain EM in Fitzpatrick et al measured a range of tau filament twists in the sample (65-80 nm and 79-90 nm crossover distance for PHF and SF, respectively). We have revised **ED Fig. 17** (now **ED Fig. 13** in the second revision) to include this range of crossover distances, as requested.

We agree that the twist of the two LOL (liftout lamella) clusters of tau filaments are within the range of *ex vivo* sarkosyl-extracted tau filaments. We also agree with the reviewer that it is safest to consider only the twist of subtomogram averages that have sufficient resolution to confidently dock PHFs and SFs. Three of the four clusters of tau filaments that could be resolved as PHFs in

tissue cryo-sections had a helical twist that was less than the range from cryoEM and negative stain in Fitzpatrick et al (see Supplementary Data Table 5), including the 8.7 Å PHF (Fig. 3h) with a 98.4 nm crossover distance. These data support the conclusion that in situ PHFs have a range of twists different to the range of twists reported for sarkosyl-extracted tau. We have amended this sentence to “Most in-tissue PHF subtomogram averages were 19 to 38% less twisted than that of previously reported *ex vivo* PHFs (79-129 nm cross over distance for filaments *in situ* versus 65-80 nm of *ex vivo* purified PHF (Fitzpatrick et al., (2017), respectively)”.

We apologise that the crossover distances shown in **ED Fig. 17** in our first revision could not be straightforwardly related to the 8.7 Å subtomogram average map in **Fig. 3j** (now **Fig. 3h**). These data were summarised in **Extended Data Table 5** (now **Supplementary Data Table 5** in the second revision of our manuscript). To clarify this further we have now also included a reference to the figures showing the maps in the graph labels on the x-axis of the helical twist graph (now **ED Fig. 13a** in the second revision of our manuscript).

Also the following statements in the Discussion should be toned down: “However, these maps exhibited a marked variability in their helical twist compared to *ex vivo*, sarkosyl-insoluble tau filaments²⁴. The similarity of filaments within the same cluster compared to those in different clusters suggest this variation in helicity is spatially restricted within each local subcellular environment.”

We agree and have toned down this statement, accordingly: “Half the tau filament cluster subtomogram average maps accommodated the atomic model of PHF (Fitzpatrick et al., 2017), most of which exhibited a decrease in their helical twist compared to *ex vivo*, sarkosyl-insoluble PHFs (Fitzpatrick et al., 2017). Since PHFs within most clusters were similar to each other, but were different between clusters, we suggest that this variability is spatially restricted or may be organised by subcellular location.”

P7. “PCA classified the majority of filaments by subcellular location (Extended Data Fig. 17b-c), supporting the idea that each tau cluster comprises structurally similar filaments”

-> This PCA is cumbersome, as the separation into clusters may pick up on systematic differences in the reconstructed densities from the different tilt series images. It would thus be better to remove this statement and the PCA analysis from the paper. Also, the statement “the spatial segregation of distinct tau filament structures within different cellular contexts from a single brain region of an individual post-mortem AD donor” in the discussion section needs toning down.

We agree with the reviewer and we have removed the PCA analysis from the second revision of our manuscript. We have also toned down this statement, as requested: “These in situ structures

of β -amyloid plaque and tau pathology revealed the heterogeneity of A β fibrils, the location-specific variability in helical twist and polarity orientation of tau filaments within different cellular contexts from a single brain region of an individual post-mortem AD donor.”

As editorial side comments:

It would be helpful to have a tracked changes version for the second round of review.

Figure legends are extremely verbose, the extended data figures are many and quite a few are larger than one page, and sometimes contain redundant information (e.g. Ext Data Figures 11 & 12). A lot of work remains to be done to make these conform to Nature’s formatting guidelines. As, consequently, the final text is expected to change considerably from this version to the final one, part of this work could already be performed in the second round of revision.

We apologise for not including tracked changes in the first revision of our manuscript.

In our second revision, we have amended with tracked changes. We have also amended figures ED figures, shortened figure legends, and removed a few references to *Nature* formatting guidelines.

Referee #3 (Remarks to the Author):

The revised manuscript entitled “In situ cryo-electron tomography of β -amyloid and tau in post-mortem Alzheimer’s disease brain” is greatly improved. There is however the following important remaining point to be considered.

We thank the referee for recognising that our manuscript is greatly improved. The reviewer raises an important point related to quantification of fitting models into subtomogram averaged maps reported in our manuscript that we address below. As described in response to referee #1 above, the figure legend to an ED Fig. 11 contained an error: the order of the LLG scores detailed in the figure legend did not match the order of the figure panels. We apologise for this error and we have corrected this (now **ED Fig. 9**) in the second revision of our manuscript and LLG scores are now detailed in **Supplementary Data Table 5**. The LLG scores are in accordance with the resolution and apparent quality of these maps. We hope that the referee now agrees that the LLG scores provide a useful measure of the fit of an atomic model in subtomogram averaged maps. We have also responded to each point of the suggested calculations below.

For the model in the map fits and the validation the following theoretical framework was used:

Millán, C., McCoy, A. J., Terwilliger, T. C. & Read, R. J. Likelihood-based docking of models

447 into cryo-EM maps. *Acta Crystallogr. Sect. D* 79, 281–289 (2023). 448 7.

Read, R. J., Millán, C., McCoy, A. J. & Terwilliger, T. C. Likelihood-based signal and noise 449 analysis for docking of models into cryo-EM maps. *Acta Crystallogr. Sect. D* 79, 271–280 450 (2023).

In both publications the LLG and CC score is applied to maps with a resolution of 1.7 Å to 8.46 Å and in one case the resolution is estimated by the authors to 9 Å to 11 Å. LLG scores with values larger than 60 are considered to be a good correlation between model and map, but they have sometimes scores larger than 10'000. The question is what is a good LLG score for the fibril system with its strong directionality along the z-axis?

Experience with LLG scores in both crystallographic molecular replacement and single-particle cryo-EM docking supports the idea that the numerical value of the LLG score provides a measure of the confidence in the result, regardless of overall resolution or anisotropy in the signal-to-noise.

The theoretical basis for this comes from understanding the LLG score as a measure of how much of the information gained by performing the experiment has been explained by the rigid-body placement of the model. Different trade-offs between data quality and quantity with model quality can yield the same degree of certainty. These considerations have now been explained in the Methods section of the second revision of the manuscript:

“When running EM placement, the map resolution was set to the value determined using the 0.143 half-map FSC threshold. Theory and experience with single-particle cryoEM data suggest that LLG scores of 60 or greater indicate a non-random fit, with higher values being observed for more accurate models and higher resolution maps (Read et al., 2023). The LLG score is related to how much of the information content of the map is explained by the model. Because the total information content of the portion of the map covered by the model depends both on its local (possibly anisotropic) resolution and its volume, an LLG score above the threshold can be reached by any combination of map quality with model size and quality. The confidence threshold itself (LLG 60) is not expected to vary with map resolution or issues arising from preferential orientations (Read et al., 2023).”

The following calculations for the LLG & CC approach is proposed:

(i) To generate a baseline how good the "model to map fit" approach is, the full single particle maps of tau PHF should be used, therefore it is suggested to calculate for each of the maps in Extended Figure 2d, Extended Figure 2e and Extended Figure 2f the LLG and CC scores and specify for each of the calculations the complex correlation, σ_A .

(ii) Instead of using the full single particle fibril maps (Extended Figure 2d-f) only a plane of the map (containing one single protein layer) for the fit should be used, that the goodness of the fit is not biased by the helical symmetry.

(iii) Point (i) & (ii) would give a baseline and then it is suggested to compare the results with tempy2.

(iv) For all the maps from cryoET data, which ranges from 8.4 Å to 32 Å, calculate LLG and CC scores (Figure 3i-j, Figure 4i-k, Extended Data Figure 11 a-f [already done] and 12 a-d).

(v) Similar to point (ii), calculate for all the maps from cryoET data (Figure 3i-j, Figure 4i-k, Extended Data Figure 11 a-f and 12 a-d), therefore use only a fibril plane of the thickness of a single protein layer.

(vi) For the cryoET map with 8.7 Å (Figure 3j) calculate a map to model fit using tempy2

As requested, we have performed EM placement fitting of the atomic model of ex vivo purified tau PHF into the final refined single-particle cryoEM map, as requested (in point 'i'). This gave scores of 11,889 LLG and 0.66 CC reflecting the higher information content of the 3 Å resolution cryoEM map, which we have included in the Methods section describing the LLG scoring.

This gave a baseline to compare with the LLG scores we obtained fitting the atomic model of PHF and SF into subtomogram averaged maps, in which we obtained a highest score for the map in Fig. 3j (now **Fig. 3h** in revised manuscript: LLG 944.6 CC 0.56. Note, the cross-correlation is not as reliable as LLG (see response below) and as described by Read et al. (2023), LLG scores above 60 are non-random.

See also CCPEM symposium 2023 demonstration of EM placement:

https://ukri.zoom.us/rec/play/D3eDYSzV_ciXnT_HzBHjaY7iayl-CrONsEjhKII64S_9f0r94QEjccBklQ2KhYFEI_0s0Itl3mRZEcMA.lqKoPKD3ieRIMkbq?canPlayFromShare=true&from=share_recording_detail&startTime=1682521917000&componentName=rec-play&originRequestUrl=https%3A%2F%2F

Reassuringly, all these LLG scores were consistent by providing a fitting score that was in accordance with the difference in the resolution of the single-particle cryoEM map and

subtomogram average maps (3 Å and 8.7-32 Å resolution for single-particle and subtomogram average maps, respectively).

We thank the reviewer for also suggesting using maps and models corresponding to a single layer of the filament (point 'ii) and comparing LLG to an alternative model to map scoring tool (tempy2, point 'vi)'). We have investigated this but have encountered several concerns with this approach:

i) Obtaining a single layer from subtomogram average maps is problematic because even the highest resolution subtomogram average map (8.7 Å resolution, Fig. 3j, now **Fig. 3h**) does not resolve the separation between layers that are spaced ~4.7 Å apart with 2_1 symmetry.

ii) The reviewer is correct that symmetry could be a bias if this parameter is unknown. However, with subtomogram averages and cryoEM, the helical symmetry is directly estimated and therefore this is not an unknown parameter.

ii) Achieving an LLG score high enough to give confidence in the fit requires a sufficient volume of the map to be explained by the model. The LLG scores obtained by fitting to maps covering 15 layers are sufficient in the better cases (~70.5 Å length of filament, see response to referee #1 on LLG above), but any scores obtained from a map as small as a single layer would be inconclusive.

iii) The scores given by tempy2 are useful for giving relative scores for the fit of a number of models to a map (Joseph *et al.* (2017), DOI: [10.1016/j.jsb.2017.05.007](https://doi.org/10.1016/j.jsb.2017.05.007)), however we favoured the likelihood-based approach. A particular advantage of scoring atomic model fit with the EM placement program is that the LLG score is absolute, allowing not only for the comparison of fit of multiple different models to EM maps, as requested by the reviewer in the first round of reviews of the manuscript, but also for judging confidence in the docking result.

We hope the referee is satisfied that the amendment to resolve the fundamental error in the figure showing EM placement of subtomogram averages (ED Fig. 9 in second revision of the manuscript) that now correctly shows the reliability of EM placement scoring of model to map fit and providing a reference using EM placement of PHF into the single-particle cryoEM map is sufficient.

Minor points of considerations are the following:

(a) Line 241: "super-resolution cryoFM" is this a typo? Otherwise, it would be great to have short paragraph in method section or Si, which specify the instrument and settings used for the super-resolution cryoFM and the resolution limit in respect to standard cryoFM.

We agree and have removed “super-resolution” from the revised manuscript.

(b) Figure 1: Please add the values of the scale bar to the figures for better readability

We agree this would be beneficial. We suspect however that this is not permitted according to Nature formatting guide. If the editor permits this we will add scale bar values to all figures.

(c) Figure 3j: In the overlay of model to map (cross section view) please use only the c-alpha trace of the backbone

We have amended Fig. 3j (now Fig3h), Fig. 4i & k (now Fig. 4h and j) and ED Fig 9, accordingly.

(d) Extended Data Figure 11: In the overlay of model and map (iii) (cross section view) please use only the c-alpha trace of the backbone

We have amended ED Fig. 11 (now ED Fig.9) accordingly.

(e) Extended Data Figure 11b: the LLG values are -77.5 and CC: 0.41, therefore no model should be fitted into the map

We agree and this was our intention. However, as we responded above, in the first revision of our manuscript we made a typographical error in which LLG scores in the figure legend did not match the order of the figure panels. We apologise for this error and we have amended this in the second revision of our manuscript (now **ED Fig. 9**), accordingly.

(f) Extended Data Figure 11f: the LLG values are 144.3 and CC: 0.47, therefore the model should be fitted into the map

We agree - see response to ‘e’ above.

Reviewer Reports on the Second Revision:

Referees' comments:

Referee #1 (Remarks to the Author):

This is OK now. It is good to see this paper has improved a lot since its original submission. I look forward to seeing it in print.

REVIEWER #1 on comments REVIEWER #3

Reviewer #3 raises a highly specific question: "what is a good LLG score for the fibril system with its strong directionality along the z-axis?"

I agree with the authors that this question becomes less relevant (and more difficult to address) at the resolutions of the sub-tomogram averages, where individual beta-strands are not separated in the maps.

Therefore, the multi-step procedure proposed by reviewer #3 is not so useful. In that sense, this specific suggestion can be ignored in this second review round.

There is however a more relevant underlying problem that I suspect the reviewer is trying to bring up with their comment, which is one that I also pointed out on page 3 of the rebuttal PDF: "LLG and real-space cross-correlation coefficients are not very useful indicators of the fit of atomic models in the subtomogram average maps". I suspect that the authors resorted to these relatively new, non-standard measures because well-established procedures to measure the fit of atomic models in cryo-EM maps like the FSC map-vs-model did not give them the beautiful results they would have liked. You can see this from their figure on page 4 in the rebuttal PDF. However, I do think that these not-so-beautiful FSC curves are actually a good representation of the quality of the various sub-tomogram averages. Although these averages are acceptable for the conclusions drawn in this paper, I expect sub-tomogram averages of amyloid filaments to become much better in the coming years, with improvements in cryo-EM sample preparation, hardware and software. Therefore, the FSC curves of model-vs-map will also become to behave much more than expected in the future.

I think the authors have made this paper a lot better than it was originally and, importantly, that the conclusions drawn are not dependent on the particular values of the LLGs, which are in dispute here. Therefore, I would suggest to move forward with publication. In order to settle this last outstanding point, I would take up the authors on their offer to include the FSC curve figure on page 4 of the rebuttal PDF into the extended data. In the Methods section, they could refer to these figures and also explicitly mention the reservation of the reviewer: that what is a good LLG score for the fibril system with its strong directionality along the z-axis remains unclear.

REVIEWER #3

We would like to thank the authors to answer most of the questions raised in last round. If the LLG/CC approach works it will be a valuable tool to answer biological questions on a cellular level, which would give cryo-ET a leap in understanding larger cellular organisation on a high resolution level.

It would therefore be valuable for the approach, especially for fibrils, to write a detailed paragraph to explain the guidelines to be used. It should contain:

- 1) Minimal length of fibril layers for the model to be reliable (70.5 Å length of filament), is there a mass or size limitation (Point 6)?
- 2) How large the map has to be in respect to the model (It is assumed to be the same length)
- 3) What sort of map has to be used for the fit; a refined map or a post processed

4) It is suggested to put the LLG fit for the 3 Å map of PHF a more prominent in the manuscript within the main part of the manuscript (somewhere around line 222):

At the moment it is only at the end of the method section:

As requested, we have performed EM placement fitting of the atomic model of ex vivo purified tau PHF into the final refined single-particle cryoEM map, as requested (in point 'i'). This gave scores of 11,889 LLG and 0.66 CC reflecting the higher information content of the 3 Å resolution cryoEM map, which we have included in the Methods section describing the LLG scoring.

5) It appears that a single layer is not suitable for the cryo-ET data, since there is no evidence that cryo-ET data has a 21 symmetry due to the fact the resolution is 8.7 Å.

i) Obtaining a single layer from subtomogram average maps is problematic because even the highest resolution subtomogram average map (8.7 Å resolution, Fig. 3j, now Fig. 3h) does not resolve the separation between layers that are spaced ~4.7 Å apart with 21 symmetry.

6) In the following statement a minimal amount of layers are proposed to have a confident fit:

ii) Achieving an LLG score high enough to give confidence in the fit requires a sufficient volume of the map to be explained by the model. The LLG scores obtained by fitting to maps covering 15 layers are sufficient in the better cases (~70.5 Å length of filament, see response to referee #1 on LLG above), but any scores obtained from a map as small as a single layer would be inconclusive.

Are these size limitation (15 layers of filaments) as well crucial for a PHF 3 Å map, or is there a possibility to perform a single layer model fit?

7). It is stated that not a single cryo-ET data were used for the LLG/CC approach, only cryo-EM data sets: EMD-11657, EMD-2984, EMD-6714, EMD-12654, EMD-28172, EMD-28172, EMD-25375,

From the technical point there is one large difference, that cryo-EM material is highly purified and the target protein is verified by mass spectrometry (LC-MS) and Western blots (Antibody).

On the other hand cryo-ET embodies a complete cellular environment with an organism specific proteome. Choosing only two models with a helical rise of 4.7 (2.37) and helical twist of -1 (179.5), gives only results, which of the two model might fit better.

Sub. Table 5:

LLG: PHF (5o3l) LLG: SF (5o3t) CC: PHF (5o3l) CC: SF (5o3t)

159.5 -64.4 0.46 0.25

944.6 234.6 0.56 0.40

Based on the LLG score both models PHF/SF are a valid fit. The SF score is lower, but based on the theory both fits should be considered. In the manuscript it is requested to precise the point that only 2 models were used for the fit. Especially going beyond 20 Å resolution and the knowledge that most disease relevant fibrils have a helical rise of 4.7 and helical twist of -1 (writing this the reviewer is aware of the colocalisation with CLEM but the fluorescence resolutions is not better than 200-300 nm.)

Further write down in the sub. table 5: the relevant points from 1-3 for each of the fits.

Minor:

1). Good practise is to use the `relion_helix_inimodel2d` as done by the authors. One should refine first with a C1 symmetry with helical rise 4.7 and twist -1, until a clear separation in 21 symmetry or C2 is observed. Otherwise the refinement can be biased by the applied a symmetry too early. It is assumee that the author followed this general procedure.

Line 693:

The helical rise was set to 2.4 Å for the PHF subset and 4.8 Å for the SF subsets, based on known structural data.

2). Sub. Table 5: Estimated Defocus (µm), Typo should be in [Å].

Author Rebuttals to Second Revision:

Referee #1 (Remarks to the Author):

This is OK now. It is good to see this paper has improved a lot since its original submission. I look forward to seeing it in print.

REVIEWER #1 on comments REVIEWER #3

Reviewer #3 raises a highly specific question: "what is a good LLG score for the fibril system with its strong directionality along the z-axis?"

I agree with the authors that this question becomes less relevant (and more difficult to address) at the resolutions of the sub-tomogram averages, where individual beta-strands are not separated in the maps.

Therefore, the multi-step procedure proposed by reviewer #3 is not so useful. In that sense, this specific suggestion can be ignored in this second review round.

There is however a more relevant underlying problem that I suspect the reviewer is trying to bring up with their comment, which is one that I also pointed out on page 3 of the rebuttal PDF: "LLG and real-space cross-correlation coefficients are not very useful indicators of the fit of atomic models in the subtomogram average maps". I suspect that the authors resorted to these relatively new, non-standard measures because well-established procedures to measure the fit of atomic models in cryo-EM maps like the FSC map-vs-model did not give them the beautiful results they would have liked. You can see this from their figure on page 4 in the rebuttal PDF. However, I do think that these not-so-beautiful FSC curves are actually a good representation of the quality of the various sub-tomogram averages. Although these averages are acceptable for the conclusions drawn in this paper, I expect sub-tomogram averages of amyloid filaments to become much better in the coming years, with improvements in cryo-EM sample preparation, hardware and software. Therefore, the FSC curves of model-vs-map will also become to behave much more than expected in the future.

I think the authors have made this paper a lot better than it was originally and, importantly, that the conclusions drawn are not dependent on the particular values of the LLGs, which are in dispute here. Therefore, I would suggest to move forward with publication. In order to settle this last outstanding point, I would take up the authors on their offer to include the FSC curve figure on page 4 of the rebuttal PDF into the extended data. In the Methods section, they could refer to these figures and also explicitly mention the reservation of the reviewer: that what is a good LLG score for the fibril system with its strong directionality along the z-axis remains unclear.

We used LLG scores (provided by `em_placement`) because this most straightforwardly addressed reviewer #3's request following the initial submission of our manuscript: "The mapping of the atomic resolution structure onto the EM tomograms: The mapping is of qualitative nature. Can one quantify the mapping?". `em_placement` quantified the mapping, in which LLG > 60 indicates better than random regardless of resolution or anisotropy, but uncertainty along the bad direction will be greater. Specifically, at a resolution where layers merge together, there will be uncertainty about the exact placement of the layers. Although `em_placement` is relatively new, the maximum-likelihood scoring in `em_placement` is conceptually similar to the approach for scoring SAD phasing and molecular replacement of X-ray crystallographic datasets by Phaser that has been widely used for several decades (doi: 10.1107/S0021889807021206 and 10.1107/S0907444910051371).

`em_placement` has the particular advantage that it provides an absolute score and is straightforward to implement (within ChimeraX, as described in a recent Biorxiv preprint: <https://doi.org/10.1101/2024.05.16.594509>). We expect this approach will be useful for cryoET more broadly by providing absolute scores to aid objective identification of constituents within a map, particularly with advances in hardware and software in the future.

We have amended the methods as requested and placed model-map FSCs as an Extended Data Fig. 14, referred to in the methods section, as suggested.

REVIEWER #3

We would like to thank the authors to answer most of the questions raised in last round. If the LLG/CC approach works it will be a valuable tool to answer biological questions on a cellular level, which would give cryo-ET a leap in understanding larger cellular organisation on a high resolution level.

It would therefore be valuable for the approach, especially for fibrils, to write a detailed paragraph to explain the guidelines to be used. It should contain:

1) Minimal length of fibril layers for the model to be reliable (70.5 Å length of filament), is there a mass or size limitation (Point 6)?

The key limitation is the amount of information that can be explained by the model. At lower resolution, a larger map volume has to be explained by a larger model. More intuitively, the volume has to have sufficient features to be recognisable and distinguishable from other objects.

2) How large the map has to be in respect to the model (It is assumed to be the same length)

The map preferably extends further than the model (to allow for smooth masking of the region being evaluated). Everything inside the spherical volume (with a radius sufficient to contain the whole model) is considered in the fit.

3) What sort of map has to be used for the fit; a refined map or a post processed

Preferably a pair of independent half-maps. If “post-processed” means denoised, density-modified or something like that, the method is not optimised for that scenario.

4) It is suggested to put the LLG fit for the 3 Å map of PHF a more prominent in the manuscript within the main part of the manuscript (somewhere around line 222):

At the moment it is only at the end of the method section:

As requested, we have performed EM placement fitting of the atomic model of ex vivo purified tau PHF into the final refined single-particle cryoEM map, as requested (in point ‘i’). This gave scores of 11,889 LLG and 0.66 CC reflecting the higher information content of the 3 Å resolution cryoEM map, which we have included in the Methods section describing the LLG scoring.

5) It appears that a single layer is not suitable for the cryo-ET data, since there is no evidence that cryo-ET data has a 21 symmetry due to the fact the resolution is 8.7 Å.

i) Obtaining a single layer from subtomogram average maps is problematic because even the highest resolution subtomogram average map (8.7 Å resolution, Fig. 3j, now Fig. 3h) does not resolve the separation between layers that are spaced ~4.7 Å apart with 21 symmetry.

6) In the following statement a minimal amount of layers are proposed to have a confident fit:

ii) Achieving an LLG score high enough to give confidence in the fit requires a sufficient volume of the map to be explained by the model. The LLG scores obtained by fitting to maps covering 15 layers are sufficient in the better cases (~70.5 Å length of filament, see response to referee #1 on LLG above), but any scores obtained from a map as small as a single layer would be inconclusive.

Are these size limitation (15 layers of filaments) as well crucial for a PHF 3Å map, or is it there a possibility to perform a single layer model fit?

Given sufficient resolution, essentially any size of model could be fit. The LLG value achieved is a good indicator for whether the volume was sufficient.

7). In is stated that not a single cryo-ET data were used for the LLG/CC approach, only cryo-EM data sets: EMD-11657, EMD-2984, EMD-6714, EMD-12654, EMD-28172, EMD-28172, EMD-25375,

From the technical point there is one large difference, that cryo-EM material is highly purified and the target protein is verified by mass spectrometry (LC-MS) and Wester blots (Antibody).

On the other hand cryo-ET embodies a complete cellular environment with an organism specific proteome. Choosing only two models with a helical rise of 4.7 (2.37) and helical twist of -1 (179.5), gives only results, which of the two model might fit better.

Sub. Table 5:

LLG: PHF (5o3l) LLG: SF (5o3t) CC: PHF (5o3l) CC: SF (5o3t)

159.5 -64.4 0.46 0.25

944.6 234.6 0.56 0.40

Based on the LLG score both models PHF/SF are a valid fit. The SF score is lower, but based on the theory both fits should be considered. In the manuscript it is requested to precise the point that only 2 models were used for the fit. Especially going beyond 20 Å resolution and the knowledge that most disease relevant fibrils have a helical rise of 4.7 and helical twist of -1 (writing this the reviewer is aware of the colocalisation with CLEM but the fluorescence resolutions is not better than 200-300 nm.)

The theory and experience suggest that an LLG of 60 or more indicates a non-random fit, which implies something is right. However, a significantly higher LLG score indicates that a model is significantly better than an alternative.

It could be very useful to implement a simple helical parameterisation of the model so that the helix parameters can be optimised. This would be possible but not trivial, so it is outside the scope of this paper. A rigid body fit has 6 degrees of freedom, so intuitively one would probably want to set the threshold for confidence about 10 log-likelihood units higher for every degree of freedom added.

Further write down in the sub. table 5: the relevant points from 1-3 for each of the fits.

Minor:

1). Good practise is to use the relion_helix_inimodel2d as done by the authors. One should refine first with a C1 symmetry with helical rise 4.7 and twist -1, until a clear separation in 21 symmetry or C2 is observed. Otherwise the

refinement can be biased by the applied a symmetry too early. It is assumee that the author followed this general procedure.

Line 693:

The helical rise was set to 2.4 Å for the PHF subset and 4.8 Å for the SF subsets, based on known structural data.

We followed this procedure that avoided bias: the first rounds of 3D classification for PHF and SF were performed without symmetry (C1) and rise of 4.8 Å until 2₁ screw symmetry was apparent for the PHF map.

2). Sub. Table 5: Estimated Defocus (µm), Typo should be in [Å].

We thank the referee for spotting this typographical error, which we have amended accordingly.